# Efficient Quadratic Corrections
# for Frank-Wolfe Algorithms

**Jannis Halbey**
Zuse Institute Berlin & TU Berlin
Berlin, Germany
`halbey@zib.de`

**Seta Rakotomandimby**
École nationale des ponts et chaussées, IP Paris
Champs-sur-Marne, France
`seta.rakotomandimby@enpc.fr`

**Mathieu Besançon**
Université Grenoble Alpes, Inria, LIG, CNRS
Grenoble, France
`mathieu.besancon@inria.fr`

**Sébastien Designolle**
Zuse Institute Berlin
Berlin, Germany
`designolle@zib.de`

**Sebastian Pokutta**
Zuse Institute Berlin & TU Berlin
Berlin, Germany
`pokutta@zib.de`

## Abstract

We develop a Frank-Wolfe algorithm with corrective steps, generalizing previous algorithms including Blended Conditional Gradients, Blended Pairwise Conditional Gradients, and Fully-Corrective Frank-Wolfe. For this, we prove tight convergence guarantees together with an optimal face identification property. Furthermore, we propose two highly efficient corrective steps for convex quadratic objectives based on linear optimization or linear system solving, akin to Wolfe's Minimum-Norm Point algorithm, and prove finite-time convergence under suitable conditions. Beyond optimization problems that are directly quadratic, we revisit two algorithms, Split Conditional Gradient and Second-Order Conditional Gradient Sliding, which can leverage quadratic corrections to accelerate the solution of their quadratic subproblems. We show improved convergence rates for the first and prove broader applicability for the second. Finally, we demonstrate substantial computational speedups for Frank-Wolfe-based algorithms with quadratic corrections across the considered problem classes.

## 1 Introduction

In this paper, we consider convex constrained optimization problems of the form

$$\min_{\mathbf{x} \in \mathcal{X}} f(\mathbf{x}),$$

where $\mathcal{X}$ is a compact, convex set and $f$ is a convex, differentiable function. A particularly interesting family of first-order methods for this setting is the class of Conditional Gradient (CG) or Frank-Wolfe (FW) methods [Levitin and Polyak, 1966, Frank and Wolfe, 1956]. One major advantage of these methods is that they access $\mathcal{X}$ only through a Linear Minimization Oracle (LMO), which solves linear subproblems over the set. The methods then construct solutions as convex combinations of the vertices returned by the LMO. In particular, these methods avoid costly projection steps while ensuring an $\mathcal{O}(1/t)$ convergence rate in general and linear convergence rates in specific settings.

39th Conference on Neural Information Processing Systems (NeurIPS 2025).

Over the years, several FW variants have been designed to exploit the so-called *active set*, i.e., the vertices used to form the current iterate, the oldest and best-known variant being the Away-step Frank-Wolfe (AFW) [Guélat and Marcotte, 1986]. By maintaining the convex combination, these methods significantly reduce the number of LMO calls. Furthermore, they achieve accelerated convergence rates on polytopes for sharp functions, whereas standard FW remains at a rate of $\Omega(1/t)$. Additionally, they have been shown to produce sparse solutions (constructed as convex combinations of a few iterates), which motivated strong interest in FW methods for sparse optimization and machine learning applications. The Fully-Corrective Frank-Wolfe (FCFW) algorithm, presented in some form in Wolfe [1976], Holloway [1974] and named in Lacoste-Julien and Jaggi [2015], drives the corrective paradigm to its limit by computing a minimizer of the objective over the convex hull of the active set after each FW step. Besides its favorable convergence properties and sparsity of solutions, FCFW is also closely related to column-generation and cutting-plane methods [Vinyes and Obozinski, 2017, Zhou et al., 2018]. Nonetheless, FCFW is often computationally impractical since minimizing over the convex hull of the active set can be computationally as hard as the original problem.

In this work, we generalize the idea of corrective steps on the active set to a new framework, *Corrective Frank-Wolfe (CFW)*, and design two new *Quadratic Correction (QC)* steps tailored to quadratic objective functions. While many applications involve quadratic objectives, such as matrix recovery [Garber et al., 2019], spectral clustering [Ding et al., 2022], and entanglement detection [Liu et al., 2025], we also revisit two FW-based algorithms that benefit from QC and strengthen their theoretical guarantees (with improved rates and applicability to new function classes).

*Split Conditional Gradient (SCG)* [Woodstock and Pokutta, 2025] tackles optimization problems over finite intersections of convex sets via splitting, avoiding the potential intractable linear minimization over the intersection. SCG is inspired by the Alternating Linear Minimization (ALM) algorithm [Braun et al., 2023] and solves the problem by alternating between update steps for iterates on the respective sets. Each iterate minimizes an auxiliary quadratic objective, incurring an increasing penalty on the squared distance to the other iterate. Besides the computational benefits of the quadratic corrections, we also prove a faster convergence rate for SCG.

Additionally, we consider *Second-Order Conditional Gradient Sliding (SOCGS)* [Carderera and Pokutta, 2020] which follows an inexact Newton approach, adapting the Conditional Gradient Sliding (CGS) method of Lan and Zhou [2016] with a quadratic approximation of the original problem at each iteration. The method enjoys very fast convergence rates and has been shown to outperform first-order Frank-Wolfe variants on various problems. However, the main computational bottleneck of SOCGS is solving the quadratic subproblems, which can be significantly accelerated using our proposed quadratic corrections. Furthermore, we establish a global linear convergence rate of SOCGS for generalized self-concordant functions [Sun and Tran-Dinh, 2019] without requiring global Lipschitz smoothness or strong convexity.

**Related work**

The literature on Frank-Wolfe methods is vast, and we only cover a few related methods here; for a more comprehensive overview, we refer to Braun et al. [2022]. The corrective framework is inspired by ideas from Blended Conditional Gradients (BCG) [Braun et al., 2019] and Blended Pairwise Conditional Gradients (BPCG) [Tsuji et al., 2022]. Furthermore, Lacoste-Julien and Jaggi [2015] present, besides the already mentioned AFW and FCFW, Pairwise Frank-Wolfe (PFW) and an approximate variant of FCFW. Additionally, they introduce a generalized version of the Minimum-Norm Point (MNP) algorithm [Wolfe, 1976] for corrections in approximate FCFW, called MNP-Correction. Finally, we follow the idea of Braun et al. [2017] to introduce a lazified variant of the corrective step framework.

**Contributions**

Our contributions can be summarized as follows:

**Corrective Frank-Wolfe**   We introduce Corrective Frank-Wolfe (CFW), a new framework for FW variants that perform corrective steps on the active set and prove linear convergence for smooth, sharp functions over polytopes. We show that active-set-based methods like AFW, BCG, and BPCG fit into the corrective step framework. Furthermore, we introduce a lazified variant of CFW, which avoids

LMO calls during iterations with corrective steps. Lastly, we prove that CFW identifies the optimal face of a polytope in finite time under strict complementarity.

**Quadratic corrections**   We introduce two new quadratic correction steps tailored for quadratic objective functions. The first one, QC-LP, is a direct linear program solving the quadratic problem over the convex hull of the active set via relaxed optimality conditions. The second type, QC-MNP, is a specialized variant of the Minimum-Norm Point (MNP) algorithm [Wolfe, 1976], solving the quadratic problem over the affine hull. Both steps are approximate variants of an FCFW step, but require only solving a linear program or linear system, respectively. Additionally, we propose a hybrid method that combines these corrections with the local pairwise steps of BPCG [Tsuji et al., 2022], yielding a highly efficient update scheme for quadratic problems. Finally, we show that CFW with QC-LP and QC-MNP converges in finite time.

**Theoretical results for SCG and SOCGS**   Besides the practical benefits of the new quadratic corrections for SCG and SOCGS, we also present new theoretical results. First, we prove that SCG converges for a smaller step size, yielding a faster rate of $\mathcal{O}(1/\sqrt{t})$ compared to the original rate of $\mathcal{O}(\log(t)/\sqrt{t})$ provided in Woodstock and Pokutta [2025]. This closes the gap to the complexity bound of $\mathcal{O}(1/\sqrt{t})$ for the underlying non-smooth problem [Yudin, 1983]. Second, we show that the Projected Variable-Metric method in SOCGS converges globally for generalized self-concordant functions.

**Experiments**   We demonstrate the excellent computational benefits of the new quadratic corrections on a variety of problems, including sparse regression, entanglement detection, projections onto the Birkhoff polytope, and tensor completion problems. Further experiments show that quadratic corrections also improve the convergence of ALM and SCG. For SOCGS, the quadratic corrections enable us to solve the quadratic subproblem efficiently.

**Preliminaries**

Let $\mathcal{X}$ be a compact convex set with diameter $D \stackrel{\text{def}}{=} \max_{\mathbf{x}, \mathbf{y} \in \mathcal{X}} \|\mathbf{x} - \mathbf{y}\|$, $\mathcal{X}^*$ be the set of minimizers of $f$ over $\mathcal{X}$, and $V(\mathcal{X})$ be the set of extreme points of $\mathcal{X}$. If $\mathcal{X}$ is a polytope, we denote its pyramidal width by $\delta$ [Lacoste-Julien and Jaggi, 2015] and the minimal face containing $\mathcal{X}^*$ by $\mathcal{F}^*$. Strict complementarity holds if there exists $\rho > 0$ such that for all $\mathbf{x}^* \in \mathcal{X}^*$ and $\mathbf{v} \in V(\mathcal{X})$, we have

$$\langle \nabla f(\mathbf{x}^*), \mathbf{v} - \mathbf{x}^* \rangle \begin{cases} \geqslant \rho & \text{if } \mathbf{v} \in V(\mathcal{X}) \setminus \mathcal{F}^*, \\ = 0 & \text{if } \mathbf{v} \in V(\mathcal{X}) \cap \mathcal{F}^*. \end{cases}$$

We denote the active set, i.e., the set of vertices used by an FW algorithm, by $S \subset \mathcal{X}$. For a given weight vector $\boldsymbol{\lambda}$, we denote the weight of an atom $\mathbf{v} \in S$ by $\boldsymbol{\lambda}_{\mathbf{v}}$. For a given convex combination $\mathbf{x} = \sum_{\mathbf{v} \in S} \boldsymbol{\lambda}_{\mathbf{v}} \mathbf{v}$, we denote the corresponding weight vector by $\boldsymbol{\lambda}(\mathbf{x}) \stackrel{\text{def}}{=} \boldsymbol{\lambda}$. Furthermore, we denote the convex hull of $S$ by $\text{conv}(S)$, the affine hull of $S$ by $\text{aff}(S)$, and the boundary of a set $M$ by $\partial M$.

Let $f \colon \mathcal{X} \to \mathbb{R}$ be a differentiable, convex function. The function $f$ is *L-smooth* if

$$f(\mathbf{y}) - f(\mathbf{x}) \leqslant \langle \nabla f(\mathbf{x}), \mathbf{y} - \mathbf{x} \rangle + \frac{L}{2} \|\mathbf{y} - \mathbf{x}\|^2 \quad \forall \mathbf{x}, \mathbf{y} \in \mathcal{X}.$$

The function $f$ is *$\mu$-strongly convex* if

$$f(\mathbf{y}) - f(\mathbf{x}) \geqslant \langle \nabla f(\mathbf{x}), \mathbf{y} - \mathbf{x} \rangle + \frac{\mu}{2} \|\mathbf{y} - \mathbf{x}\|^2 \quad \forall \mathbf{x}, \mathbf{y} \in \mathcal{X}.$$

In the following, we will use the notion of sharpness, generalizing strong convexity. Let $f^*$ denote the optimal value of $f$ over $\mathcal{X}$. A proper convex function $f$ is *$(c, \theta)$-sharp* with $c > 0$ and $\theta \in (0, 1]$ if for all $\mathbf{x} \in \mathcal{X}$,

$$c\big(f(\mathbf{x}) - f^*\big)^\theta \geqslant \min_{\mathbf{y} \in \mathcal{X}^*} \|\mathbf{x} - \mathbf{y}\|.$$

In particular, a $\mu$-strongly convex function is $\left(\frac{\sqrt{2}}{\sqrt{\mu}}, \frac{1}{2}\right)$-sharp over any compact set.

## 2 Corrective Frank-Wolfe

We now present the Corrective Frank-Wolfe (CFW) algorithm in Algorithm 1, its convergence analysis, and specialized corrections for quadratic objectives.

CFW provides a general framework for FW methods that perform corrective steps on the active set. Unlike approximate FCFW, CFW does not perform an FW step in each iteration but instead compares the so-called local pairwise gap $\langle \nabla f(\mathbf{x}_t), \mathbf{a}_t - \mathbf{s}_t \rangle$ with the global Frank-Wolfe gap $\langle \nabla f(\mathbf{x}_t), \mathbf{x}_t - \mathbf{v}_t \rangle$ to decide whether to perform a corrective step or an FW step. Algorithm 2 is a template for corrective steps that ensures convergence of CFW.

---

**Algorithm 1** Corrective Frank-Wolfe (CFW)

---

**Require:** convex smooth function $f$, starting point $\mathbf{x}_0 \in V(\mathcal{X})$.
1: $S_0 \leftarrow \{\mathbf{x}_0\}$
2: **for** $t = 0$ to $T - 1$ **do**
3:      $\mathbf{a}_t \leftarrow \arg\max_{\mathbf{v} \in S_t} \langle \nabla f(\mathbf{x}_t), \mathbf{v} \rangle$                              ▷ away vertex
4:      $\mathbf{s}_t \leftarrow \arg\min_{\mathbf{v} \in S_t} \langle \nabla f(\mathbf{x}_t), \mathbf{v} \rangle$                                ▷ local FW
5:      $\mathbf{v}_t \leftarrow \arg\min_{\mathbf{v} \in V(\mathcal{X})} \langle \nabla f(\mathbf{x}_t), \mathbf{v} \rangle$                        ▷ global FW
6:      **if** $\langle \nabla f(\mathbf{x}_t), \mathbf{a}_t - \mathbf{s}_t \rangle \geqslant \langle \nabla f(\mathbf{x}_t), \mathbf{x}_t - \mathbf{v}_t \rangle$ **then**
7:          $\mathbf{x}_{t+1}, S_{t+1} \leftarrow \text{CS}(S_t, \mathbf{x}_t, \mathbf{a}_t, \mathbf{s}_t)$
8:      **else**
9:          $\gamma_t \leftarrow \arg\min_{\gamma \in [0,1]} f(\mathbf{x}_t - \gamma(\mathbf{x}_t - \mathbf{v}_t))$
10:          $\mathbf{x}_{t+1} \leftarrow \mathbf{x}_t - \gamma_t(\mathbf{x}_t - \mathbf{v}_t)$
11:          $S_{t+1} \leftarrow S_t \cup \{\mathbf{v}_t\}$
12:      **end if**
13: **end for**

---

**Algorithm 2** Corrective Step $\text{CS}(S, \mathbf{x}, \mathbf{a}, \mathbf{s})$

---

**Require:** $S \subset \mathcal{X}, \mathbf{x}, \mathbf{a}, \mathbf{s} \in \mathcal{X}$
**Return:** $S' \subseteq S, \mathbf{x}' \in \text{conv}(S')$ satisfying
    (i) $f(\mathbf{x}') \leqslant f(\mathbf{x})$ and $S' \subsetneq S$ or                                         ▷ drop step
    (ii) $f(\mathbf{x}) - f(\mathbf{x}') \geqslant \frac{\langle \nabla f(\mathbf{x}), \mathbf{a} - \mathbf{s} \rangle^2}{2LD^2}$                       ▷ descent step

---

A corrective step is either a descent step that yields sufficient primal progress or a drop step that decreases the size of the active set without increasing the objective value. In contrast to approximate FCFW, we omit any conditions on the away gap and compare the primal progress only to the local pairwise gap rather than the global FW step. The update steps in BCG, BPCG, and FCFW meet the criteria of a corrective step. The proof is given in Section A.

**Proposition 1.** *Algorithm 3, Algorithm 4, and the simplex gradient descent step from Braun et al. [2019] satisfy the criteria of Algorithm 2.*

---

**Algorithm 3** Local Pairwise Step $\text{LPS}(S, \mathbf{x}, \mathbf{a}, \mathbf{s})$

---

**Require:** $S \subset \mathcal{X}, \mathbf{x}, \mathbf{a}, \mathbf{s} \in \mathcal{X}$
**Return:** $S', \mathbf{x}'$
1: $\gamma^* \leftarrow \arg\min_{\gamma \in [0, \boldsymbol{\lambda}_{\mathbf{a}}(\mathbf{x})]} f(\mathbf{x} + \gamma(\mathbf{s} - \mathbf{a}))$
2: $\mathbf{x}' \leftarrow \mathbf{x} + \gamma^*(\mathbf{s} - \mathbf{a})$
3: $S' \leftarrow \begin{cases} S \setminus \{\mathbf{a}\} & \text{if } \gamma^* = \boldsymbol{\lambda}_{\mathbf{a}}(\mathbf{x}), \\ S & \text{otherwise.} \end{cases}$

---

**Algorithm 4** Fully-Corrective Step $\text{FCS}(S)$

---

**Require:** $S \subset \mathcal{X}$
**Return:** $S', \mathbf{x}'$
1: $\mathbf{x}' \leftarrow \arg\min_{\mathbf{x} \in \text{conv}(S)} f(\mathbf{x})$
2: $S' \leftarrow \{\mathbf{v} \in S \mid \boldsymbol{\lambda}_{\mathbf{v}}(\mathbf{x}') > 0\}$

---

This result enables designing new corrective steps without theoretical verification. The conditions of a drop step can be verified easily at runtime. For the descent step, one can compare the primal value of the new point with the result of the local pairwise step. If neither of the conditions is met, one can perform the local pairwise step, which provides an inexpensive fallback option. Furthermore, one can also use hybrid methods that combine different steps to balance computational effort and progress.

*Remark* 1. The result of Proposition 1 also applies to AFW if the gap comparison in Algorithm 1 uses the away gap instead of the local pairwise gap. We choose the pairwise gap since it can be computed with the same complexity and leads to fewer FW steps.

## 2.1 Convergence analysis

In the following, we will state two convergence results for CFW and its lazified variant, as well as a result on active set identification. The proofs are independent of the specific implementation of the corrective step, thereby simplifying the verification of these properties for new variants.

**Theorem 2.** *Let $\mathcal{X}$ be a convex feasible set with diameter $D$ and $f$ be a convex, $L$-smooth function over $\mathcal{X}$. Consider the sequence $\{\mathbf{x}_t\}_{t=0}^T \subset \mathcal{X}$ obtained by Algorithm 1. Then we have*

$$f(\mathbf{x}_T) - f^* \leqslant \frac{4LD^2}{T}. \tag{1}$$

*If additionally $f$ is $(c, \frac{1}{2})$-sharp and $\mathcal{X}$ is a polytope with pyramidal width $\delta$, then*

$$f(\mathbf{x}_T) - f^* \leqslant (f(\mathbf{x}_0) - f^*) \exp(-c_{f,\mathcal{X}} T), \tag{2}$$

*where $c_{f,\mathcal{X}} \stackrel{\text{def}}{=} \min\left\{\frac{1}{4}, \frac{\delta^2}{16Lc^2 D^2}\right\}$.*

The proof is given in Section A.

The CFW algorithm needs to call the LMO at each iteration to compare the FW gap with the local pairwise gap, even if the FW step is not taken after all. We follow the approach of Braun et al. [2017] and adopt a lazified version of CFW to avoid this requirement.

The Lazified Corrective Frank-Wolfe (LCFW) method replaces the FW gap with an estimate $\Phi_t$, which is updated if the gap becomes smaller than $\Phi_t/J$ for some parameter $J \geqslant 1$. The convergence results are stated in Theorem 3, and we present its proof together with the algorithm in Section B.

**Theorem 3.** *Let $\mathcal{X}$ be a convex feasible set with diameter $D$ and $f$ be a convex, $L$-smooth function over $\mathcal{X}$. Consider the sequence $\{\mathbf{x}_t\}_{t=0}^T \subset \mathcal{X}$ obtained by the Algorithm 7. Then we have*

$$f(\mathbf{x}_T) - f^* = \mathcal{O}\left(\frac{1}{T}\right). \tag{3}$$

*If additionally $f$ is $(c, \frac{1}{2})$-sharp and $\mathcal{X}$ is a polytope with pyramidal width $\delta$, then*

$$f(\mathbf{x}_T) - f^* = \mathcal{O}\left(\exp(-aT)\right), \tag{4}$$

*for some constant $a > 0$ independent of $T$.*

Finally, we state a theorem that ensures identification of the optimal face by CFW in finite time under strict complementarity. Identifying the optimal face $\mathcal{F}^*$ of $\mathcal{X}$ is crucial, as it simplifies the optimization problem to one over $\mathcal{F}^*$ instead of $\mathcal{X}$. An early result by Guélat and Marcotte [1986] proves this property for AFW under the additional assumption of strong convexity of $f$. Bomze et al. [2020] extend this result to AFW for general convex functions and Wirth et al. [2025] prove the same result for BPCG. The proof of Theorem 4 is given in Section A.

**Theorem 4.** *Let $\mathcal{X}$ be a polytope and let $f$ be a convex, $L$-smooth function over $\mathcal{X}$. Assume that strict complementarity holds. Consider the sequence $\{\mathbf{x}_t\}_{t=0}^T$ generated by Algorithm 1. Then there exists an iteration $\tilde{T}$ such that*

$$\mathbf{x}_t \in \mathcal{F}^* \quad \text{for all } t \geqslant \tilde{T}.$$

## 2.2 Quadratic corrections

In this section, we introduce two types of corrective steps for the CFW algorithm that are especially suited for quadratic problems: Quadratic Correction LP (QC-LP) and Quadratic Correction MNP (QC-MNP). Here, we assume that $f$ is a convex quadratic function, i.e., can be written as $f(\mathbf{x}) = \frac{1}{2}\langle \mathbf{x}, \mathbf{A}\mathbf{x}\rangle + \langle \mathbf{b}, \mathbf{x}\rangle + c$, where $\mathbf{A}$ is a symmetric positive semidefinite matrix. This can be extended to general finite-dimensional Hilbert spaces, in which case $\mathbf{A}$ is a symmetric linear operator.

The quadratic corrections are inspired by FCFW, which solves the minimization problem $\min_{\mathbf{x} \in \text{conv}(S)} f(\mathbf{x})$ in each iteration. Unfortunately, minimizing quadratic functions over polytopes such as $\text{conv}(S)$ is computationally demanding (we discuss this in Remark 2). To circumvent this difficulty, the QC algorithms instead exploit the finiteness of the active set $S$ and relax this problem into minimizing the function $f$ over the *affine* hull of $S$,

$$\min_{\mathbf{x} \in \text{aff}(S)} f(\mathbf{x}) \left(\leqslant \min_{\mathbf{x} \in \text{conv}(S)} f(\mathbf{x})\right), \tag{5}$$

and then reconstruct a point in $\text{conv}(S)$.

**Solving the minimization problem over the affine hull**  Let $\mathbf{x}^*_{\text{aff}} \in \operatorname{argmin}_{\mathbf{x} \in \text{aff}(S)} f(\mathbf{x})$. First-order optimality conditions imply that $\langle \nabla f(\mathbf{x}^*_{\text{aff}}), \mathbf{d} \rangle = 0$ for any feasible direction $\mathbf{d}$. The feasible directions in $\text{aff}(S)$ are spanned by $\mathbf{d} = \mathbf{v} - \mathbf{w}$ for $\mathbf{v}, \mathbf{w} \in S$. Thus, the first-order optimality conditions are equivalent to the system of equalities,

$$\langle \nabla f(\mathbf{x}^*_{\text{aff}}), \mathbf{v} - \mathbf{w} \rangle = 0 \quad \forall \, \mathbf{v}, \mathbf{w} \in S,$$

which is equivalent to the gradient at $\mathbf{x}^*_{\text{aff}}$ being orthogonal to the affine space. Writing $\mathbf{x}^*_{\text{aff}}$ as an affine combination of the atoms in $S$ yields $\nabla f(\mathbf{x}^*_{\text{aff}}) = \mathbf{A}\mathbf{x}^*_{\text{aff}} + \mathbf{b} = \mathbf{A} \sum_{\mathbf{v} \in S} \boldsymbol{\lambda}_{\mathbf{v}}(\mathbf{x}^*_{\text{aff}}) \mathbf{v} + \mathbf{b} = \mathbf{A}\mathbf{V}\boldsymbol{\lambda}(\mathbf{x}^*_{\text{aff}}) + \mathbf{b}$. By fixing an anchor $\mathbf{w} \in S$, we obtain a linear system,

$$\langle \mathbf{A}\mathbf{V}\boldsymbol{\lambda} + \mathbf{b}, \mathbf{v} - \mathbf{w} \rangle = 0 \quad \forall \, \mathbf{v} \in S \setminus \{\mathbf{w}\}, \quad \sum_{\mathbf{v} \in S} \boldsymbol{\lambda}_{\mathbf{v}} = 1. \tag{6}$$

*Remark* 2. Applying first-order optimality conditions to $\mathbf{x}^*_{\text{conv}} \in \arg\min_{\mathbf{x} \in \text{conv}(S)} f(\mathbf{x})$ with the barycentric coordinates yields $\langle \mathbf{A}\mathbf{V}\boldsymbol{\lambda} + \mathbf{b}, \mathbf{v} - \mathbf{V}\boldsymbol{\lambda} \rangle \geqslant 0$ for all $\mathbf{v} \in S$, which is a quadratic inequality system in $\boldsymbol{\lambda}$ that can be as hard to solve as the original problem. However, if $\mathbf{x}^*_{\text{conv}}$ lies in the relative interior of $\text{conv}(S)$, these quadratic inequalities simplify to linear equalities as in (6).

The linear system (6) is equivalent to the KKT conditions of (5) and thus sufficient. As formalized in the following proposition, an affine minimizer exists if the linear term $\mathbf{b}$ yields no descent in any direction in the affine hull in which $f$ is not curved. Otherwise, the problem is unbounded and the system is infeasible. The proof is given in Section C.2.

**Proposition 5.** *Equation* (6) *is feasible if and only if* $\mathbf{b} \perp \text{span}(S) \cap \ker(\mathbf{A})$. *In particular, it is feasible if* $\mathbf{A}$ *is positive definite.*

**Quadratic correction through Linear Programs**  To ensure that the new weights $\boldsymbol{\lambda}$ are non-negative, we consider two approaches. The QC-LP approach enforces this directly, yielding

$$\langle \mathbf{A}\mathbf{V}\boldsymbol{\lambda} + \mathbf{b}, \mathbf{v} - \mathbf{w} \rangle = 0 \quad \forall \, \mathbf{v} \in S \setminus \{\mathbf{w}\}, \quad \sum_{\mathbf{v} \in S} \boldsymbol{\lambda}_{\mathbf{v}} = 1, \quad \boldsymbol{\lambda} \geqslant 0. \tag{7}$$

The LP is only feasible if the affine minimizer exists and lies in $\text{conv}(S)$. Otherwise, we perform a local pairwise step as a computationally cheap fallback option. The method is stated in Algorithm 5.

---

**Algorithm 5** Quadratic Correction LP

**Require:** $S \subset \mathcal{X}, \mathbf{x}, \mathbf{a}, \mathbf{s} \in \mathcal{X}$
**Return:** $S', \mathbf{x}'$
  $\boldsymbol{\lambda}' \leftarrow$ Solve (7)
  **if** feasible **then**         ▷ FCFW step
    $\mathbf{x}' \leftarrow \sum_{\mathbf{v} \in S} \boldsymbol{\lambda}'_{\mathbf{v}} \mathbf{v}$
    $S' \leftarrow S$
  **else**              ▷ Fallback step
    $S', \mathbf{x}' \leftarrow \text{LPS}(S, \mathbf{x}, \mathbf{a}, \mathbf{s})$
  **end if**

---

**Algorithm 6** Quadratic Correction MNP

**Require:** $S \subset \mathcal{X}, \mathbf{x}, \mathbf{a}, \mathbf{s} \in \mathcal{X}$
**Return:** $S', \mathbf{x}'$
  $\tilde{\boldsymbol{\lambda}} \leftarrow$ Solve (6)
  **if** feasible **then**
    **if** $\tilde{\boldsymbol{\lambda}} \geqslant 0$ **then**         ▷ FCFW step
      $\mathbf{x}' \leftarrow \sum_{\mathbf{v} \in S} \tilde{\boldsymbol{\lambda}}_{\mathbf{v}} \mathbf{v}$
      $S' \leftarrow S$
    **else**            ▷ MNP step
      $\tau \leftarrow \min \left\{ \frac{\boldsymbol{\lambda}_{\mathbf{v}}(\mathbf{x})}{\boldsymbol{\lambda}_{\mathbf{v}}(\mathbf{x}) - \tilde{\boldsymbol{\lambda}}_{\mathbf{v}}} \Big| \tilde{\boldsymbol{\lambda}}_{\mathbf{v}} < \boldsymbol{\lambda}_{\mathbf{v}}(\mathbf{x}) \right\}$
      $\mathbf{x}' \leftarrow \sum_{\mathbf{v} \in S} (\tau \tilde{\boldsymbol{\lambda}}_{\mathbf{v}} + (1 - \tau)\boldsymbol{\lambda}_{\mathbf{v}}(\mathbf{x})) \mathbf{v}$
      $S' \leftarrow \{\mathbf{v} \in S \mid \boldsymbol{\lambda}_{\mathbf{v}}(\mathbf{x}') > 0\}$
    **end if**
  **else**              ▷ Fallback step
    $S', \mathbf{x}' \leftarrow \text{LPS}(S, \mathbf{x}, \mathbf{a}, \mathbf{s})$
  **end if**

---

**Quadratic correction through the Minimum-Norm-Point algorithm**  The second approach, QC-MNP, is inspired by the Minimum-Norm-Point algorithm in Wolfe [1976]. Instead of enforcing the non-negativity directly, one considers the line segment between the weights of the current point and the affine minimizer. If at least one of the new weights is negative, we perform a ratio test to find the intersection of the line segment with the $|S|$-simplex. This step will drop at least one atom from the active set, improving sparsity. However, it can happen that this atom belongs to the optimal face. Unlike the MNP-Correction step from Lacoste-Julien and Jaggi [2015], QC-MNP will stop after a

single truncation, alleviating the need to solve multiple linear systems in a single iteration. This also avoids dropping more than one atom per iteration, which can be problematic in some cases. As for QC-LP, we perform a local pairwise step if there is no affine minimizer. The standard implementation of QC-MNP is presented in Algorithm 6, and we derive a more efficient variant for the case of multiple affine minimizers in Section C.1.

Both QC-LP and QC-MNP are corrective steps in the sense of Algorithm 2. Furthermore, an affine minimizer is guaranteed to exist if the current iterate lies in the optimal face.

**Proposition 6.** *QC-LP and QC-MNP satisfy the criteria of corrective steps in Algorithm 2.*

**Proposition 7.** *Let $\mathcal{X}$ be a polytope and $\mathcal{F}^*$ be the minimal face of $\mathcal{X}$ containing the optimal points $\mathcal{X}^*$. If $S \subset \mathcal{F}^*$, then $\operatorname{argmin}_{x \in \operatorname{aff}(S)} f(x) \neq \emptyset$. If additionally there exists $\mathbf{x}^* \in \mathcal{X}^*$ with $\mathbf{x}^* \in \operatorname{conv}(S)$, then $\mathbf{x}^* \in \operatorname{argmin}_{x \in \operatorname{aff}(S)} f(x)$.*

We would like to emphasize that the sizes of the linear system and the LP do not depend on the ambient dimension but only on the size of the active set. Furthermore, similarly to Besançon et al. [2025], we can cache the scalar products $\langle \mathbf{v}, \mathbf{A}\mathbf{w} \rangle$ and $\langle \mathbf{b}, \mathbf{v} \rangle$ for $\mathbf{v}, \mathbf{w} \in S$, accelerating the setup of the linear system and LP while only marginally increasing storage.

*Remark* 3. The linear system in (6) is usually not symmetric, which can be problematic for solvers like Conjugate Gradient (CG). However, subtracting $\mathbf{W}^\top \mathbf{A}\mathbf{w}\mathbf{1}^\top \boldsymbol{\lambda} = \mathbf{W}^\top \mathbf{A}\mathbf{w}$ from (6) yields the equivalent symmetric system

$$\mathbf{W}^\top \mathbf{A}\mathbf{W}\boldsymbol{\mu} = -\mathbf{W}^\top (\mathbf{A}\mathbf{w} + \mathbf{b}), \tag{8}$$

where the matrix $\mathbf{W}$ has columns $\mathbf{v} - \mathbf{w}$ for $\mathbf{v} \in S \setminus \{\mathbf{w}\}$. The new weights are then given by

$$\boldsymbol{\lambda}_{\mathbf{v}} = \begin{cases} 1 - \mathbf{1}^\top \boldsymbol{\mu} & \text{if } \mathbf{v} = \mathbf{w}, \\ \boldsymbol{\mu}_{\mathbf{v}} & \text{otherwise.} \end{cases}$$

Finally, we come to the main result of this section. As seen in Theorem 4, CFW identifies the optimal face of a polytope in finitely many iterations if strict complementarity holds. This is crucial for fully-corrective steps as they converge to an optimal solution if the optimal face is identified and the convex hull of the active set contains at least one solution. Thus, CFW with QC-LP, QC-MNP or FCFW converges to an optimal solution in finitely many iterations. The proof is given in Section C.2.

**Theorem 8.** *Let $\mathcal{X}$ be a polytope and let $f$ be a convex, quadratic function over $\mathcal{X}$, i.e., $f(\mathbf{x}) = \frac{1}{2} \langle \mathbf{x}, \mathbf{A}\mathbf{x} \rangle + \langle \mathbf{b}, \mathbf{x} \rangle + c$ where $\mathbf{A}$ is a symmetric positive semidefinite matrix. Assume that strict complementarity holds. Consider the sequence $\{\mathbf{x}_t\}_{t=0}^T$ generated by Algorithm 1 with Algorithm 4, Algorithm 5, or Algorithm 8. Then, there exists an iteration $\tilde{T}$ such that $\mathbf{x}_{\tilde{T}} \in \mathcal{X}^*$.*

## 3 Accelerated algorithms through quadratic corrections

In this section, we leverage quadratic corrections for two classes of algorithms that require solving quadratic subproblems and provide additional theoretical results that might be of independent interest.

### 3.1 Split Conditional Gradient

A key limitation of Frank-Wolfe methods is their reliance on a Linear Minimization Oracle. This becomes particularly problematic for optimization over intersections of convex sets, where no efficient procedure for linear minimization is generally available. Braun et al. [2023] introduced the Alternating Linear Minimization (ALM) method for finding a point in the intersection of two convex sets $P$ and $Q$, drawing a parallel with von Neumann's alternating projection. ALM alternates between Frank-Wolfe steps, solving $\min_{\mathbf{x} \in P, \mathbf{y} \in Q} \|\mathbf{x} - \mathbf{y}\|^2$, and therefore avoids projections in an FW-like manner. Woodstock and Pokutta [2025] extended their approach to optimization problems over intersections by penalizing the distance,

$$\min_{\mathbf{x} \in P, \mathbf{y} \in Q} F_\lambda(\mathbf{x}, \mathbf{y}) = f\left(\frac{\mathbf{x} + \mathbf{y}}{2}\right) + \frac{\lambda}{2} \|\mathbf{x} - \mathbf{y}\|^2.$$

Their method, Split Conditional Gradient (SCG), performs parallel FW steps on both sets and can therefore be easily extended to use corrective steps. Particularly, if the original objective is linear or quadratic, one can leverage quadratic corrections.

Independently of CFW, we prove a tighter convergence rate for the original SCG. Woodstock and Pokutta [2025] prove that the primal gap of $F_\lambda$ converges at a rate of $\mathcal{O}\left(\log t/\sqrt{t}\right)$. By analyzing a different auxiliary objective, we show that replacing the step size $\gamma_t = \frac{2}{\sqrt{t+2}}$ by the smaller step size $\gamma_t = \frac{2}{\sqrt{t+2}\log(t+2)}$ improves the convergence rate of SCG to $\mathcal{O}\left(1/\sqrt{t}\right)$. The proof for the more general case of finite intersections over Hilbert spaces is provided in Section D.

**Theorem 9.** *Let $f\colon \mathbb{R}^n \to \mathbb{R}$ be convex and $L$-smooth, $P, Q \subset \mathbb{R}^n$ be non-empty compact and convex sets with diameters $D_P$ and $D_Q$ such that $P \cap Q \neq \emptyset$. For $t \geqslant 0$, set $\lambda_t = \ln(t+2)$, $(\mathbf{x}_t^*, \mathbf{y}_t^*) \in \operatorname{argmin}_{(\mathbf{x},\mathbf{y}) \in P \times Q} F_{\lambda_t}(\mathbf{x}, \mathbf{y})$. For $\gamma_t = \frac{2}{\sqrt{t+2}\ln(t+2)}$, the iterates of Algorithm 9 satisfy*

$$0 \leqslant F_{\lambda_t}(\mathbf{x}_t, \mathbf{y}_t) - F_{\lambda_t}(\mathbf{x}_t^*, \mathbf{y}_t^*) \leqslant \frac{(D_P + D_Q)^2(L+1) + \sqrt{2}c_f}{\sqrt{t+2}} \qquad (9)$$

*for all $t \geqslant 0$, where $c_f = \max_{(\mathbf{x},\mathbf{y}),(\tilde{\mathbf{x}},\tilde{\mathbf{y}}) \in P \times Q} f\left(\frac{\mathbf{x}+\mathbf{y}}{2}\right) - f\left(\frac{\tilde{\mathbf{x}}+\tilde{\mathbf{y}}}{2}\right) < \infty$.*

### 3.2 (Second-Order) Conditional Gradient Sliding

Another class of algorithms that benefit from quadratic corrections is sliding methods, specifically Conditional Gradient Sliding (CGS) [Lan and Zhou, 2016] and Second-Order Conditional Gradient Sliding (SOCGS) [Carderera and Pokutta, 2020]. These methods reduce the number of first-order oracle calls by iteratively solving quadratic problems with a FW variant. While the original papers considered vanilla FW and AFW steps, one can use quadratic corrections to accelerate the inner steps.

Unlike regular FW variants, CGS requires only $\mathcal{O}(1/\sqrt{\varepsilon})$ instead of $\mathcal{O}(1/\varepsilon)$ gradient calls to achieve $\varepsilon$-optimality, matching existing lower bounds. SOCGS achieves a linear rate for strongly convex functions over polytopes, replacing the Euclidean projection subproblem in CGS with a projected variable-metric problem. The convergence of CGS and SOCGS was analyzed on globally smooth, strongly convex functions. We extend this rate to generalized self-concordant functions, a detailed description is given in Section E.

**Theorem 10.** *Let $\mathcal{X}$ be a compact convex set of diameter $D$ and $f$ be a $(M, \nu)$-generalized self-concordant function with $\nu \geqslant 2$ such that $f$ is strongly convex on $\operatorname{dom}(f) \cap \mathcal{X}$ if $\nu = 3$. Given a starting point $\mathbf{x}_0 \in \mathcal{X} \cap \operatorname{dom}(f)$, the Algorithm 11 with a step size $\gamma_k$ guarantees for all $k \geqslant 0$:*

$$f(\mathbf{x}_{k+1}) - f(\mathbf{x}^*) \leqslant (1 - c(\gamma_k))\left(f(\mathbf{x}_k) - f(\mathbf{x}^*)\right),$$

*where $c(\gamma_k) < 1$ is a constant depending on the step size $\gamma_k$.*

## 4 Experiments

In this section, we present numerical experiments on two classical quadratic problems and on two applications of quadratic corrections to SCG and SOCGS. In all experiments, we use a hybrid of BPCG and QC steps, performing a QC step whenever $N$ new atoms have been added to the active set. In the first two experiments, we compare QC-LP and QC-MNP with four baselines: FW, AFW, PFW, and BPCG. For SCG and SOCGS, we compare against the original used FW variant and BPCG, to demonstrate the effectiveness of the quadratic corrections. Additional details and further experiments are provided in Section F. All runs were executed on a cluster with Intel Xeon Gold 6338 CPUs at 2 GHz and 12 GB RAM, with time and iteration limits chosen according to problem size.

### 4.1 Regression over the $K$-Sparse polytope

In the first experiment, we consider a sparse regression problem over the $K$-Sparse polytope, $P_K(\tau) = B_1(\tau K) \cap B_\infty(\tau)$. The problem is to minimize $f(\mathbf{x}) = \sum_{i=1}^m (\langle \mathbf{x}, \mathbf{a}_i \rangle - y_i)^2 = \|\mathbf{A}\mathbf{x} - \mathbf{y}\|_2^2$ over $P_K(\tau)$ given data points $\{(\mathbf{a}_i, y_i)\}_{i=1}^m \subset \mathbb{R}^n \times \mathbb{R}$. We generated synthetic normally distributed $\mathbf{a}_i$ and $y_i$ with $n = 500$, $m = 10000$ and $K \in \{5, 20\}$, and used a fixed interval length of $N = 10$ for the quadratic correction steps. As shown in Figure 1, all methods except FW converge linearly, with QC-LP and QC-MNP providing a substantial acceleration for both instances. For smaller values of $K$, the benefit of QC is more pronounced as the optimal face contains more atoms and therefore the active set is larger. Both QC methods reach optimality significantly faster than the baselines.

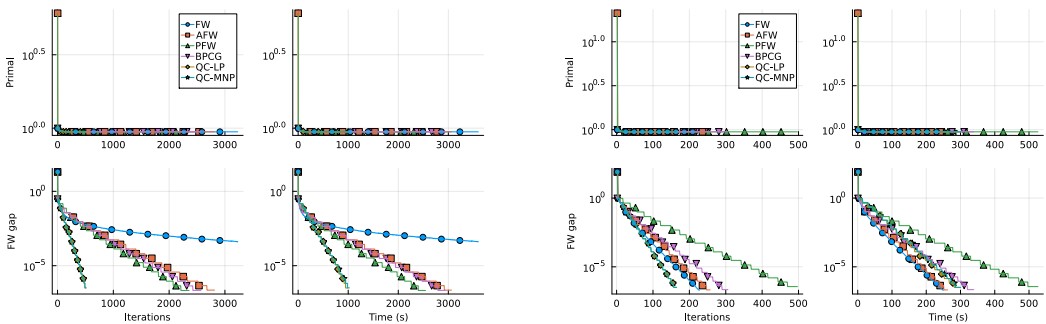

Figure 1: Sparse regression over the $K$-Sparse polytope for $K \in \{5, 20\}$

## 4.2 Entanglement detection

Certifying the entanglement of quantum states is a fundamental challenge in quantum information theory. Shang and Gühne [2018] use Gilbert's algorithm [Gilbert, 1966], a simplified FW method, together with a heuristic LMO for computing the minimal distance of a given state to the set of separable quantum states. Liu et al. [2025] extend their approach to general FW methods and propose an approximate LMO with a provable multiplicative error. For our experiments, we consider a family of $3 \times 3$ entangled states proposed in Horodecki [1997] and use a fixed interval length of $N = 1$ for QC-MNP and $N = 10$ for QC-LP. A complete description of the setup as well as results for noisy states are given in Section F. The results for $a \in \{0.25, 0.5\}$ are shown in Figure 2. We use a logarithmic scale on the horizontal axis to visualize the differences between the methods more clearly. QC-MNP shows a drastic initial acceleration, solving the problem to optimality in a fraction of the iterations and time of the other baselines. The LP in the QC-LP is rarely feasible, leading to pairwise steps and thus a similar trajectory to BPCG. FW and PFW perform worse than the other baselines in terms of time due to the high number of LMO calls.

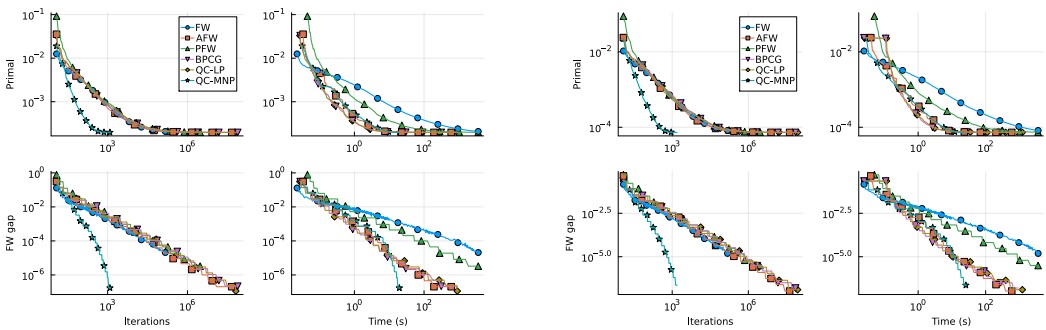

Figure 2: Entanglement detection for $a \in \{0.25, 0.5\}$

## 4.3 Projection onto intersections with Split Conditional Gradient

In this experiment, we use SCG to project onto the intersection of the Birkhoff polytope $B(n)$, the set of all $n \times n$ doubly stochastic matrices, and a shifted $\ell_2$ ball. For the shift, we sample a point on a face of $B(n)$ and move the center of the ball from there in the direction of the normal vector of the face. The shift is parametrized by $c \in [0, 1]$ and $q \in [0, 1]$; a complete description is given in the appendix. The objective is given by $f(\mathbf{X}) = \frac{1}{n^2}\|\mathbf{X} - \mathbf{X}_0\|_F^2$ where $\mathbf{X}_0 \in \mathbb{R}^{n \times n}$ is sampled with uniformly distributed entries. The results for $n \in \{300, 500\}$ using $c = 0.9$, $q = 0.1$ and $N = 1$ are given in Figure 3. We compare the hybrid methods QC-MNP and QC-LP with BPCG and the FW variant from [Woodstock and Pokutta, 2025]. The split approach, with its dynamically changing objective, makes this problem more difficult than the previous ones. Both QC methods hit numerical precision limits on these instances and would need adjusted tolerances for longer runs. Nevertheless, QC-MNP and QC-LP outperform BPCG for $n = 500$ and achieve similar results for $n = 300$. The FW variant from [Woodstock and Pokutta, 2025] performs the worst on both instances.

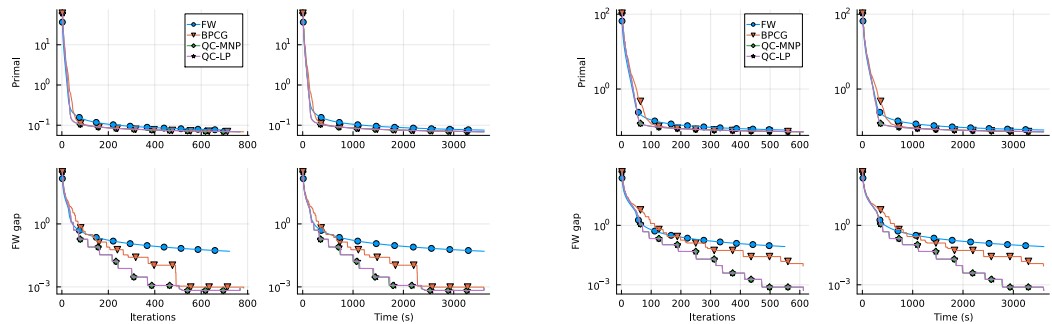

Figure 3: Projection onto the intersection of the Birkhoff polytope and a shifted $\ell_2$ ball for dimension $n \in \{300, 500\}$

## 4.4 Projecting with quadratic corrections in Second-Order Conditional Gradient Sliding

In the last experiment, we solve a logistic regression problem over the $\ell_1$ ball with SOCGS. The objective is to minimize $f(\mathbf{x}) = \frac{1}{m} \sum_{i=1}^{m} \ln \left(1 + \exp(-y_i \langle \mathbf{x}, \mathbf{z}_i \rangle)\right) + \frac{1}{2m} \|\mathbf{x}\|^2$. The labels $y_i \in \{-1, 1\}$ and feature vectors $\mathbf{z}_i \in \mathbb{R}^n$ are taken from the gisette training dataset [Guyon et al., 2007], with $n = 5000$ and $m = 6000$. Recall that each iteration of SOCGS corresponds to solving an outer problem and an inner problem. We use the BPCG step for the outer problem and compare BPCG, QC-MNP, and QC-LP for the inner step. Additionally, we also test AFW for the inner step as proposed in [Carderera and Pokutta, 2020]. The results for $k \in \{50, 200\}$ inner iterations are depicted in Figure 4. All methods follow an identical trajectory in the initial phase, indicating that SOCGS chooses only the outer step. Once the distance to the optimum is sufficiently small, the QC methods exhibit substantial improvements over BPCG in both instances. Further analysis reveals that QC-MNP and QC-LP perform more FW steps than BPCG on the inner problem by avoiding additional pairwise steps, which yields a drastic initial decrease in primal values and the FW gap. For longer runs of the inner problem, i.e., larger $k$, the acceleration of the QC methods is less pronounced, yielding a similar performance as AFW. The hybrid method with QC-MNP performs more quadratic corrections than QC-LP, yielding more significant acceleration but also longer runs in wall-clock time.

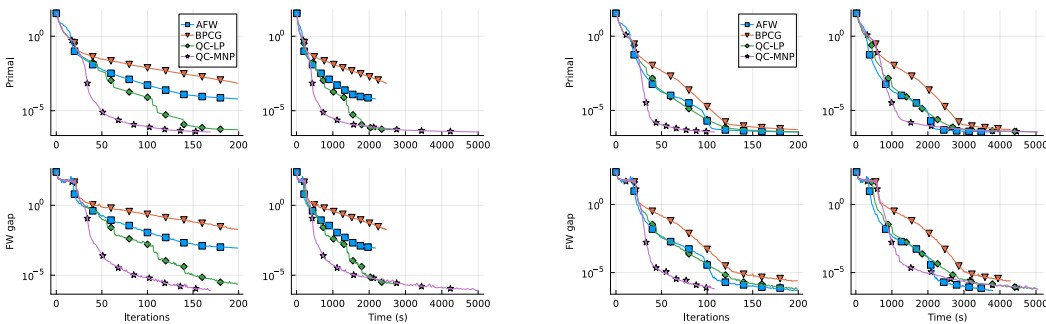

Figure 4: Logistic regression over the $\ell_1$-ball for maximum number of inner steps $k \in \{50, 200\}$

## 5 Conclusion

In this paper, we present a new framework for conditional gradient methods, Corrective Frank-Wolfe, that uses corrective steps on the active set to improve convergence while avoiding additional LMO calls. Instances of the proposed framework provide state-of-the-art convergence rates for sharp functions over polytopes and identify the optimal face of a polytope in finite time. The proposed quadratic corrections are very effective in practice, outperforming methods of comparable convergence both in iteration count and runtime. This allows us to revisit and accelerate previous algorithms, Split Conditional Gradient and Second-Order Conditional Gradient Sliding, by applying quadratic corrections to the arising subproblems.

## Acknowledgments and Disclosure of Funding

Research reported in this paper was partially supported by the Deutsche Forschungsgemeinschaft (DFG, German Research Foundation) under Germany's Excellence Strategy - The Berlin Mathematics Research Center MATH+ (EXC-2046/1, project ID: 390685689). The work of Mathieu Besançon was supported by MIAI at Université Grenoble Alpes (grant ANR-19-P3IA-0003).

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

# A   Proofs for Corrective Frank-Wolfe

In this section, we provide proofs for the theoretical results on CFW stated in Section 2. Before we present a more general version of Theorem 2 and its proof, we recall two lemmas resulting from the sharpness of the objective function and a conversion from contraction to convergence rates.

**Lemma 11.** *[Braun et al., 2022, Lemma 3.32] Let $\mathcal{X}$ be a polytope of pyramidal width $\delta > 0$ and let $f$ be a $(c, \theta)$-sharp convex function over $\mathcal{X}$. Let $\mathbf{a}$ and $\mathbf{v}$ be defined as in Algorithm 1, then*

$$\frac{1}{c}(f(\mathbf{x}) - f^*)^{1-\theta} \leqslant \frac{\langle \nabla f(\mathbf{x}), \mathbf{a} - \mathbf{v} \rangle}{\delta}. \tag{10}$$

**Lemma 12.** *[Braun et al., 2022, Lemma 2.21] Let $\{h_t\}_t$ be a sequence of positive numbers and let $c_0, c_1, c_2, \alpha$ be positive numbers with $c_1 < 1$ such that $h_1 \leqslant c_0$ and $h_t - h_{t+1} \geqslant h_t \min\{c_1, c_2 h_t^\alpha\}$ for all $t \geqslant 1$, then*

$$h_t \leqslant \begin{cases} c_0(1 - c_1)^{t-1} & \text{if } 1 \leqslant t \leqslant t_0 \\ \frac{(c_1/c_2)^{1/\alpha}}{(1 + c_1\alpha(t - t_0))^{1/\alpha}} = \mathcal{O}(1/t^{1/\alpha}) & \text{if } t \geqslant t_0, \end{cases}$$

*where*

$$t_0 \stackrel{\text{def}}{=} \max\left\{\left\lfloor \log_{1-c_1}\left(\frac{(c_1/c_2)^{1/\alpha}}{c_0}\right)\right\rfloor + 2, 1\right\}.$$

**Theorem 2.** *Let $\mathcal{X}$ be a convex feasible set with diameter $D$ and $f$ be a convex, $L$-smooth function over $\mathcal{X}$. Consider the sequence $\{\mathbf{x}_t\}_{t=0}^T \subset \mathcal{X}$ obtained by Algorithm 1. Then we have*

$$f(\mathbf{x}_T) - f^* \leqslant \frac{4LD^2}{T}. \tag{1}$$

*If additionally $f$ is $(c, \frac{1}{2})$-sharp and $\mathcal{X}$ is a polytope with pyramidal width $\delta$, then*

$$f(\mathbf{x}_T) - f^* \leqslant (f(\mathbf{x}_0) - f^*) \exp(-c_{f,\mathcal{X}}T), \tag{2}$$

*where $c_{f,\mathcal{X}} \stackrel{\text{def}}{=} \min\left\{\frac{1}{4}, \frac{\delta^2}{16Lc^2D^2}\right\}$.*

*Proof.* Let $T$ be the number of iterations of the algorithm, $T_{\text{FW}}$ the number of Frank-Wolfe steps, $T_{\text{desc}}$ the number of descent steps, and $T_{\text{drop}}$ the number of drop steps. Frank-Wolfe steps are the only steps that add vertices to the active set, and they do so one at a time. As a drop step reduces the active set by at least one vertex, we have $T_{\text{drop}} \leqslant T_{\text{FW}}$ and thus,

$$T = T_{\text{FW}} + T_{\text{drop}} + T_{\text{desc}} \leqslant 2T_{\text{FW}} + T_{\text{desc}} \leqslant 2(T_{\text{FW}} + T_{\text{desc}}). \tag{11}$$

Next, we bound the primal gap $h_t \stackrel{\text{def}}{=} f(\mathbf{x}_t) - f^*$ at iteration $t$.

First, we consider the case when a Frank-Wolfe step is taken. Using $L$-smoothness of $f$, the definition of the FW step $\mathbf{x}_{t+1} = \mathbf{x}_t + \gamma_t(\mathbf{v}_t - \mathbf{x}_t)$, and the diameter $D$ of the feasible region yields

$$\begin{aligned} h_t - h_{t+1} &= f(\mathbf{x}_t) - f(\mathbf{x}_{t+1}) \\ &\geqslant \gamma_t \langle \nabla f(\mathbf{x}_t), \mathbf{x}_t - \mathbf{v}_t \rangle - \frac{\gamma_t^2 L}{2} \|\mathbf{x}_t - \mathbf{v}_t\|^2 \\ &\geqslant \gamma_t \langle \nabla f(\mathbf{x}_t), \mathbf{x}_t - \mathbf{v}_t \rangle - \frac{\gamma_t^2 LD^2}{2}. \end{aligned} \tag{12}$$

As $\mathbf{v}_t \in \text{argmax}_{\mathbf{v} \in V(\mathcal{X})} \langle \nabla f(\mathbf{x}_t), \mathbf{x}_t - \mathbf{v} \rangle$ by definition, and by convexity of $f$, we get

$$h_t - h_{t+1} \geqslant \gamma_t \langle \nabla f(\mathbf{x}_t), \mathbf{x}_t - \mathbf{x}_t^* \rangle - \frac{\gamma_t^2 LD^2}{2} \geqslant \gamma_t h_t - \frac{\gamma_t^2 LD^2}{2}. \tag{13}$$

Let $\hat{\gamma}_t = \frac{h_t}{LD^2}$, which yields a lower bound on the progress made by the exact line search if $\hat{\gamma}_t \leqslant 1$. For (13) we have then

$$h_t - h_{t+1} \geqslant \hat{\gamma}_t h_t - \frac{\hat{\gamma}_t^2 LD^2}{2} = \frac{h_t^2}{2LD^2}. \tag{14}$$

If $\hat{\gamma}_t > 1$, we have by definition $h_t \geqslant LD^2$. We can bound the progress of the line search using $\gamma_t = 1$ in (13), which yields

$$h_t - h_{t+1} \geqslant h_t - \frac{LD^2}{2} \geqslant \frac{1}{2} h_t. \tag{15}$$

Combining (14), (15), we have

$$h_{t+1} \leqslant \begin{cases} h_t - \frac{h_t^2}{2LD^2} & \text{if } \hat{\gamma}_t \leqslant 1 \\ \frac{1}{2} h_t & \text{if } \hat{\gamma}_t > 1 \end{cases}. \tag{16}$$

Next, we consider the case when a descent step is taken. Using the criterion for performing a corrective step, the optimality of $\mathbf{v}_t$ and the convexity of $f$, we have

$$
\begin{aligned}
h_t - h_{t+1} = f(\mathbf{x}_t) - f(\mathbf{x}_{t+1}) &\geqslant \frac{\langle \nabla f(\mathbf{x}_t), \mathbf{a}_t - \mathbf{s}_t \rangle^2}{2LD^2} \\
&\geqslant \frac{\langle \nabla f(\mathbf{x}_t), \mathbf{x}_t - \mathbf{v}_t \rangle^2}{2LD^2} \\
&\geqslant \frac{\langle \nabla f(\mathbf{x}_t), \mathbf{x}_t - \mathbf{x}^* \rangle^2}{2LD^2} \\
&\geqslant \frac{h_t^2}{2LD^2}. 
\end{aligned} \tag{17}
$$

Together with (16), we get

$$h_{t+1} \leqslant \frac{2LD^2}{T_{\text{FW}} + T_{\text{desc}}},$$

which can be shown in the same way as in the proof of Corollary 4.2 in Braun et al. [2019], with $2LD^2$ replacing the value $4L_f$. Together with (11), we get

$$h_T \leqslant \frac{4LD^2}{T},$$

which proves (1).

Now we prove the accelerated convergence rate (2) with the additional assumptions that $f$ is $(c, \theta)$-sharp and $\mathcal{X}$ is a polytope. Again, we will bound the primal gap $h_t$ for different steps of the algorithm.

First, we consider the iterations when the Frank-Wolfe step is taken. By line 6 of Algorithm 1 we have

$$\langle \nabla f(\mathbf{x}_t), \mathbf{x}_t - \mathbf{v}_t \rangle \geqslant \langle \nabla f(\mathbf{x}_t), \mathbf{a}_t - \mathbf{s}_t \rangle \geqslant \langle \nabla f(\mathbf{x}_t), \mathbf{a}_t - \mathbf{x}_t \rangle.$$

Adding the left-hand side to both sides yields:

$$2 \langle \nabla f(\mathbf{x}_t), \mathbf{x}_t - \mathbf{v}_t \rangle \geqslant \langle \nabla f(\mathbf{x}_t), \mathbf{a}_t - \mathbf{v}_t \rangle. \tag{18}$$

Similar to the proof of the sublinear rate, we consider a primal progress bound induced by smoothness, specifically (12), and choose a specific step size, $\tilde{\gamma}_t = \frac{\langle \nabla f(\mathbf{x}_t), \mathbf{x}_t - \mathbf{v}_t \rangle}{LD^2}$. For the case $\tilde{\gamma}_t < 1$, we have

$$h_t - h_{t+1} \geqslant \frac{\langle \nabla f(\mathbf{x}_t), \mathbf{x}_t - \mathbf{v}_t \rangle^2}{2LD^2}. \tag{19}$$

Together with (18) and Lemma 11 we get

$$
\begin{aligned}
h_t - h_{t+1} &\geqslant \frac{\langle \nabla f(\mathbf{x}_t), \mathbf{x}_t - \mathbf{v}_t \rangle^2}{2LD^2} \\
&\geqslant \frac{\langle \nabla f(\mathbf{x}_t), \mathbf{a}_t - \mathbf{v}_t \rangle^2}{8LD^2} \\
&\geqslant \frac{\delta^2}{8c^2 LD^2} h_t^{2(1-\theta)}. 
\end{aligned} \tag{20}
$$

In the case $\tilde{\gamma}_t \geqslant 1$ we have $\langle \nabla f(\mathbf{x}_t), \mathbf{x}_t - \mathbf{v}_t \rangle \geqslant LD^2$. Again, we can bound the progress of the exact line search using $\gamma_t = 1$ in (12), which yields

$$
\begin{aligned}
h_t - h_{t+1} &\geqslant \langle \nabla f(\mathbf{x}_t), \mathbf{x}_t - \mathbf{v}_t \rangle - \frac{LD^2}{2} \\
&\geqslant \frac{\langle \nabla f(\mathbf{x}_t), \mathbf{x}_t - \mathbf{v}_t \rangle}{2} \tag{21} \\
&\geqslant \frac{\langle \nabla f(\mathbf{x}_t), \mathbf{x}_t - \mathbf{x}_t^* \rangle}{2} \geqslant \frac{1}{2} h_t. \tag{22}
\end{aligned}
$$

For the descent step, we have

$$
\langle \nabla f(\mathbf{x}_t), \mathbf{a}_t - \mathbf{s}_t \rangle \geqslant \langle \nabla f(\mathbf{x}_t), \mathbf{x}_t - \mathbf{v}_t \rangle \geqslant \langle \nabla f(\mathbf{x}_t), \mathbf{s}_t - \mathbf{v}_t \rangle .
$$

Adding $\langle \nabla f(\mathbf{x}_t), \mathbf{a}_t - \mathbf{s}_t \rangle$ to both sides yields

$$
2 \langle \nabla f(\mathbf{x}_t), \mathbf{a}_t - \mathbf{s}_t \rangle \geqslant \langle \nabla f(\mathbf{x}_t), \mathbf{a}_t - \mathbf{v}_t \rangle .
$$

Together with the definition of a descent step, line 6 of Algorithm 1 and again Lemma 11, we get

$$
\begin{aligned}
h_t - h_{t+1} &\geqslant \frac{\langle \nabla f(\mathbf{x}_t), \mathbf{a}_t - \mathbf{s}_t \rangle^2}{2LD^2} \\
&\geqslant \frac{\langle \nabla f(\mathbf{x}_t), \mathbf{a}_t - \mathbf{v}_t \rangle^2}{8LD^2} \\
&\geqslant \frac{\delta^2}{8c^2 LD^2} h_t^{2(1-\theta)}. \tag{23}
\end{aligned}
$$

Together with (20) and (22) we have

$$
h_t - h_{t+1} \geqslant h_t \min \left\{ \frac{1}{2}, \frac{\delta^2}{8c^2 LD^2} h_t^{1-2\theta} \right\}, \tag{24}
$$

for any iteration $t$ that is not a drop step. For $\theta = \frac{1}{2}$, this simplifies to

$$
h_{t+1} \leqslant \min \left\{ \frac{1}{2}, 1 - \frac{\delta^2}{8c^2 LD^2} \right\} h_t.
$$

By (11) and the fact that drop steps are non-increasing, i.e., $h_{t+1} \leqslant h_t$, we have

$$
\begin{aligned}
h_T &\leqslant h_0 \min \left\{ \frac{1}{2}, 1 - \frac{\delta^2}{8c^2 LD^2} \right\}^{T_{\text{FW}} + T_{\text{desc}}} \\
&\leqslant h_0 (1 - 2c_{f,\mathcal{X}})^{\frac{T}{2}} \\
&\leqslant h_0 \exp\left(-c_{f,\mathcal{X}} T\right),
\end{aligned}
$$

where $c_{f,\mathcal{X}} = \min \left\{ \frac{1}{4}, \frac{\delta^2}{16c^2 LD^2} \right\}$.

$\square$

Next, we prove that different types of steps arising in typical FW variants satisfy the requirement from the CFW framework.

**Proposition 1.** *Algorithm 3, Algorithm 4, and the simplex gradient descent step from Braun et al. [2019] satisfy the criteria of Algorithm 2.*

*Proof.* First, we consider the local pairwise step given in Algorithm 3. Let $\hat{\mathbf{x}} = \mathbf{x} - \hat{\gamma}(\mathbf{a} - \mathbf{s})$ be the point produced by a short step, i.e., $\hat{\gamma} = \frac{\langle \nabla f(\mathbf{x}), \mathbf{a} - \mathbf{s} \rangle}{L \|\mathbf{a} - \mathbf{s}\|^2}$. If $\hat{\gamma} \leqslant \gamma_{\max}$, we have by smoothness of $f$ that

$$
f(\mathbf{x}) - f(\hat{\mathbf{x}}) \geqslant \hat{\gamma} \langle \nabla f(\mathbf{x}), \mathbf{a} - \mathbf{s} \rangle - \frac{L\hat{\gamma}^2}{2} \|\mathbf{a} - \mathbf{s}\|^2 = \frac{\langle \nabla f(\mathbf{x}), \mathbf{a} - \mathbf{s} \rangle^2}{2L \|\mathbf{a} - \mathbf{s}\|^2} \geqslant \frac{\langle \nabla f(\mathbf{x}), \mathbf{a} - \mathbf{s} \rangle^2}{2LD^2}.
$$

Hence, $\hat{\mathbf{x}}$ satisfies the criteria of the descent step, and thus so does $\mathbf{x}'$ produced by the optimal step size $\gamma^*$.

If $\hat{\gamma} > \gamma_{\max}$, convexity of $f$ yields that either $f(\mathbf{x}') \leqslant f(\mathbf{x})$ or $\gamma^* = \gamma_{\max}$. In the first case, we have a valid descent step, and in the second case, we have a drop step.

Second, we consider the fully-corrective step in Algorithm 4. We compare the update to $\hat{\mathbf{x}}$. If $\hat{\gamma} \leqslant \gamma_{\max}$ and therefore $\hat{\mathbf{x}} \in \mathrm{conv}(S)$, the descent criterion is satisfied. We now analyze the case $\hat{\mathbf{x}} \notin \mathrm{conv}(S)$. If $f(\mathbf{x}') \leqslant f(\hat{\mathbf{x}})$ then we have a valid descent step. However, if $f(\mathbf{x}') > f(\hat{\mathbf{x}})$, we know that $\mathbf{x}' \notin \mathrm{relint}(\mathrm{conv}(S))$. Otherwise, there is a $\tilde{\mathbf{x}} \in \mathrm{conv}(S)$ on the line segment between $\hat{\mathbf{x}}$ and $\mathbf{x}'$ with $f(\tilde{\mathbf{x}}) < f(\mathbf{x}')$. This contradicts the optimality of $\mathbf{x}'$, thus we have $\mathbf{x}' \in \partial \mathrm{conv}(S)$. Consequently, there exists an atom $\mathbf{u} \in S$ which can be dropped without increasing the objective.

Finally, the simplex gradient descent step of BCG is a valid corrective step, as shown in Lemma 4.1 in Braun et al. [2019]. □

Finally, we prove the active set identification property of CFW in finite time under the assumption of strict complementarity.

**Theorem 4.** *Let $\mathcal{X}$ be a polytope and let $f$ be a convex, $L$-smooth function over $\mathcal{X}$. Assume that strict complementarity holds. Consider the sequence $\{\mathbf{x}_t\}_{t=0}^T$ generated by Algorithm 1. Then there exists an iteration $\tilde{T}$ such that*
$$\mathbf{x}_t \in \mathcal{F}^* \quad \text{for all } t \geqslant \tilde{T}.$$

*Proof.* Let $\varepsilon > 0$. For any $\mathbf{x}^* \in \mathcal{X}^*$, $\mathbf{v} \in V(\mathcal{X})$ and $\mathbf{x} \in \mathcal{X}$ with $\|\mathbf{x} - \mathbf{x}^*\| \leqslant \varepsilon$, $L$-smoothness of $f$ yields

$$
\begin{aligned}
\langle \nabla f(\mathbf{x}), \mathbf{v} - \mathbf{x} \rangle &= \langle \nabla f(\mathbf{x}^*), \mathbf{v} - \mathbf{x} \rangle + \langle \nabla f(\mathbf{x}) - \nabla f(\mathbf{x}^*), \mathbf{v} - \mathbf{x} \rangle \\
&= \langle \nabla f(\mathbf{x}^*), \mathbf{v} - \mathbf{x}^* \rangle + \langle \nabla f(\mathbf{x}^*), \mathbf{x}^* - \mathbf{x} \rangle + \langle \nabla f(\mathbf{x}) - \nabla f(\mathbf{x}^*), \mathbf{v} - \mathbf{x} \rangle \\
&\geqslant \langle \nabla f(\mathbf{x}^*), \mathbf{v} - \mathbf{x}^* \rangle - \|\nabla f(\mathbf{x}^*)\| \cdot \|\mathbf{x}^* - \mathbf{x}\| - \|\nabla f(\mathbf{x}) - \nabla f(\mathbf{x}^*)\| \cdot \|\mathbf{v} - \mathbf{x}\| \\
&\geqslant \langle \nabla f(\mathbf{x}^*), \mathbf{v} - \mathbf{x}^* \rangle - \|\nabla f(\mathbf{x}^*)\| \cdot \|\mathbf{x}^* - \mathbf{x}\| - L\|\mathbf{x} - \mathbf{x}^*\| \cdot \|\mathbf{v} - \mathbf{x}\| \\
&\geqslant \langle \nabla f(\mathbf{x}^*), \mathbf{v} - \mathbf{x}^* \rangle - \|\nabla f(\mathbf{x}^*)\| \cdot \varepsilon - L \cdot \varepsilon \cdot D \\
&= \langle \nabla f(\mathbf{x}^*), \mathbf{v} - \mathbf{x}^* \rangle - \varepsilon \cdot \underbrace{(\|\nabla f(\mathbf{x}^*)\| + LD)}_{\stackrel{\mathrm{def}}{=} c}
\end{aligned}
$$

Strict complementarity with constant $\rho > 0$ yields now

$$
\langle \nabla f(\mathbf{x}), \mathbf{v} - \mathbf{x} \rangle \geqslant \begin{cases} \rho - \varepsilon c & \mathbf{v} \notin \mathcal{F}^*, \\ -\varepsilon c & \mathbf{v} \in \mathcal{F}^*. \end{cases}
$$

For sufficiently small $\varepsilon$ we have that $\rho - \varepsilon c > \varepsilon c$, e.g., for $\varepsilon = \frac{\rho}{3c}$. By Theorem 2, $f(\mathbf{x}_T)$ converges to $f^*$. Since $f$ is continuous and $\mathcal{X}^*$ is compact, it follows that $\mathrm{dist}(\mathbf{x}_t, \mathcal{X}^*)$ converges to 0. Thus, there is a $T_1$ such that $\mathrm{dist}(\mathbf{x}_t, \mathcal{X}^*) \leqslant \varepsilon$ for all $t \geqslant T_1$. Therefore, there exists a $\mathbf{x}_t^* \in \mathcal{X}^*$ such that $\|\mathbf{x}_t - \mathbf{x}_t^*\| \leqslant \varepsilon$.

Let $\mathbf{v}_t, \mathbf{a}_t, \mathbf{s}_t$ be defined as in Algorithm 1. Next, we consider the cases where the current iterate $\mathbf{x}_t$ lies inside or outside the optimal face $\mathcal{F}^*$. If $\mathbf{x}_t \notin \mathcal{F}^*$, then there exists a $\mathbf{w} \in S_t \setminus \mathcal{F}^*$. Consequently, we have

$$\langle \nabla f(\mathbf{x}_t), \mathbf{a}_t - \mathbf{s}_t \rangle \geqslant \langle \nabla f(\mathbf{x}_t), \mathbf{w} - \mathbf{x}_t \rangle \geqslant \rho - \varepsilon c > \varepsilon c \geqslant \langle \nabla f(\mathbf{x}_t), \mathbf{x}_t - \mathbf{v}_t \rangle. \tag{25}$$

Therefore, CFW would choose to perform a corrective step. Let now $\tilde{\varepsilon} = \frac{(\rho - \varepsilon c)^2}{2LD^2}$. By Theorem 2 there exists a $T_2$ such that $f(\mathbf{x}_t) - f^* \leqslant \tilde{\varepsilon}$ for all $t \geqslant T_2$. Next, (25) yields

$$f(\mathbf{x}_t) - f^* \leqslant \frac{(\rho - \varepsilon c)^2}{2LD^2} \leqslant \frac{\langle \nabla f(\mathbf{x}_t), \mathbf{a}_t - \mathbf{s}_t \rangle^2}{2LD^2}$$

and thus the optimal primal progress is smaller than the required progress for a valid descent step in Algorithm 2. Since we cannot perform descent steps and since drop steps are non-increasing, CFW has to perform drop steps in every iteration $t \geqslant \max(T_1, T_2)$ where $\mathbf{x}_t \notin \mathcal{F}^*$.

Assume now that we always keep one vertex $\tilde{\mathbf{v}} \notin \mathcal{F}^*$ in the active set. Then in each iteration $\mathbf{x}_t$ stays outside $\mathcal{F}^*$, and we would keep dropping vertices until $\mathbf{x}_t = \tilde{\mathbf{v}}$. This contradicts the convergence

of Theorem 2. Therefore, there exists an iteration $\tilde{T} \geqslant \max(T_1, T_2)$ such that $S_{\tilde{T}} \subseteq \mathcal{F}^*$ and thus $\mathbf{x}_{\tilde{T}} \in \mathcal{F}^*$.

We turn to the case where $\mathbf{x}_t \in \mathcal{F}^*$. Strict complementarity and convexity yield

$$\langle \nabla f(\mathbf{x}_t), \mathbf{v}_t - \mathbf{x}_t \rangle \leqslant \langle \nabla f(\mathbf{x}_t), \mathbf{x}_t^* - \mathbf{x}_t \rangle \leqslant 0 < \rho - \varepsilon c \leqslant \langle \nabla f(\mathbf{x}_t), \mathbf{v} - \mathbf{x}_t \rangle$$

for any $\mathbf{v} \in V(\mathcal{X}) \setminus \mathcal{F}^*$. Therefore, we have $\mathbf{v}_t \in \mathcal{F}^*$ and thus $\mathbf{x}_{t+1} \in \mathcal{F}^*$ as neither FW step nor corrective steps move the iterate $\mathbf{x}_t$ out of $\mathcal{F}^*$. Finally, induction yields $\mathbf{x}_t \in \mathcal{F}^*$ for all $t \geqslant \tilde{T}$. □

# B   Lazified Corrective Frank-Wolfe

In this section, we will prove the convergence results of a more general version of Theorem 3 for the Lazified Corrective Frank-Wolfe method (LCFW) proposed in Algorithm 7.

---

**Algorithm 7** Lazified Corrective Frank-Wolfe (LCFW)

---

**Require:** convex smooth function $f$, starting point $\mathbf{x}_0 \in V(\mathcal{X})$, accuracy parameter $J \geqslant 1$.
1:  $\Phi_0 \leftarrow \max_{\mathbf{v} \in V(\mathcal{X})} \langle \nabla f(\mathbf{x}_0), \mathbf{x}_0 - \mathbf{v} \rangle / 2$
2:  $S_0 \leftarrow \{\mathbf{x}_0\}$
3:  **for** $t = 0$ to $T - 1$ **do**
4:      $\mathbf{a}_t \leftarrow \operatorname{argmax}_{\mathbf{v} \in S_t} \langle \nabla f(\mathbf{x}_t), \mathbf{v} \rangle$          ▷ away vertex
5:      $\mathbf{s}_t \leftarrow \operatorname{argmin}_{\mathbf{v} \in S_t} \langle \nabla f(\mathbf{x}_t), \mathbf{v} \rangle$          ▷ local FW
6:      **if** $\langle \nabla f(\mathbf{x}_t), \mathbf{a}_t - \mathbf{s}_t \rangle \geqslant \Phi_t$ **then**
7:          $\mathbf{x}_{t+1}, S_{t+1} \leftarrow \text{CorrectiveStep}(S_t, \mathbf{x}_t, \mathbf{a}_t, \mathbf{s}_t)$
8:          $\Phi_{t+1} \leftarrow \Phi_t$
9:      **else**
10:          $\mathbf{v}_t \leftarrow \operatorname{argmax}_{\mathbf{v} \in V(\mathcal{X})} \langle \nabla f(\mathbf{x}_t), \mathbf{x}_t - \mathbf{v} \rangle$          ▷ global FW
11:          **if** $\langle \nabla f(\mathbf{x}_t), \mathbf{x}_t - \mathbf{v}_t \rangle \geqslant \Phi_t / J$ **then**
12:              $\mathbf{d}_t \leftarrow \mathbf{x}_t - \mathbf{v}_t$
13:              $\gamma_t \leftarrow \operatorname{argmin}_{\gamma \in [0,1]} f(\mathbf{x}_t - \gamma \mathbf{d}_t)$
14:              $\mathbf{x}_{t+1} \leftarrow \mathbf{x}_t - \gamma_t \mathbf{d}_t$
15:              $S_{t+1} \leftarrow S_t \cup \{\mathbf{v}_t\}$
16:              $\Phi_{t+1} \leftarrow \Phi_t$          ▷ FW step
17:          **else**
18:              $\mathbf{x}_{t+1} \leftarrow \mathbf{x}_t$
19:              $S_{t+1} \leftarrow S_t$
20:              $\Phi_{t+1} \leftarrow \Phi_t / 2$          ▷ gap step
21:          **end if**
22:      **end if**
23:  **end for**

---

**Theorem 3.** *Let $\mathcal{X}$ be a convex feasible set with diameter $D$ and $f$ be a convex, $L$-smooth function over $\mathcal{X}$. Consider the sequence $\{\mathbf{x}_t\}_{t=0}^T \subset \mathcal{X}$ obtained by the Algorithm 7. Then we have*

$$f(\mathbf{x}_T) - f^* = \mathcal{O}\left(\frac{1}{T}\right). \tag{3}$$

*If additionally $f$ is $(c, \frac{1}{2})$-sharp and $\mathcal{X}$ is a polytope with pyramidal width $\delta$, then*

$$f(\mathbf{x}_T) - f^* = \mathcal{O}\left(\exp(-aT)\right), \tag{4}$$

*for some constant $a > 0$ independent of $T$.*

*Proof.* Similar to the proof of Theorem 2, we will prove an upper bound on the total number of necessary iterations $T$ for achieving $\varepsilon$ primal accuracy.

Let $N_{\text{FW}}, N_{\text{desc}}, N_{\text{drop}}, N_{\text{gap}}$ denote the number of FW, descent, drop, and gap steps, respectively. Let $t_1, \dots, t_{N_{\text{gap}}}$ be the iterations of the gap steps. We set $t_0 = -1$ for consistency. Then let $N_{\text{FW}}^i$ and $N_{\text{desc}}^i$ be the number of FW and descent steps in the $i$-th epoch, the iterations between $t_{i-1}$ and $t_i$. To

bound the total number of iterations $T$ necessary to achieve $\varepsilon$ primal accuracy, we first bound the number of gap steps $N_{\text{gap}}$.

Let $u = t_i + 1$ for some $i \in \{1, \ldots, N_{\text{gap}}\}$ be the iteration after a gap step. Note that $\mathbf{x}_u = \mathbf{x}_{t_i}$ and thus $\mathbf{v}_u = \mathbf{v}_{t_i}$. By convexity and the optimality of $\mathbf{v}_u$, we have

$$f(\mathbf{x}_u) - f^* \leqslant \langle \nabla f(\mathbf{x}_u), \mathbf{x}_u - \mathbf{x}^* \rangle \leqslant \langle \nabla f(\mathbf{x}_u), \mathbf{x}_u - \mathbf{v}_u \rangle \leqslant \frac{2\Phi_u}{J} \leqslant 2\Phi_u. \tag{26}$$

By definition of $\Phi_0$, this also holds for $u = 0$. Furthermore, the gap step always halves the value of $\Phi$, so we have $\Phi_u = \Phi_{t_i+1} = 2^{-i}\Phi_0$. By bounding the right-hand side of (26) by $\varepsilon$, we get

$$N_{\text{gap}} \leqslant \left\lceil \log_2 \frac{2\Phi_0}{\varepsilon} \right\rceil. \tag{27}$$

With $N_{\text{drop}} \leqslant N_{\text{FW}}$ we have that

$$\begin{aligned}
T &\leqslant N_{\text{FW}} + N_{\text{desc}} + N_{\text{gap}} + N_{\text{drop}} \\
&\leqslant 2N_{\text{FW}} + N_{\text{desc}} + N_{\text{gap}} \\
&\leqslant \sum_{i=1}^{N_{\text{gap}}} 2N_{\text{FW}}^i + N_{\text{desc}}^i + N_{\text{gap}}.
\end{aligned} \tag{28}$$

In the remainder of the proof, we find upper bounds for $2N_{\text{FW}}^i + N_{\text{desc}}^i$ by lower bounding the progress of FW and descent steps for both cases considered for $f$. First, we consider the case where $f$ is convex and smooth.

Let $t \in (t_i, t_{i+1})$, so $\Phi_t = \Phi_u$. If we perform a FW step, we have that $\langle \nabla f(\mathbf{x}_t), \mathbf{x}_t - \mathbf{v}_t \rangle \geqslant \Phi_t/J = \Phi_u/J$. Here we can use results from the proof of Theorem 2, i.e., (19) and (21), which yield

$$\begin{aligned}
f(\mathbf{x}_t) - f(\mathbf{x}_{t+1}) &\geqslant \min \left\{ \frac{\langle \nabla f(\mathbf{x}_t), \mathbf{x}_t - \mathbf{v}_t \rangle^2}{2LD^2}, \frac{1}{2} \langle \nabla f(\mathbf{x}_t), \mathbf{x}_t - \mathbf{v}_t \rangle \right\} \\
&\geqslant \frac{\Phi_u}{2J} \min \left\{ \frac{\Phi_u}{LD^2 J}, 1 \right\}.
\end{aligned} \tag{29}$$

Next, we consider the case of a descent step, i.e., $\langle \nabla f(\mathbf{x}_t), \mathbf{a}_t - \mathbf{s}_t \rangle \geqslant \Phi_t$. Together with the definition of the descent step, we have

$$f(\mathbf{x}_t) - f(\mathbf{x}_{t+1}) \geqslant \frac{\langle \nabla f(\mathbf{x}_t), \mathbf{a}_t - \mathbf{s}_t \rangle^2}{2LD^2} \geqslant \frac{\Phi_u^2}{2LD^2}. \tag{30}$$

We now use (26) as an upper bound of the primal progress in the $i$-th epoch, and we use (29) and (30) to lower bound the primal progress. Let $I = \lceil \log_2 \frac{\Phi_0}{LD^2 J} \rceil$. For $i < I$ we have that $\Phi_0 \geqslant 2^i LD^2 J$ and thus $\Phi_u \geqslant LD^2 J$. Then we get with (26), (29) and (30),

$$\begin{aligned}
2\Phi_u &\geqslant f(\mathbf{x}_u) - f^* \\
&\geqslant f(\mathbf{x}_u) - f(\mathbf{x}_{t_{i+1}}) \\
&\geqslant N_{\text{FW}}^i \frac{\Phi_u}{2J} + N_{\text{desc}}^i \frac{\Phi_u^2}{2LD^2} \\
&\geqslant 2N_{\text{FW}}^i \frac{\Phi_u}{4J} + N_{\text{desc}}^i \frac{\Phi_u J}{2} \\
&= \Phi_u \left( 2N_{\text{FW}}^i \frac{1}{4J} + N_{\text{desc}}^i \frac{J}{2} \right),
\end{aligned}$$

and thus

$$2N_{\text{FW}}^i + N_{\text{desc}}^i \leqslant \max \left\{ 8J, \frac{4}{J} \right\}. \tag{31}$$

For $i \geqslant I$ we have that $\Phi_u \leqslant LD^2 J$ and thus with (26), (29) and (30),

$$
\begin{aligned}
2\Phi_u &\geqslant f(\mathbf{x}_u) - f^* \\
&\geqslant f(\mathbf{x}_u) - f(\mathbf{x}_{t_{i+1}}) \\
&\geqslant N_{\text{FW}}^i \frac{\Phi_u^2}{2J^2 LD^2} + N_{\text{desc}}^i \frac{\Phi_u^2}{2LD^2} \\
&= \Phi_u^2 \left( 2N_{\text{FW}}^i \frac{1}{4LD^2 J^2} + N_{\text{desc}}^i \frac{1}{2LD^2} \right)
\end{aligned}
$$

Thus we have

$$
2N_{\text{FW}}^i + N_{\text{desc}}^i \leqslant \frac{1}{\Phi_u} \max\left\{ 8LD^2 J^2, 4LD^2 \right\} = \frac{2^i}{\Phi_0} \max\left\{ 8LD^2 J^2, 4LD^2 \right\}. \tag{32}
$$

We can now bound the number of total iterations using (28) and (27),

$$
\begin{aligned}
T &\leqslant \sum_{i=1}^{N_{\text{gap}}} 2N_{\text{FW}}^i + N_{\text{desc}}^i + N_{\text{gap}} \\
&\leqslant I \cdot \max\left\{ 8J, \frac{4}{J} \right\} + \sum_{i=I}^{N_{\text{gap}}} \frac{2^i}{\Phi_0} \max\left\{ 8LD^2 J^2, 4LD^2 \right\} + N_{\text{gap}} \\
&\leqslant I \cdot \max\left\{ 8J, \frac{4}{J} \right\} + (2^{N_{\text{gap}}+1} - 1)\frac{1}{\Phi_0} \max\left\{ 8LD^2 J^2, 4LD^2 \right\} + N_{\text{gap}} \\
&\leqslant I \cdot \max\left\{ 8J, \frac{4}{J} \right\} + (2^{\log_2 \frac{2\Phi_0}{\varepsilon}+2} - 1)\frac{1}{\Phi_0} \max\left\{ 8LD^2 J^2, 4LD^2 \right\} + \left\lceil \log_2 \frac{2\Phi_0}{\varepsilon} \right\rceil \\
&\leqslant I \cdot \max\left\{ 8J, \frac{4}{J} \right\} + \frac{4}{\varepsilon} \max\left\{ 8LD^2 J^2, 4LD^2 \right\} + \left\lceil \log_2 \frac{2\Phi_0}{\varepsilon} \right\rceil.
\end{aligned}
$$

Finally, (26) yields

$$
f(\mathbf{x}_T) - f^* \leqslant \frac{\Phi_T}{2} \leqslant \frac{\varepsilon}{2} = \mathcal{O}\left( \frac{1}{T} \right).
$$

Consider now the case where $f$ is $(c, \theta)$-sharp and $\mathcal{X}$ is a polytope. Let again $u = t_i + 1$ for some $i \in \{1, \ldots, N_{\text{gap}}\}$ be the iteration after a gap step. Then we have

$$
\langle \nabla f(\mathbf{x}_u), \mathbf{a}_u - \mathbf{v}_u \rangle \leqslant \langle \nabla f(\mathbf{x}_u), \mathbf{a}_u - \mathbf{s}_u \rangle + \langle \nabla f(\mathbf{x}_u), \mathbf{x}_u - \mathbf{v}_u \rangle < \Phi_u + \Phi_u/J \leqslant 2\Phi_u.
$$

With Lemma 11 we have that

$$
f(\mathbf{x}_u) - f^* \leqslant \left( \frac{c \langle \nabla f(\mathbf{x}_u), \mathbf{a}_u - \mathbf{v}_u \rangle}{\delta} \right)^{\frac{1}{1-\theta}} \leqslant \left( \frac{2c\Phi_u}{\delta} \right)^{\frac{1}{1-\theta}}. \tag{33}
$$

With this new upper bound on the primal gap, we can improve the bound on $2N_{\text{FW}}^i + N_{\text{desc}}^i$ for $i \geqslant I$. Using (33), (29) and (30), we get

$$
\begin{aligned}
\left( \frac{2c\Phi_u}{\delta} \right)^{\frac{1}{1-\theta}} &\geqslant f(\mathbf{x}_u) - f^* \\
&\geqslant f(\mathbf{x}_u) - f(\mathbf{x}_{t_{i+1}}) \\
&\geqslant N_{\text{FW}}^i \frac{\Phi_u^2}{2J^2 LD^2} + N_{\text{desc}}^i \frac{\Phi_u^2}{2LD^2} \\
&\geqslant \frac{\Phi_u^2}{LD^2} \left( \frac{2N_{\text{FW}}^i}{4J^2} + \frac{N_{\text{desc}}^i}{2} \right).
\end{aligned}
$$

Consequently, we have

$$
\begin{aligned}
2N_{\text{FW}}^i + N_{\text{desc}}^i &\leqslant \left( \frac{2c}{\delta} \right)^{\frac{1}{1-\theta}} LD^2 \max\left\{ 4J^2, 2 \right\} \Phi_u^{\frac{1}{1-\theta}-2} \\
&\leqslant \left( \frac{2c}{\delta} \right)^{\frac{1}{1-\theta}} LD^2 \max\left\{ 4J^2, 2 \right\} \left( \frac{\Phi_0}{2^i} \right)^{\frac{1}{1-\theta}-2}. \tag{34}
\end{aligned}
$$

For $\theta = \frac{1}{2}$ this yields

$$\sum_{i=I}^{N_{\text{gap}}} 2N_{\text{FW}}^i + N_{\text{desc}}^i \leqslant N_{\text{gap}} \frac{4c^2}{\delta^2} LD^2 \max\left\{4J^2, 2\right\}. \tag{35}$$

The bounds for the case $i < I$ in (31) stay unchanged. Together with (27) and (28) this yields,

$$
\begin{aligned}
T &\leqslant \sum_{i=1}^{N_{\text{gap}}} 2N_{\text{FW}}^i + N_{\text{desc}}^i + N_{\text{gap}} \\
&\leqslant I \cdot \max\left\{8J, \frac{4}{J}\right\} + (N_{\text{gap}} - I) \cdot \frac{4c^2 LD^2}{\delta^2} \max\left\{4J^2, 2\right\} + N_{\text{gap}} \\
&\leqslant C_1 N_{\text{gap}} \leqslant C_1 \left\lceil \log_2 \frac{2\Phi_0}{\varepsilon} \right\rceil,
\end{aligned}
$$

where $C_1 = 1 + \max\left\{\max\left\{8J, \frac{4}{J}\right\}, \frac{4c^2 LD^2}{\delta^2} \max\left\{4J^2, 2\right\}\right\}$. This is equivalent to

$$\varepsilon \leqslant 2\Phi_0 e^{\frac{1}{C_1}} \exp\left(-\frac{1}{C_1}T\right) = \mathcal{O}\left(\exp\left(-\frac{1}{C_1}T\right)\right).$$

Finally, with (33) we get

$$f(\mathbf{x}_T) - f^* \leqslant \left(\frac{c\langle \nabla f(\mathbf{x}_T), \mathbf{a}_T - \mathbf{v}_T \rangle}{\delta}\right)^2 \leqslant \left(\frac{2c\Phi_T}{\delta}\right)^2 = \mathcal{O}(\exp(-aT)),$$

with $a = \frac{2}{C_1}$.

$\square$

## C  Quadratic Corrections

### C.1  QC-MNP for convex objectives

In this section, we derive a variant of the QC-MNP method tailored for non-strongly convex objectives, i.e., when the affine minimizer might not be unique. This is not only more efficient in practice but also relevant for Theorem 8 to guarantee finite convergence.

For convenience, we restate the relevant linear system,

$$\langle \mathbf{AV}\boldsymbol{\lambda} + \mathbf{b}, \mathbf{v} - \mathbf{w} \rangle = 0 \quad \forall\, \mathbf{v} \in S \setminus \{\mathbf{w}\}, \quad \sum_{\mathbf{v} \in S} \boldsymbol{\lambda}_{\mathbf{v}} = 1. \tag{36}$$

For a given solution $\lambda$, the ratio test in Algorithm 6 maximizes $\tau$ such that the new weights $\tau\boldsymbol{\lambda} + (1 - \tau)\boldsymbol{\lambda}(\mathbf{x})$ are non-negative. If the solution is not unique, we can choose $\boldsymbol{\lambda}$ that allows for the largest $\tau$. This will ensure that we pick an affine minimizer inside the convex hull if such exists.

Adding the non-negativity constraint and the objective to (36) yields the following non-linear program:

$$
\begin{aligned}
\max_{\boldsymbol{\lambda} \in \mathbb{R}^m, \tau \in [0,1]} \quad & \tau \\
\text{s.t.} \quad & \langle \mathbf{AV}\boldsymbol{\lambda} + \mathbf{b}, \mathbf{v} - \mathbf{w} \rangle = 0 \quad \forall\, \mathbf{v} \in S \setminus \{\mathbf{w}\}, \\
& \mathbf{1}^\top \boldsymbol{\lambda} = 1, \\
& \tau\boldsymbol{\lambda} + (1 - \tau)\boldsymbol{\lambda}(\mathbf{x}) \geqslant 0.
\end{aligned}
$$

As $\mathbf{x}$ lies in the relative interior of $\text{conv}(S)$, there always exists a feasible $\tau > 0$. To obtain a linear inequality constraint, we divide the inequality constraint by $\tau$ and set $\beta = \frac{1 - \tau}{\tau}$,

$$\boldsymbol{\lambda} + \beta\boldsymbol{\lambda}(\mathbf{x}) \geqslant 0.$$

Since maximizing $\tau$ on $(0, 1]$ is equivalent to minimizing $\beta$ on $[0, \infty)$, we obtain the following linear program:

$$\min_{\boldsymbol{\lambda} \in \mathbb{R}^m, \beta \geqslant 0} \quad \beta \tag{37}$$
$$\text{s.t.} \quad \langle \mathbf{AV}\boldsymbol{\lambda} + \mathbf{b}, \mathbf{v} - \mathbf{w} \rangle = 0 \quad \forall\, \mathbf{v} \in S \setminus \{\mathbf{w}\},$$
$$\mathbf{1}^\top \boldsymbol{\lambda} = 1,$$
$$\boldsymbol{\lambda} + \beta \boldsymbol{\lambda}(\mathbf{x}) \geqslant 0.$$

Algorithm 8 presents the alternative implementation of the QC-MNP-correction step, relying on the newly derived LP. We use again Algorithm 3 as a fallback step if there exists no affine minimizer.

---

**Algorithm 8** Quadratic Correction MNP

---

**Require:** $S \subset \mathcal{X}, \mathbf{x} \in \mathcal{X}$
  $\beta, \tilde{\boldsymbol{\lambda}} \leftarrow$ Solve (37)
  **if** feasible **then**
    **if** $\beta = 0$ **then**                                                     $\triangleright$ FCFW step
      $\mathbf{x}' \leftarrow \sum_{\mathbf{v} \in S} \tilde{\boldsymbol{\lambda}}_{\mathbf{v}} \mathbf{v}$
      $S' \leftarrow S$
    **else**                                                                 $\triangleright$ MNP step
      $\tau \leftarrow \frac{1}{\beta+1}$
      $\mathbf{x}' \leftarrow \sum_{\mathbf{v} \in S} (\tau \tilde{\boldsymbol{\lambda}}_{\mathbf{v}} + (1-\tau)\boldsymbol{\lambda}_{\mathbf{v}}(\mathbf{x}))\mathbf{v}$
      $S' \leftarrow \{\mathbf{v} \in S \mid \boldsymbol{\lambda}_{\mathbf{v}}(\mathbf{x}') > 0\}$
    **end if**
  **else**                                                                    $\triangleright$ Fallback step
    $S', \mathbf{x}' \leftarrow \text{LPS}(S, \mathbf{x}, \mathbf{a}, \mathbf{s})$
  **end if**

---

We can now prove that Algorithm 8 always chooses an affine minimizer in $\text{conv}(S)$ if such exists.

**Lemma 13.** *Let $S \subset \mathcal{X}$. If $\emptyset \neq \arg\min_{\mathbf{x} \in \text{conv}(S)} f(\mathbf{x}) \subseteq \arg\min_{\mathbf{x} \in \text{aff}(S)} f(\mathbf{x})$, then Algorithm 8 performs an FCFW step.*

*Proof.* Since the linear system in (36) is a sufficient optimality condition for the affine minimizer, we know by assumption that it has at least one solution. Furthermore, there exists a solution which additionally satisfies $\boldsymbol{\lambda} \geqslant 0$. Therefore, $\beta = 0$ and $\boldsymbol{\lambda}$ are optimal solutions to (37). Consequently, Algorithm 8 performs an FCFW step. $\qquad\square$

### C.2 Proofs for Quadratic Corrections

In this section, we prove the results for the quadratic corrections given in Section 2.2.

**Proposition 5.** *Equation* (6) *is feasible if and only if* $\mathbf{b} \perp \text{span}(S) \cap \ker(\mathbf{A})$. *In particular, it is feasible if* $\mathbf{A}$ *is positive definite.*

*Proof.* Instead of (6), we consider the equivalent system (8). Since the matrix $\mathbf{W}^\top \mathbf{A} \mathbf{W}$ is symmetric, the system is feasible if and only if

$$\mathbf{W}^\top (\mathbf{A}\mathbf{w} + \mathbf{b}) \in \text{im}(\mathbf{W}^\top \mathbf{A} \mathbf{W}) = \ker(\mathbf{W}^\top \mathbf{A} \mathbf{W})^\perp.$$

One can easily see that $\ker(\mathbf{A}\mathbf{W}) \subseteq \ker(\mathbf{W}^\top \mathbf{A} \mathbf{W})$. Let $\mathbf{v} \in \ker(\mathbf{W}^\top \mathbf{A} \mathbf{W})$, then we have

$$\mathbf{v}^\top \mathbf{W}^\top \mathbf{A} \mathbf{W} \mathbf{v} = (\mathbf{W}\mathbf{v})^\top \mathbf{A} (\mathbf{W}\mathbf{v}) = 0,$$

and thus $\mathbf{W}\mathbf{v} \in \ker(\mathbf{A})$ since $\mathbf{A}$ is positive semi-definite. Consequently, (8) is feasible if and only if $\mathbf{W}^\top (\mathbf{A}\mathbf{w} + \mathbf{b}) \perp \mathbf{v}$ for all $\mathbf{v} \in \ker(\mathbf{A}\mathbf{W})$. The condition simplifies as $\mathbf{A}$ is symmetric,

$$\mathbf{v}^\top \mathbf{W}^\top (\mathbf{A}\mathbf{w} + \mathbf{b}) = \mathbf{v}^\top \mathbf{W}^\top \mathbf{A}^\top \mathbf{w} + \mathbf{v}^\top \mathbf{W}^\top \mathbf{b} = (\mathbf{W}\mathbf{v})^\top \mathbf{b} = 0.$$

Therefore, (8) is feasible if and only if $\mathbf{b} \perp \mathbf{z}$ for all $\mathbf{z} \in \text{im}(\mathbf{W}) \cap \ker(\mathbf{A})$. This is equivalent to $\mathbf{b} \perp \text{span}(S) \cap \ker(\mathbf{A})$. $\qquad\square$

**Proposition 6.** *QC-LP and QC-MNP satisfy the criteria of corrective steps in Algorithm 2.*

*Proof.* For the QC-LP, if the LP in (7) is feasible, the solution coincides with the FCFW step. If the LP is infeasible, we perform a local pairwise step, which satisfies the criteria of a corrective step, as shown in Proposition 1.

For QC-MNP, if a solution of (6) exists and lies already in the $|S|$-simplex, the step is equivalent to the FCFW step. Otherwise, we perform a drop step, because at least one new weight is negative. The primal value does not increase, as $f$ is convex and one moves towards the affine minimizer. If there exists no affine minimizer, we perform a local pairwise step. $\square$

**Proposition 7.** *Let $\mathcal{X}$ be a polytope and $\mathcal{F}^*$ be the minimal face of $\mathcal{X}$ containing the optimal points $\mathcal{X}^*$. If $S \subset \mathcal{F}^*$, then $\operatorname{argmin}_{x\in\operatorname{aff}(S)} f(x) \neq \emptyset$. If additionally there exists $\mathbf{x}^* \in \mathcal{X}^*$ with $\mathbf{x}^* \in \operatorname{conv}(S)$, then $\mathbf{x}^* \in \operatorname{argmin}_{x\in\operatorname{aff}(S)} f(x)$.*

*Proof.* For contradiction, assume the problem $\min_{\mathbf{x}\in\operatorname{aff}(S)} f(\mathbf{x})$ would be unbounded. Then there exists an $\tilde{\mathbf{x}} \in \operatorname{aff}(S)$ such that $f(\tilde{\mathbf{x}}) < f^*$. If the optimal face $\mathcal{F}^*$ is a singleton, $S$ would be also a singleton yielding $\tilde{\mathbf{x}} \in \operatorname{aff}(S) \subset \mathcal{X}$ which contradicts the minimality of $f^*$. If $\mathcal{F}^*$ is at least one dimensional, there exists an $\mathbf{x}^* \in \operatorname{relint}(\mathcal{F}^*)$ due to minimality and convexity of $\mathcal{F}^*$. Consequently, there exists a $\lambda \in (0,1)$ with $\mathbf{y} = \lambda\tilde{\mathbf{x}} + (1-\lambda)\mathbf{x}^* \in \mathcal{F}^* \subset \mathcal{X}$. Convexity yields then,

$$f(\mathbf{y}) = f(\lambda\tilde{\mathbf{x}} + (1-\lambda)\mathbf{x}^*) \leqslant \lambda f(\tilde{\mathbf{x}}) + (1-\lambda)f(\mathbf{x}^*) < f^*,$$

which contradicts the minimality of $f^*$. Consequently, the problem is bounded below by $f^*$. If there exists an $\mathbf{x}^* \in \mathcal{X}^* \cap \operatorname{conv}(S)$, then it is also a solution to the affine problem since $\mathbf{x}^* \in \operatorname{aff}(S)$. $\square$

**Theorem 8.** *Let $\mathcal{X}$ be a polytope and let $f$ be a convex, quadratic function over $\mathcal{X}$, i.e., $f(\mathbf{x}) = \frac{1}{2}\langle \mathbf{x}, \mathbf{A}\mathbf{x}\rangle + \langle \mathbf{b}, \mathbf{x}\rangle + c$ where $\mathbf{A}$ is a symmetric positive semidefinite matrix. Assume that strict complementarity holds. Consider the sequence $\{\mathbf{x}_t\}_{t=0}^T$ generated by Algorithm 1 with Algorithm 4, Algorithm 5, or Algorithm 8. Then, there exists an iteration $\tilde{T}$ such that $\mathbf{x}_{\tilde{T}} \in \mathcal{X}^*$.*

*Proof.* By Theorem 4 we know that there exists a $T_1$ such that $S_t \subset \mathcal{F}^*$ holds for all $t \geqslant T_1$. Let

$$\varepsilon = \min_{\substack{S\subset V(\mathcal{F}^*) \\ \operatorname{conv}(S)\cap\mathcal{X}^*=\emptyset}} \operatorname{dist}(\operatorname{conv}(S), \mathcal{X}^*).$$

By Theorem 2 there exists a $T_2$ such that $\operatorname{dist}(\mathbf{x}_t, \mathcal{X}^*) \leqslant \varepsilon$ and thus $\operatorname{dist}(\operatorname{conv}(S_t), \mathcal{X}^*) < \varepsilon$ holds for all $t \geqslant T_2$. Consequently, we have $\operatorname{conv}(S_t) \cap \mathcal{X}^* \neq \emptyset$ for all $t \geqslant T_2$. In total, we have $S_t \subset \mathcal{F}^*$ and $\operatorname{conv}(S_t) \cap \mathcal{X}^* \neq \emptyset$ for all $t \geqslant \tilde{T} = \max(T_1, T_2)$.

By Proposition 7, we know there exists an $x^* \in \operatorname{argmin}_{\mathbf{x}\in\operatorname{aff}(S_t)} f(\mathbf{x}) \cap \mathcal{X}^* \subset \operatorname{conv}(S_t)$ for all $t \geqslant \tilde{T}$. If CFW performs now a fully-corrective or a QC-LP step, we converge directly to an $x^* \in \mathcal{X}^*$. For QC-MNP, we need to use the implementation in Algorithm 8. Otherwise, we might pick an affine minimizer not in $\mathcal{X}$, see Lemma 13.

Finally, it remains to show that CFW performs a corrective step after at most some finite iterations after $\tilde{T}$. By Theorem 4 we know that $\mathbf{v}_t \in \mathcal{F}^*$ for all $t \geqslant \tilde{T}$. If $\mathbf{v}_t \in S_t$, then $\langle \nabla f(\mathbf{x}_t), \mathbf{v}_t\rangle = \langle \nabla f(\mathbf{x}_t), \mathbf{s}_t\rangle$ and thus

$$\langle \nabla f(\mathbf{x}_t), \mathbf{x}_t - \mathbf{v}_t\rangle \leqslant \langle \nabla f(\mathbf{x}_t), \mathbf{a}_t - \mathbf{s}_t\rangle.$$

Therefore, CFW performs an FW step if $\mathbf{v}_{t_-} \notin S_t$. This yields that CFW could perform at most $|V(\mathcal{F}^*)| - |S_{\tilde{T}}|$ consecutive FW steps after $\tilde{T}$, before performing a corrective step. $\square$

# D   Split Conditional Gradient

In this section, we consider the Split Conditional Gradient (SCG) method of Woodstock and Pokutta [2025]. The algorithm is presented in Algorithm 9.

---
**Algorithm 9** Split Conditional Gradient (SCG)
---
**Require:** Convex, smooth function $f$, weights $\{w_i\}_{i \in I} \subset (0, 1)$ with $\sum_{i \in I} w_i = 1$, starting point
$\mathbf{x}_0 \in \bigtimes_{i \in I} \mathcal{X}_i$
1: $\bar{\mathbf{x}}_0 \leftarrow \sum_{i \in I} w_i \mathbf{x}_0$
2: **for** $t = 0, 1, \dots$ **do**
3:     Choose penalty parameter $\lambda_t > 0$
4:     Choose step size $\gamma_t \in (0, 1)$
5:     $\mathbf{g}_t \leftarrow \nabla f(\bar{\mathbf{x}}_t)$
6:     **for** $i \in I$ **do**
7:         $\mathbf{v}_t^i \leftarrow \operatorname{argmin}_{\mathbf{v} \in \mathcal{H}} \langle \mathbf{g}_t + \lambda_t(\mathbf{x}_t^i - \bar{\mathbf{x}}_t), \mathbf{v} \rangle$
8:         $\mathbf{x}_t^i \leftarrow \mathbf{x}_t^i + \gamma_t(\mathbf{v}_t^i - \mathbf{x}_t^i)$
9:     **end for**
10:    $\bar{\mathbf{x}}_{t+1} \leftarrow \sum_{i \in I} w_i \mathbf{x}_t^i$
11: **end for**
---

We prove the convergence rate of the SCG method stated in Theorem 9 in a more general setting, i.e., for arbitrary Hilbert spaces and any finite intersection. Let $\mathcal{H}$ be a real Hilbert space, let $\boldsymbol{\mathcal{H}} = \mathcal{H}^m$ be the product space of $\mathcal{H}$. We denote the components of $\mathbf{x} \in \boldsymbol{\mathcal{H}}$ as $\mathbf{x} = (\mathbf{x}^1, \dots, \mathbf{x}^m)$. Let $\boldsymbol{D} = \{\mathbf{x} \in \boldsymbol{\mathcal{H}} \mid \mathbf{x}^1 = \mathbf{x}^2 = \dots = \mathbf{x}^m\}$ denote the diagonal space of $\boldsymbol{\mathcal{H}}$. Furthermore, let $I = \{1, \dots, m\}$ and let $\{w_i\}_{i \in I}$ be a selection of weights such that $w_i > 0$ for all $i \in I$ and $\sum_{i \in I} w_i = 1$. The averaging operator is defined as $A \colon \boldsymbol{\mathcal{H}} \to \mathcal{H} \colon \mathbf{x} \mapsto \sum_{i \in I} w_i \mathbf{x}^i$.

**Proposition 14.** *[Woodstock and Pokutta, 2025, Proposition 2.13] Let $f : \mathcal{H} \to \mathbb{R}$, let $(\mathcal{X}_i)_{i \in I}$ be a finite selection of non-empty, compact, and convex sets of $\mathcal{H}$. For every $\lambda \geqslant 0$, set $F_\lambda(\mathbf{x}) = f(A\mathbf{x}) + \frac{\lambda}{2} \operatorname{dist}_{\boldsymbol{D}}^2(\mathbf{x})$. Suppose that $(\lambda_n)_{n \in \mathbb{N}} \nearrow \infty$. Then*

$$\lim_{t \to \infty} \left( \inf_{\mathbf{x} \in \bigtimes_{i \in I} \mathcal{X}_i} F_{\lambda_t}(\mathbf{x}) \right) = \inf_{\mathbf{x} \in \bigtimes_{i \in I} \mathcal{X}_i} \left( \lim_{t \to \infty} F_{\lambda_t}(\mathbf{x}) \right) = \inf_{\mathbf{x} \in \bigcap_{i \in I} \mathcal{X}_i} f(\mathbf{x}).$$

**Theorem 15.** *Let $f \colon \mathcal{H} \to \mathbb{R}$ be convex and $L$-smooth and let $(\mathcal{X}_i)_{i \in I}$ be a finite selection of non-empty, compact, and convex sets of $\mathcal{H}$ with diameters $\{R_i\}_{i \in I}$ and $\bigcap_{i \in I} \mathcal{X}_i \neq \emptyset$. For $\lambda \geqslant 0$, set $F_\lambda(\mathbf{x}) = f(A\mathbf{x}) + \frac{\lambda}{2} \operatorname{dist}_{\boldsymbol{D}}^2(\mathbf{x})$, $\mathbf{x}_t^* \in \operatorname{argmin}_{\mathbf{x} \in \boldsymbol{\mathcal{H}}} F_{\lambda_t}(\mathbf{x})$ and $H_t = F_{\lambda_t}(\mathbf{x}_t) - F_{\lambda_t}(\mathbf{x}_t^*)$. For $\lambda_t = \ln(t + 2)$ and the step size $\gamma_t = \frac{2}{\sqrt{t+2} \ln(t+2)}$ for all $t \geqslant 0$, the iterates of SCG satisfy*

$$0 \leqslant H_t \leqslant \frac{2R^2(L+1) + \sqrt{2} c_f}{\sqrt{t+2}} \tag{38}$$

*for all $t \geqslant 0$, where $c_f \stackrel{\text{def}}{=} \max_{x, y \in \bigtimes_{i \in I} \mathcal{X}_i} f(Ax) - f(Ay) < \infty$ and $R^2 \stackrel{\text{def}}{=} \sum_{i \in I} w_i R_i^2$.*

*Furthermore, we get that $\lim_{t \to \infty} F_{\lambda_t}(\mathbf{x}_t) = \inf_{\mathbf{x} \in \bigtimes_{i \in I} \mathcal{X}_i} f(\mathbf{x})$ and $\lim_{t \to \infty} \operatorname{dist}_{\boldsymbol{D}}(\mathbf{x}_t) = 0$. In particular, any accumulation point $\tilde{\mathbf{x}}$ of the sequence $\{\mathbf{x}_t\}_{t \geqslant 0}$ lies in $\bigcap_{i \in I} \mathcal{X}_i$ and satisfies $f(A\tilde{\mathbf{x}}) = \inf_{\mathbf{x} \in \bigtimes_{i \in I} \mathcal{X}_i} f(\mathbf{x})$.*

*Proof.* This proof is inspired by the proof of Woodstock and Pokutta [2025]. The key difference is to analyze the alternative auxiliary objective,

$$\widetilde{F}_\lambda(\mathbf{x}) = \frac{F_\lambda(\mathbf{x})}{\lambda} = \frac{f(A\mathbf{x})}{\lambda} + \frac{1}{2} \operatorname{dist}_{\mathbf{D}}^2(\mathbf{x}).$$

We will see that one can find smaller upper bounds on the primal gap $\widetilde{H}_t = \widetilde{F}_{\lambda_t}(\mathbf{x}_t) - \widetilde{F}_{\lambda_t}(\mathbf{x}_t^*)$. However, even for $\lambda_t \widetilde{H}_t = H_t$, we achieve faster rates of convergence. Note that

$$\mathbf{x}_t^* \in \operatorname*{argmin}_{\mathbf{x} \in \bigtimes_{i \in I} \mathcal{X}_i} F_{\lambda_t}(\mathbf{x}) = \operatorname*{argmin}_{\mathbf{x} \in \bigtimes_{i \in I} \mathcal{X}_i} \widetilde{F}_{\lambda_t}(\mathbf{x}),$$

since $F_{\lambda_t}$ and $\widetilde{F}_{\lambda_t}$ only differ in scaling. Furthermore, the function $\widetilde{F}_{\lambda_t}$ is $\left( \frac{L}{\lambda_t} + 1 \right)$-smooth.

Analogous to Lemma 3.2 in Woodstock and Pokutta [2025], we first prove a bound on the primal gap $\widetilde{H}_t$. Using smoothness, the FW update rule, the optimality of the FW vertex $\mathbf{v}_t$, and convexity, we get

$$
\begin{aligned}
& \widetilde{F}_{\lambda_t}\left(\mathbf{x}_{t+1}\right) - \widetilde{F}_{\lambda_t}\left(\mathbf{x}_t\right) \\
& \leqslant \left\langle \nabla\widetilde{F}_{\lambda_t}\left(\mathbf{x}_t\right), \mathbf{x}_{t+1} - \mathbf{x}_t \right\rangle + \frac{\frac{L}{\lambda_t}+1}{2}\left\| \mathbf{x}_{t+1} - \mathbf{x}_t \right\|^2 \\
& = \gamma_t \left\langle \nabla\widetilde{F}_{\lambda_t}(\mathbf{x}_t), \mathbf{v}_t - \mathbf{x}_t \right\rangle + \frac{\frac{L}{\lambda_t}+1}{2}\gamma_t^2\left\| \mathbf{v}_t - \mathbf{x}_t \right\|^2 \\
& \leqslant \gamma_t \left\langle \nabla\widetilde{F}_{\lambda_t}(\mathbf{x}_t), \mathbf{x}_t^* - \mathbf{x}_t \right\rangle + \frac{\frac{L}{\lambda_t}+1}{2}\gamma_t^2 R^2 \\
& \leqslant \gamma_t \left( \widetilde{F}\left(\mathbf{x}_t^*\right) - \widetilde{F}_{\lambda_t}(\mathbf{x}_t) \right) + \gamma_t^2 R^2 \frac{\frac{L}{\lambda_t}+1}{2}
\end{aligned}
$$

where $R^2$ is an upper bound on $\|\mathbf{x} - \mathbf{y}\|^2$ for all $\mathbf{x}, \mathbf{y} \in \bigcap_{i \in I}\mathcal{X}_i$, see Woodstock and Pokutta [2025, Lemma 3.1].

Adding $\widetilde{F}_{\lambda_t}(\mathbf{x}_t) - \widetilde{F}_{\lambda_t}(\mathbf{x}_t^*)$ to both sides leads to

$$
\widetilde{F}_{\lambda_t}(\mathbf{x}_{t+1}) - \widetilde{F}_{\lambda_t}(\mathbf{x}_t^*) \leqslant (1-\gamma_t)\underbrace{(\widetilde{F}_{\lambda_t}(\mathbf{x}_t) - \widetilde{F}_{\lambda_t}(\mathbf{x}_t^*))}_{\widetilde{H}_t} + \gamma_t^2 R^2 \frac{\frac{L}{\lambda_t}+1}{2}.
$$

Together with the definition of $\widetilde{F}_\lambda$ and the optimality of $\mathbf{x}_t^*$ yields

$$
\begin{aligned}
\widetilde{H}_{t+1} & = \widetilde{F}_{\lambda_{t+1}}\left(\mathbf{x}_{t+1}\right) - \widetilde{F}_{\lambda_{t+1}}\left(\mathbf{x}_{t+1}^*\right) \\
& = \widetilde{F}_{\lambda_t}(\mathbf{x}_{t+1}) - \widetilde{F}_{\lambda_t}(\mathbf{x}_{t+1}^*) + \left( \frac{1}{\lambda_{t+1}} - \frac{1}{\lambda_t} \right)\left( f(A\mathbf{x}_{t+1}) - f(A\mathbf{x}_{t+1}^*) \right) \\
& = \widetilde{F}_{\lambda_t}(\mathbf{x}_{t+1}) - \widetilde{F}_{\lambda_t}(\mathbf{x}_{t+1}^*) + \left( \frac{1}{\lambda_t} - \frac{1}{\lambda_{t+1}} \right)\left( f(A\mathbf{x}_{t+1}^*) - f(A\mathbf{x}_{t+1}) \right) \\
& \leqslant \widetilde{F}_{\lambda_t}(\mathbf{x}_{t+1}) - \widetilde{F}_{\lambda_t}(\mathbf{x}_{t+1}^*) + c_f\left( \frac{1}{\lambda_t} - \frac{1}{\lambda_{t+1}} \right) \\
& \leqslant \widetilde{F}_{\lambda_t}(\mathbf{x}_{t+1}) - \widetilde{F}_{\lambda_t}(\mathbf{x}_t^*) + c_f\left( \frac{1}{\lambda_t} - \frac{1}{\lambda_{t+1}} \right) \\
& \leqslant (1-\gamma_t)\widetilde{H}_t + \gamma_t^2 R^2 \frac{\frac{L}{\lambda_t}+1}{2} + c_f\left( \frac{1}{\lambda_t} - \frac{1}{\lambda_{t+1}} \right),
\end{aligned}
\tag{39}
$$

where $c_f = \max_{\mathbf{x},\mathbf{y} \in \times_{i \in I}\mathcal{X}_i} f(A\mathbf{x}) - f(A\mathbf{y}) < \infty$. The result in (39) is analogous to the result in Lemma 3.2 in Woodstock and Pokutta [2025]. However, it involves a difference of inverse values of $\lambda_t$, which will be the key difference here. This allows us to use a larger step size $\gamma_t = \frac{2}{\sqrt{t+2}\ln(t+2)}$ for the SCG method.

For the given $\lambda_t = \ln(t+2)$ and by using $\ln(x) \leqslant x - 1$, we obtain

$$
\frac{1}{\lambda_t} - \frac{1}{\lambda_{t+1}} \leqslant \frac{\lambda_{t+1} - \lambda_t}{\lambda_t{}^2} = \frac{\ln\left(\frac{t+3}{t+2}\right)}{\ln(t+2)^2} \leqslant \frac{1}{(t+2)\ln(t+2)^2}.
\tag{40}
$$

We can now show by induction that

$$
0 \leqslant \widetilde{H}_t \leqslant \widetilde{G}_t \overset{\text{def}}{=} \frac{2R^2(L+1) + \sqrt{2}c_f}{\sqrt{t+2}\ln(t+2)}.
\tag{41}
$$

For $t = 0$ we have

$$\widetilde{H}_0 = \widetilde{F}_{\lambda_0}(\mathbf{x}_0) - \widetilde{F}_{\lambda_0}(\mathbf{x}_0^*)$$

$$= \frac{f(A\mathbf{x}_0) - f(A\mathbf{x}_0^*)}{\lambda_0} + \frac{\text{dist}_{\boldsymbol{D}}^2(\mathbf{x}_0) - \text{dist}_{\boldsymbol{D}}^2(\mathbf{x}_0^*)}{2}$$

$$\leqslant \frac{c_f}{\ln(2)} + \frac{R^2}{2}$$

$$\leqslant \frac{2R^2(L+1) + \sqrt{2}c_f}{\sqrt{2}\ln(2)} = \widetilde{G}_0.$$

The induction hypothesis together with (39) and (40) yields

$$\widetilde{H}_{t+1} \leqslant (1 - \gamma_t)\widetilde{H}_t + \gamma_t^2 R^2 \frac{\left(\frac{L}{\lambda_t} + 1\right)}{2} + c_f\left(\frac{1}{\lambda_t} - \frac{1}{\lambda_{t+1}}\right)$$

$$\leqslant \frac{\sqrt{t+2}\ln(t+2) - 2}{\sqrt{t+2}\ln(t+2)}\widetilde{G}_t + \frac{2R^2(L+1)}{(t+2)\ln(t+2)^2} + \frac{\sqrt{2}c_f}{(t+2)\ln(t+2)^2}$$

$$= \frac{\sqrt{t+2}\ln(t+2) - 2 + 1}{(t+2)\ln(t+2)^2} \cdot (2R^2(L+1) + \sqrt{2}c_f)$$

$$\leqslant \frac{1}{\sqrt{t+3}\ln(t+3)} \cdot (2R^2(L+1) + \sqrt{2}c_f) = \widetilde{G}_{t+1}.$$

To finish the induction, it is left to show that

$$\frac{\sqrt{t+2}\ln(t+2) - 1}{(t+2)\ln(t+2)^2} \leqslant \frac{1}{\sqrt{t+3}\ln(t+3)}.$$

Consider

$$\sqrt{t+2}\ln(t+2) - \sqrt{t+1}\ln(t+1)$$

$$= \ln\left(\frac{(t+2)^{\sqrt{t+2}}}{(t+1)^{\sqrt{t+1}}}\right)$$

$$= \ln\left(\left(\frac{t+2}{t+1}\right)^{\sqrt{t+2}} \cdot \frac{1}{(t+1)^{\sqrt{t+2}-\sqrt{t+1}}}\right)$$

$$= \sqrt{t+2}\ln\left(1 + \frac{1}{t+1}\right) - (\sqrt{t+2} - \sqrt{t+1})\ln(t+1)$$

$$\leqslant \frac{\sqrt{t+2}}{t+1} \overset{\text{def}}{=} \phi(t).$$

The function $\phi(t)$ is monotonically decreasing and satisfies $\phi(t) \leqslant 1$ for $t \geqslant 1$. Thus, we have

$$\sqrt{t+2}\ln(t+2) - 1 \leqslant \sqrt{t+1}\ln(t+1), \tag{42}$$

for all $t \geqslant 1$. One can even see that (42) holds for $t = 0$. It remains to show that

$$\sqrt{t+1}\ln(t+1)\sqrt{t+3}\ln(t+3) \leqslant (t+2)\ln(t+2)^2.$$

One can easily see that

$$\sqrt{t+1}\sqrt{t+3} = \sqrt{t^2 + 4t + 3} \leqslant \sqrt{t^2 + 4t + 4} = t + 2.$$

Furthermore, using twice the concavity of $\ln$ yields

$$\ln(\ln(t+1) \cdot \ln(t+3)) = \ln(\ln(t+1)) + \ln(\ln(t+3))$$

$$\leqslant 2\ln\left(\frac{\ln(t+1)}{2} + \frac{\ln(t+3)}{2}\right)$$

$$\leqslant 2\ln\left(\ln\left(\frac{t+1}{2} + \frac{t+3}{2}\right)\right)$$

$$= 2\ln(\ln(t+2)) = \ln(\ln(t+2)^2),$$

and thus

$$\ln(t+1) \cdot \ln(t+3) \leqslant \ln(t+2)^2.$$

This concludes the induction. Using $H_t = \lambda_t \widetilde{H}_t$ yields the proposed upper boundary

$$H_t = \lambda_t \widetilde{H}_t \leqslant \frac{2R^2(L+1) + \sqrt{2}c_f}{\sqrt{t+2}} = \mathcal{O}\left(\frac{1}{\sqrt{t}}\right).$$

Together with Proposition 14 we get that $\lim_{t \to \infty} F_{\lambda_t}(\mathbf{x}_t)$ exists and that

$$\lim_{t \to \infty} F_{\lambda_t}(\mathbf{x}_t) = \lim_{t \to \infty} F_{\lambda_t}(\mathbf{x}_t^*) = \inf_{\mathbf{x} \in \times_{i \in I} \mathcal{X}_i} f(\mathbf{x}).$$

As $\lambda_t \to \infty$, we have $\text{dist}_{\mathbf{D}}^2(\mathbf{x}_t) \to 0$. Thus, every accumulation point $\tilde{\mathbf{x}}$ lies in the diagonal space $\mathbf{D}$ and therefore $A\tilde{\mathbf{x}} \in \bigcap_{i \in I} \mathcal{X}_i$. Considering a subsequence $(t_k)_{k \in \mathbb{N}}$ we get

$$\inf_{\mathbf{x} \in \bigcap_{i \in I} \mathcal{X}_i} f(\mathbf{x}) \leqslant f(A\tilde{\mathbf{x}}) = \lim_{k \to \infty} f(A\mathbf{x}_{t_k}) \leqslant \lim_{k \to \infty} F_{\lambda_{t_k}}(\mathbf{x}_{t_k}) = \inf_{\mathbf{x} \in \bigcap_{i \in I} \mathcal{X}_i} f(\mathbf{x}).$$

$\square$

# E   Second-Order Conditional Gradient Sliding

In this section, we first introduce the SOCGS algorithm before proving its convergence for self-concordant functions.

## E.1   SOCGS algorithm

Introduced in Carderera and Pokutta [2020], the SOCGS algorithm minimizes a smooth strongly convex function $f$ with Lipschitz continuous Hessian $\nabla^2 f$ over a polytope $\mathcal{X}$. For each iteration $t$, a quadratic approximation $\hat{f}_t$ is minimized over the polytope $\mathcal{X}$ using a projection-free method. Minimizing such quadratic forms over a polytope amounts to a *Projected Variable-Metric* (PVM) algorithm Nesterov [2018], Ben-Tal and Nemirovski [Spring 2023] (we state PVM in Algorithm 11).

To do so, a Hessian oracle $\Omega$ yields for each iteration $t$ an approximation $\mathbf{H}_t$ of the Hessian $\nabla^2 f(\mathbf{x}_t)$ at the current iterate $\mathbf{x}_t$. The quadratic approximation[1] $\hat{f}_t(\mathbf{x}) = \langle \nabla f(\mathbf{x}_t), \mathbf{x} - \mathbf{x}_t \rangle + \frac{1}{2} \|\mathbf{x} - \mathbf{x}_t\|_{\mathbf{H}_t}^2$ is then built and minimized inexactly using a Corrective Frank-Wolfe algorithm. We call this an *Inexact PVM step*. This step is solved up to some precision on the Frank-Wolfe gap given by the threshold $\varepsilon_t$. This threshold involves the computation of a lower bound $lb(\mathbf{x}_t) \leqslant f(\mathbf{x}_t) - f^*$ on the primal gap.

On top of the Inexact PVM step, the SOCGS algorithm also performs independent corrective steps, as presented in Section 2. We call these steps the *Outer Corrective Steps* (OCS), or *outer steps* for short. Hence, the SOCGS algorithm enjoys the global convergence rate of the outer steps and the local convergence rate of the Inexact PVM steps.

Now, we present the pseudo-code of the SOCGS algorithm. The statement of Algorithm 10 is directly adapted from Carderera and Pokutta [2020]. Compared to its original statement, we distinguish the OCS from the *Inner Corrective Steps* (ICS), or *inner steps* for short, used inside the Inexact PVM step.

---

[1]The norm $\|\mathbf{x} - \mathbf{y}\|_{\mathbf{H}} = \left\|\mathbf{H}^{1/2}(\mathbf{x} - \mathbf{y})\right\|$ is induced by a symmetric positive definite matrix $\mathbf{H}$.

**Algorithm 10** Second-Order Conditional Gradient Sliding (SOCGS)

**Require:** Point $\mathbf{x} \in \mathcal{X}$
**Return:** Point $\mathbf{x}_T \in \mathcal{X}$

1: $\mathbf{x}_0 \leftarrow \operatorname{argmin}_{\mathbf{v} \in \mathcal{X}} \langle \nabla f(\mathbf{x}), \mathbf{v} \rangle, S_0 \leftarrow \{\mathbf{x}_0\}, \boldsymbol{\lambda}_0(\mathbf{x}_0) \leftarrow 1$
2: $\mathbf{x}_0^{\text{OCS}} \leftarrow \mathbf{x}_0, S_0^{\text{OCS}} \leftarrow S_0, \boldsymbol{\lambda}_0^{\text{OCS}}(\mathbf{x}_0) \leftarrow 1$
3: **for** $t = 0$ to $T - 1$ **do**
4: $\quad$ $\mathbf{x}_{t+1}^{\text{OCS}}, S_{t+1}^{\text{OCS}}, \boldsymbol{\lambda}_{t+1}^{\text{OCS}} \leftarrow \text{OCS}(\nabla f(\mathbf{x}_t), \mathbf{x}_t^{\text{OCS}}, S_t^{\text{OCS}}, \boldsymbol{\lambda}_t^{\text{OCS}})$ $\qquad$ ▷ Outer Corrective Step
5: $\quad$ $\mathbf{H}_t \leftarrow \Omega(\mathbf{x}_t)$ $\qquad\qquad\qquad\qquad\qquad\qquad\qquad\qquad\qquad\qquad$ ▷ Call Hessian oracle
6: $\quad$ $\hat{f}_t(\mathbf{x}) \leftarrow \langle \nabla f(\mathbf{x}_t), \mathbf{x} - \mathbf{x}_t \rangle + \frac{1}{2} \|\mathbf{x} - \mathbf{x}_t\|_{\mathbf{H}_t}^2$ $\qquad\qquad$ ▷ Build quadratic approximation
7: $\quad$ $\varepsilon_t \leftarrow \left( \frac{lb(\mathbf{x}_t)}{\|\nabla f(x_t)\|} \right)^4$
8: $\quad$ $\tilde{\mathbf{x}}_{t+1}^0 \leftarrow \mathbf{x}_t, \tilde{S}_{t+1}^0 \leftarrow S_t, \tilde{\boldsymbol{\lambda}}_{t+1}^0 \leftarrow \boldsymbol{\lambda}_t, h \leftarrow 0$
9: $\quad$ **while** $\max_{\mathbf{v} \in \mathcal{X}} \left\langle \nabla \hat{f}_t(\tilde{\mathbf{x}}_{t+1}^h), \tilde{\mathbf{x}}_{t+1}^h - \mathbf{v} \right\rangle \geqslant \varepsilon_t$ **do** $\qquad$ ▷ Compute Inexact PVM step
10: $\quad\quad$ $\tilde{\mathbf{x}}_{t+1}^{h+1}, \tilde{S}_{t+1}^{h+1}, \tilde{\boldsymbol{\lambda}}_{t+1}^{h+1} \leftarrow \text{ICS}(\nabla \hat{f}_t(\tilde{\mathbf{x}}_{t+1}^h), \tilde{\mathbf{x}}_{t+1}^h, \tilde{S}_{t+1}^h, \tilde{\boldsymbol{\lambda}}_{t+1}^h)$ $\qquad$ ▷ Inner Corrective Step
11: $\quad\quad$ $h \leftarrow h + 1$
12: $\quad$ **end while**
13: $\quad$ $\mathbf{x}_{t+1}^{\text{PVM}} \leftarrow \tilde{\mathbf{x}}_{t+1}^h, S_{t+1}^{\text{PVM}} \leftarrow \tilde{S}_{t+1}^h, \boldsymbol{\lambda}_{t+1}^{\text{PVM}} \leftarrow \tilde{\boldsymbol{\lambda}}_{t+1}^h$
14: $\quad$ **if** $f(\mathbf{x}_{t+1}^{\text{PVM}}) \leqslant f(\mathbf{x}_{t+1}^{\text{OCS}})$ **then**
15: $\quad\quad$ $\mathbf{x}_{t+1} \leftarrow \mathbf{x}_{t+1}^{\text{PVM}}, S_{t+1} \leftarrow S_{t+1}^{\text{PVM}}, \boldsymbol{\lambda}_{t+1} \leftarrow \boldsymbol{\lambda}_{t+1}^{\text{PVM}}$ $\qquad\qquad$ ▷ Choose Inexact PVM step
16: $\quad$ **else**
17: $\quad\quad$ $\mathbf{x}_{t+1} \leftarrow \mathbf{x}_{t+1}^{\text{OCS}}, S_{t+1} \leftarrow S_{t+1}^{\text{OCS}}, \boldsymbol{\lambda}_{t+1} \leftarrow \boldsymbol{\lambda}_{t+1}^{\text{OCS}}$ $\qquad\qquad$ ▷ Choose Outer Corrective Step
18: $\quad$ **end if**
19: **end for**

### E.2 Convergence with generalized self-concordant functions

We first recall the definition of generalized self-concordant functions as introduced in Sun and Tran-Dinh [2019, Definition 2]. Let $f : \mathbb{R}^n \to \mathbb{R} \cup \{+\infty\}$ be a convex function, with its effective domain $\operatorname{dom}(f) \stackrel{\text{def}}{=} \{\mathbf{x} \in \mathbb{R}^n | f(\mathbf{x}) < +\infty\}$. We assume that $\operatorname{dom}(f)$ is an open set and that the function $f$ is three times continuously differentiable on $\operatorname{dom}(f)$ with third order derivative $\nabla^3 f$. The function $f$ is a $(M, \nu)$-*generalized self-concordant function of order* $\nu > 0$ *and constant* $M \geqslant 0$ if

$$\left| \langle \nabla^3 f(\mathbf{x})_{\mathbf{v}} \mathbf{u}, \mathbf{u} \rangle \right| \leqslant M \|\mathbf{u}\|_{\nabla^2 f(\mathbf{x})}^2 \|\mathbf{v}\|_{\nabla^2 f(\mathbf{x})}^{\nu - 2} \|\mathbf{v}\|^{3 - \nu}, \quad \forall \mathbf{x} \in \operatorname{dom}(f), \forall \mathbf{u}, \mathbf{v} \in \mathbb{R}^n. \tag{43}$$

This bound on the third derivative can be used to derive inequalities akin to generalized smoothness and generalized strong convexity.

**Proposition 16.** *[Sun and Tran-Dinh, 2019, Proposition 10] Given an $(M, \nu)$-generalized self-concordant function $f$, then for $\nu \geqslant 2$, we have that:*

$$f(\mathbf{y}) - f(\mathbf{x}) - \langle \nabla f(\mathbf{x}), \mathbf{y} - \mathbf{x} \rangle \leqslant \omega_\nu(d_\nu(\mathbf{x}, \mathbf{y})) \|\mathbf{y} - \mathbf{x}\|_{\nabla^2 f(\mathbf{x})}^2, \tag{44}$$

$$f(\mathbf{y}) - f(\mathbf{x}) - \langle \nabla f(\mathbf{x}), \mathbf{y} - \mathbf{x} \rangle \geqslant \omega_\nu(-d_\nu(\mathbf{x}, \mathbf{y})) \|\mathbf{y} - \mathbf{x}\|_{\nabla^2 f(\mathbf{x})}^2, \tag{45}$$

*where* (44) *holds if $d_\nu(\mathbf{x}, \mathbf{y}) < 1$ for $\nu > 2$, and we have that,*

$$d_\nu(\mathbf{x}, \mathbf{y}) \stackrel{\text{def}}{=} \begin{cases} M \|\mathbf{y} - \mathbf{x}\| & \text{if } \nu = 2 \\ (\frac{\nu}{2} - 1) M \|\mathbf{y} - \mathbf{x}\|^{3 - \nu} \|\mathbf{y} - \mathbf{x}\|_{\nabla^2 f(\mathbf{x})}^{\nu - 2} & \text{if } \nu > 2, \end{cases} \tag{46}$$

*where:*

$$\omega_\nu(\tau) \stackrel{\text{def}}{=} \begin{cases} \frac{e^\tau - \tau - 1}{\tau^2} & \text{if } \nu = 2 \\ \frac{-\tau - \ln(1 - \tau)}{\tau^2} & \text{if } \nu = 3 \\ \frac{(1 - \tau) \ln(1 - \tau) + \tau}{\tau^2} & \text{if } \nu = 4 \\ \left( \frac{\nu - 2}{4 - \nu} \right) \frac{1}{\tau} \left[ \frac{\nu - 2}{2(3 - \nu)\tau} \left( (1 - \tau)^{\frac{2(3 - \nu)}{2 - \nu}} - 1 \right) - 1 \right] & \text{otherwise.} \end{cases} \tag{47}$$

**Lemma 17.** *[Karimireddy et al., 2018, Lemma 9] Given a convex set $\mathcal{X}$ and $\mathbf{H} \in \mathbb{S}_{++}^n$, and two scalars $\alpha > 0$, $\beta > 0$ such that $\alpha\beta \geqslant 1$, we have that:*

$$\min_{\mathbf{x} \in \mathcal{X}} \langle \nabla f(\mathbf{x}_k), \mathbf{x} - \mathbf{x}_k \rangle + \frac{\alpha}{2} \|\mathbf{x} - \mathbf{x}_k\|_{\mathbf{H}}^2 \leqslant \frac{1}{\alpha\beta} \min_{\mathbf{x} \in \mathcal{X}} \langle \nabla f(\mathbf{x}_k), \mathbf{x} - \mathbf{x}_k \rangle + \frac{1}{2\beta} \|\mathbf{x} - \mathbf{x}_k\|_{\mathbf{H}}^2.$$

**Lemma 18.** *[Carderera and Pokutta, 2020, Lemma A.6] Given two matrices $\mathbf{P}, \mathbf{Q} \in \mathcal{S}_{++}^n$, then we have for all $\mathbf{v} \in \mathbb{R}^n$:*

$$\frac{1}{\eta} \|\mathbf{v}\|_{\mathbf{P}}^2 \leqslant \|\mathbf{v}\|_{\mathbf{Q}}^2 \leqslant \eta \|\mathbf{v}\|_{\mathbf{P}}^2 , \tag{48}$$

*with $\eta = \max\left\{ \lambda_{\max}\left(\mathbf{P}^{-1}\mathbf{Q}\right), \lambda_{\max}\left(\mathbf{Q}^{-1}\mathbf{P}\right) \right\} \geqslant 1$.*

We now present the PVM algorithm in Algorithm 11 from Carderera and Pokutta [2020] and its convergence for generalized self-concordant functions in Theorem 19.

---

**Algorithm 11** Projected Variable-Metric (PVM) algorithm

---

**Require:** Point $\mathbf{x}_0 \in \mathcal{X}$, step sizes $\{\gamma_0, \ldots, \gamma_K\}$
**Return:** Point $\mathbf{x}_K \in \mathcal{X}$
1: **for** $k = 0$ to $K - 1$ **do**
2: $\quad \hat{\mathbf{x}} \leftarrow \operatorname{argmin}_{\mathbf{x} \in \mathcal{X}} \left( f(\mathbf{x}_k) + \langle \nabla f(\mathbf{x}_k), \mathbf{x} - \mathbf{x}_k \rangle + \frac{1}{2} \|\mathbf{x} - \mathbf{x}_k\|_{\mathbf{H}_k}^2 \right)$
3: $\quad \mathbf{x}_{k+1} \leftarrow \mathbf{x}_k + \gamma_k(\hat{\mathbf{x}} - \mathbf{x}_k)$
4: **end for**

---

**Theorem 19** (Global convergence of the Projected Variable-Metric algorithm on generalized self-concordant functions.). *Let $\mathcal{X}$ be a compact convex set of diameter $D$ and $f$ be a $(M, \nu)$-generalized self-concordant function with $\nu \geqslant 2$ such that $f$ is strongly convex on $\operatorname{dom}(f) \cap \mathcal{X}$ if $\nu = 3$. Given a starting point $\mathbf{x}_0 \in \mathcal{X} \cap \operatorname{dom}(f)$, the Projected Variable-Metric algorithm (Algorithm 11) with a step size $\gamma_k$ guarantees for all $k \geqslant 0$:*

$$f(\mathbf{x}_{k+1}) - f(\mathbf{x}^*) \leqslant \left( 1 - \frac{\omega_\nu(1/2)\gamma_k^2}{\eta_k} \right) (f(\mathbf{x}_k) - f(\mathbf{x}^*)),$$

*where the parameter $\eta_k$ measures how well $\mathbf{H}_k$ approximates $\nabla^2 f(\mathbf{x}_k)$ in the sense of Lemma 18, and $\gamma_k$ is such that*

$$\gamma_k \leqslant \min\left\{ \frac{1}{2\eta_k}, \eta_k \right\} \frac{1}{\omega_\nu(\frac{1}{2})} \tag{49}$$

*and additionally,*

$$\gamma_k \leqslant \frac{1}{2\eta_k \omega_\nu(MD)} \text{ if } \nu = 2 \tag{50}$$

$$\gamma_k \leqslant \Gamma, \text{ if } \nu > 2 \tag{51}$$

*where $\Gamma$ is the maximum value, such that:*

$$\|\bar{\mathbf{x}} - \mathbf{x}_k\|_{\mathbf{H}_k} \leqslant \frac{\mu_0^{3-\nu}}{\eta_k M(2\nu - 1)} \tag{52}$$

*where $\bar{\mathbf{x}} = \operatorname*{argmin}_{\mathbf{x}} \langle \nabla f(\mathbf{x}_k), \mathbf{x} - \mathbf{x}_k \rangle + \frac{1}{2\Gamma} \|\mathbf{x} - \mathbf{x}_k\|_{\mathbf{H}_k}^2$,*

$$\text{with } \mu_0 = \begin{cases} 1 & \text{if } \nu = 3 \\ \min_{\substack{\mathbf{d} \in \mathbb{R}^n, \|\mathbf{d}\|_2 = 1 \\ \mathbf{x} \in \mathcal{X}, f(\mathbf{x}) \leqslant f(\mathbf{x}_0)}} \|\mathbf{d}\|_{\nabla^2 f(\mathbf{x})} & \text{otherwise.} \end{cases}$$

*Proof.* The iterate $\mathbf{x}_{k+1}$ can be rewritten as:

$$\mathbf{x}_{k+1} = \operatorname*{argmin}_{\mathbf{x} \in (1-\gamma_k)\mathbf{x}_k + \gamma_k \mathcal{X}} \langle \nabla f(\mathbf{x}_k), \mathbf{x} - \mathbf{x}_k \rangle + \frac{1}{2\gamma_k} \|\mathbf{x} - \mathbf{x}_k\|_{\mathbf{H}_k}^2 . \tag{53}$$

Using $(M, \nu)$-generalized self-concordance of $f$ and Proposition 16, we can write:

$$f(\mathbf{x}_{k+1}) - f(\mathbf{x}_k) \leqslant \langle \nabla f(\mathbf{x}_k), \mathbf{x}_{k+1} - \mathbf{x}_k \rangle + \omega_\nu(d_\nu(\mathbf{x}_k, \mathbf{x}_{k+1})) \|\mathbf{x}_{k+1} - \mathbf{x}_k\|_{\nabla^2 f(\mathbf{x}_k)}^2 \tag{54}$$

$$\leqslant \langle \nabla f(\mathbf{x}_k), \mathbf{x}_{k+1} - \mathbf{x}_k \rangle + \eta_k \omega_\nu(d_\nu(\mathbf{x}_k, \mathbf{x}_{k+1})) \|\mathbf{x}_{k+1} - \mathbf{x}_k\|_{\mathbf{H}_k}^2 , \tag{55}$$

where (55) follows from the $\eta_k$ approximation of the Hessian by $\mathbf{H}_k$. If $\nu = 2$, we have from (50) that

$$\gamma_k \leqslant \frac{1}{2\eta_k \omega_\nu(MD)} \leqslant \frac{1}{2\eta_k \omega_\nu(d_\nu(\mathbf{x}_k, \mathbf{x}_{k+1}))}.$$

Note that we use the fact that $\omega_\nu(a) \leqslant \omega_\nu(1/2)$ hold for all $a \leqslant 1/2$, which we will also use for other $\nu$ values. If $\nu > 2$, using the upper bound assumption on $\gamma_k$ (51) and the associated upper bound on the $\mathbf{H}_k$-norm (52), we can ensure that:

$$d_\nu(\mathbf{x}_{k+1}, \mathbf{x}_k) = \left(\frac{\nu}{2} - 1\right) M \|\mathbf{x}_{k+1} - \mathbf{x}_k\|_2^{3-\nu} \|\mathbf{x}_{k+1} - \mathbf{x}_k\|_{\nabla f(\mathbf{x}_k)}^{\nu-2}$$

$$\leqslant \left(\frac{\nu}{2} - 1\right) M \frac{1}{\mu_0^{3-\nu}} \|\mathbf{x}_{k+1} - \mathbf{x}_k\|_{\nabla f(\mathbf{x}_k)}$$

$$\leqslant \left(\frac{\nu}{2} - 1\right) M \frac{\eta_k}{\mu_0^{3-\nu}} \|\mathbf{x}_{k+1} - \mathbf{x}_k\|_{\mathbf{H}_k} \leqslant \frac{1}{2},$$

where the last inequality uses the maximum distance in local norm between $\mathbf{x}_k$ and $\mathbf{x}_{k+1}$ as a solution to the subproblem. Note that $\Gamma > 0$ can be ensured, among other means, by the fact that $\nabla f(\mathbf{x}_k)$ is bounded on any finite sublevel set even if the function does not possess a global finite Lipschitz smoothness constant. These two cases ensure that $\frac{1}{2\gamma_k} \geqslant \eta_k \omega_\nu(d_\nu(\mathbf{x}_k, \mathbf{x}_{k+1}))$. Continuing the chain of inequalities:

$$f(\mathbf{x}_{k+1}) - f(\mathbf{x}_k) \leqslant \langle \nabla f(\mathbf{x}_k), \mathbf{x}_{k+1} - \mathbf{x}_k \rangle + \eta_k \omega_\nu(d_\nu(\mathbf{x}_k, \mathbf{x}_{k+1})) \|\mathbf{x}_{k+1} - \mathbf{x}_k\|_{\mathbf{H}_k}^2 \quad (56)$$

$$\leqslant \langle \nabla f(\mathbf{x}_k), \mathbf{x}_{k+1} - \mathbf{x}_k \rangle + \frac{1}{2\gamma_k} \|\mathbf{x}_{k+1} - \mathbf{x}_k\|_{\mathbf{H}_k}^2 \quad (57)$$

$$= \min_{\mathbf{x} \in (1-\gamma_k)\mathbf{x}_k + \gamma_k \mathcal{X}} \left( \langle \nabla f(\mathbf{x}_k), \mathbf{x} - \mathbf{x}_k \rangle + \frac{1}{2\gamma_k} \|\mathbf{x} - \mathbf{x}_k\|_{\mathbf{H}_k}^2 \right), \quad (58)$$

where (57) follows from the upper bound on $\gamma_k$ from (49). (58) directly follows from the definition of $\mathbf{x}_{k+1}$ in (53).

$$f(\mathbf{x}_{k+1}) - f(\mathbf{x}_k)$$

$$\leqslant \min_{\mathbf{x} \in (1-\gamma_k)\mathbf{x}_k + \gamma_k \mathcal{X}} \left( \langle \nabla f(\mathbf{x}_k), \mathbf{x} - \mathbf{x}_k \rangle + \frac{1}{2\gamma_k} \|\mathbf{x} - \mathbf{x}_k\|_{\mathbf{H}_k}^2 \right) \quad (59)$$

$$\leqslant \frac{\omega_\nu(-d_\nu(\mathbf{x}_k, \mathbf{x}^*))\gamma_k}{\eta_k} \min_{\mathbf{x} \in (1-\gamma_k)\mathbf{x}_k + \gamma_k \mathcal{X}} \left( \langle \nabla f(\mathbf{x}_k), \mathbf{x} - \mathbf{x}_k \rangle + \frac{\omega_\nu(-d_\nu(\mathbf{x}_k, \mathbf{x}^*))}{2\eta_k} \|\mathbf{x} - \mathbf{x}_k\|_{\mathbf{H}_k}^2 \right) \quad (60)$$

$$\leqslant \frac{\omega_\nu(-d_\nu(\mathbf{x}_k, \mathbf{x}^*))\gamma_k}{\eta_k} \min_{\mathbf{x} \in (1-\gamma_k)\mathbf{x}_k + \gamma_k \mathcal{X}} \left( \langle \nabla f(\mathbf{x}_k), \mathbf{x} - \mathbf{x}_k \rangle + \omega_\nu(-d_\nu(\mathbf{x}_k, \mathbf{x}^*)) \|\mathbf{x} - \mathbf{x}_k\|_{\nabla^2 f(\mathbf{x}_k)}^2 \right) \quad (61)$$

$$\leqslant \frac{\omega_\nu(-d_\nu(\mathbf{x}_k, \mathbf{x}^*))\gamma_k^2}{\eta_k} \left( \langle \nabla f(\mathbf{x}_k), \mathbf{x}^* - \mathbf{x}_k \rangle + \omega_\nu(-d_\nu(\mathbf{x}_k, \mathbf{x}^*))\gamma_k \|\mathbf{x}^* - \mathbf{x}_k\|_{\nabla^2 f(\mathbf{x}_k)}^2 \right) \quad (62)$$

$$\leqslant \frac{\omega_\nu(-d_\nu(\mathbf{x}_k, \mathbf{x}^*))\gamma_k^2}{\eta_k} \left( \langle \nabla f(\mathbf{x}_k), \mathbf{x}^* - \mathbf{x}_k \rangle + \omega_\nu(-d_\nu(\mathbf{x}_k, \mathbf{x}^*)) \|\mathbf{x}^* - \mathbf{x}_k\|_{\nabla^2 f(\mathbf{x}_k)}^2 \right) \quad (63)$$

$$\leqslant \frac{\omega_\nu(1/2)\gamma_k^2}{\eta_k} \left( f(\mathbf{x}^*) - f(\mathbf{x}_k) \right). \quad (64)$$

We obtain (60) by applying Lemma 17 with $\alpha = 1/\gamma_k$ and $\beta = \eta_k/\omega_\nu(-d_\nu(\mathbf{x}_k, \mathbf{x}^*))$. Note that the lemma requirements impose $\gamma_k \leqslant \eta_k/\omega_\nu(-d_\nu(\mathbf{x}_k, \mathbf{x}^*))$, which is ensured by (49) by monotonicity of $\omega_\nu$. (61) follows from the Hessian-induced norm approximation with $\mathbf{H}_k$, i.e., $1/\eta_k \|\mathbf{x} - \mathbf{x}_k\|_{\mathbf{H}_k}^2 \leqslant \|\mathbf{x} - \mathbf{x}_k\|_{\nabla^2 f(\mathbf{x}_k)}^2$ following Lemma 18. (62) follows from setting in $\mathbf{x} = (1-\gamma_k)\mathbf{x}_k + \gamma_k \mathbf{x}^*$ into (61) (since $\mathbf{x}^* \in \mathcal{X}$). We obtain (63) by considering that $\gamma_k \leqslant 1$, since $\mathbf{x}_{k+1}$ is constructed as a convex combination of $\hat{\mathbf{x}}$ and $\mathbf{x}_k$. Finally, (64) follows from (45) and from the fact that $\omega_\nu(a) \leqslant \omega_\nu(1/2)$ hold for all $a \leqslant 1/2$. We can finally rewrite the expression as:

$$f(\mathbf{x}_{k+1}) - f(\mathbf{x}^*) \leqslant \left( 1 - \frac{\omega_\nu(1/2)\gamma_k^2}{\eta_k} \right) \left( f(\mathbf{x}_k) - f(\mathbf{x}^*) \right).$$

If $\eta_k$ remains bounded above and below across iterations, a fixed step size respecting the hypotheses from the initial statement achieves linear convergence. A step size $\gamma_k$ obtained through line search will achieve more progress per iteration than a fixed step size respecting the provided bounds and hence also achieves linear convergence. $\qquad\square$

**Corollary 20.** *The SOCGS algorithm applied to a generalized self-concordant objective $f$ of parameters $(M, \nu)$ with $\nu \geqslant 2$ on a polytope $\mathcal{X}$ achieves linear convergence when performing Blended Pairwise Conditional Gradients or Away-Step Frank-Wolfe inner steps for the subproblems if $f$ is strongly convex or if it is the composition of a log-homogeneous barrier with an affine map.*

Linear convergence of BPCG and AFW on strongly convex generalized self-concordant functions was established in Carderera et al. [2024] while the case of the composition of a log-homogeneous barrier (a special case of self-concordant functions) was tackled in Zhao [2025] for AFW and extended to BPCG in Hendrych et al. [2023]. Linear convergence of SOCGS itself follows from observing that the algorithm selects the best primal progress between the FW variant and the Projected Variable-Metric step, both of which provide linear convergence. Finally, we highlight that this result also applies to CGS by using, e.g., AFW or BPCG algorithms for the projection subproblems and starting from a first point in $\mathrm{dom}(f)$. Since the algorithm provides primal progress at each iteration, convergence follows from $\eta_k = \max\{L_0, 1/\mu_0\}$ where $L_0, \mu_0$ are *local* smoothness and strong convexity parameters computed on the sublevel set of $f(\mathbf{x}_0)$, the value of the initial point.

# F  Experiment details and additional computational results

In this section, we provide additional details on the experiments presented in Section 4 as well as present additional experiments. For all problems, we use hybrid methods combining QC-MNP or QC-LP with local pairwise steps of the BPCG algorithm. The steps are combined as follows. We use LCFW with $J = 2$ and local pairwise steps as the default corrective step. Additionally, if a given number of atoms $N$ is added to the active set, a single QC step is performed and the counter is reset afterwards. This hybrid approach yields a good trade-off between the computational cost of solving the linear system or LP, respectively, and the gained acceleration by the QC methods. We consider hybrid approaches with other FW methods in Section F.6. As baselines, we use lazified versions of FW, AFW, PFW, and BPCG. All methods use the secant line search strategy from Hendrych et al. [2025], yielding sufficient progress. For the actual implementation, we have used the `FrankWolfe.jl` package [Besançon et al., 2022]. Furthermore, we use the implementation and setup of Liu et al. [2025] for the entanglement detection problem.

## F.1  $K$-Sparse regression

In the first experiment, we consider a sparse regression problem over the $K$-Sparse polytope, i.e., the intersection of an $\ell_1$-norm ball and an $\ell_\infty$-norm ball, $P_K(\tau) = B_1(\tau K) \cap B_\infty(\tau)$. The vertices of the polytope are given by the vectors with entries in $\{-\tau, 0, \tau\}$ with at most $K$ non-zeros. We consider a classical linear regression problem, i.e., solving

$$\min_{\mathbf{x} \in P_K(\tau)} f(\mathbf{x}) = \sum_{i=1}^{m} (\langle \mathbf{x}, \mathbf{a}_i \rangle - y_i)^2 = \|\mathbf{A}\mathbf{x} - \mathbf{y}\|_2^2$$

given data points $\{(\mathbf{a}_i, y_i)\}_{i=1}^{m} \subset \mathbb{R}^n \times \mathbb{R}$. We used synthetic data for the experiment and generated normally distributed $\mathbf{a}_i$ and $y_i$ with $n = 500$ and $m = 10000$. In Section 4 we already presented the results for the case $K \in \{5, 20\}$ and $\tau = 1$. Here we provide some more insights into the advantage of the QC methods by analyzing the size of the active set for the more extreme cases $K \in \{3, 30\}$. Again, we use a fixed interval length of $N = 10$ for the quadratic correction steps. The results are shown in Figure 5. For smaller $K$, the necessary active set size for reaching optimality is larger. Both QC methods benefit from avoiding running many pairwise steps like BPCG. Therefore, these methods perform earlier FW steps and add more atoms to the active set during earlier iterations. Consequently, both QC methods accelerate the convergence of the primal values and the FW gap.

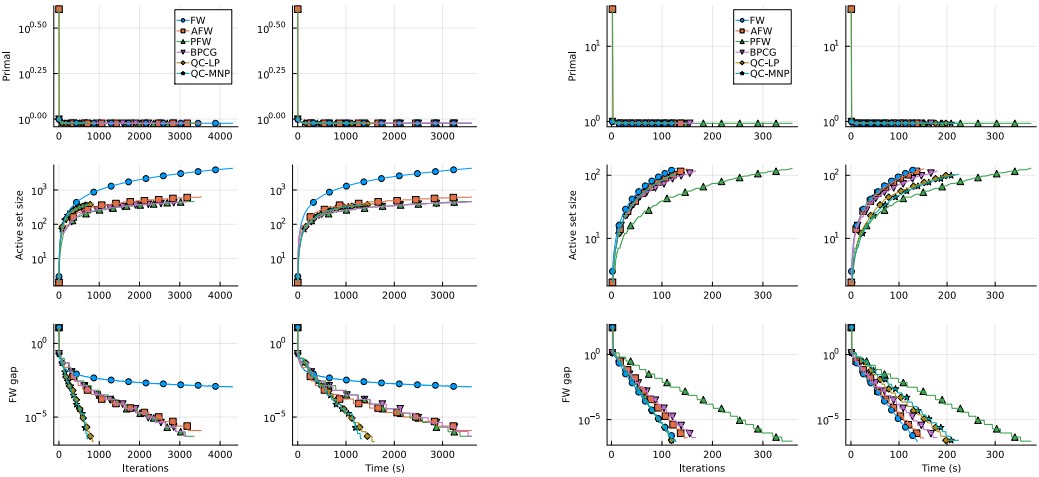

Figure 5: Sparse regression with $K \in \{3, 30\}$ and $\tau = 1$

## F.2 Entanglement detection

In the second experiment, we consider bipartite entanglement detection. Solving this problem is equivalent to projecting a given state onto the set of separable states,

$$\mathcal{S}_{AB} = \text{conv}\{\boldsymbol{\rho}_A \otimes \boldsymbol{\rho}_B : \boldsymbol{\rho}_A \in \mathbb{D}(A), \boldsymbol{\rho}_B \in \mathbb{D}(B)\},$$

where $\otimes$ denotes the tensor product and where $\mathbb{D}(A)$ and $\mathbb{D}(B)$ are the sets of density matrices on systems $A$ and $B$ respectively, i.e., (hermitian) positive semidefinite matrices with unit trace. Consequently, the projection problem can be written as

$$\min_{\boldsymbol{\rho} \in \mathcal{S}_{AB}} \|\boldsymbol{\rho} - \boldsymbol{\rho}_0\|_F^2,$$

where $\|\cdot\|_F$ is the Frobenius norm and $\boldsymbol{\rho}_0$ is the given state. In our experiments, we consider a family of bipartite $3 \times 3$ entangled states proposed in Horodecki [1997]. Given a parameter $a \in [0, 1]$, these states are defined by

$$\boldsymbol{\rho}_H^a = \frac{1}{8a+1} \begin{pmatrix} a & 0 & 0 & 0 & a & 0 & 0 & 0 & a \\ 0 & a & 0 & 0 & 0 & 0 & 0 & 0 & 0 \\ 0 & 0 & a & 0 & 0 & 0 & 0 & 0 & 0 \\ 0 & 0 & 0 & a & 0 & 0 & 0 & 0 & 0 \\ a & 0 & 0 & 0 & a & 0 & 0 & 0 & a \\ 0 & 0 & 0 & 0 & 0 & a & 0 & 0 & 0 \\ 0 & 0 & 0 & 0 & 0 & 0 & \frac{1+a}{2} & 0 & \frac{\sqrt{1-a^2}}{2} \\ 0 & 0 & 0 & 0 & 0 & 0 & 0 & a & 0 \\ a & 0 & 0 & 0 & a & 0 & \frac{\sqrt{1-a^2}}{2} & 0 & \frac{1+a}{2} \end{pmatrix}. \tag{65}$$

The positive partial transpose (PPT) criterion yields necessary and sufficient conditions for systems of the sizes $2 \times 2$ and $2 \times 3$ to be separable; however, it is only necessary for higher-dimensional systems [Horodecki et al., 1996]. For $a \in [0, 1)$, the entangled states $\boldsymbol{\rho}_H^a$ are not detected by PPT [Horodecki, 1997], making them weakly entangled and thus harder to detect, which justifies our choice.

Liu et al. [2025] consider adding white noise to the state, i.e.,

$$\boldsymbol{\rho}_0 = v \boldsymbol{\rho}_H^a + \frac{1-v}{9} \mathbf{I}, \tag{66}$$

for a given noise level $v \in [0, 1]$. In Figure 6 we present the results for different noise levels $v \in \{0.95, 0.97\}$ for a fixed state with parameter $a = 0.5$. We used again a correction interval of $N = 1$ for QC-MNP and $N = 10$ for QC-LP. Comparing the two plots, one can see that adding white noise decreases the distance to $\mathcal{S}_{AB}$ and therefore leads to smaller primal values. Interestingly,

QC-LP is performing worse than BPCG for the pure state. This is because the LP solved by QC-LP is often infeasible, leading to computational overhead without any benefit in terms of primal progress. On the other hand, QC-MNP is reaching optimality in both cases by far the fastest. This indicates that QC-MNP is more suited than QC-LP for non-polytope domains like the set of separable states $\mathcal{S}_{AB}$.

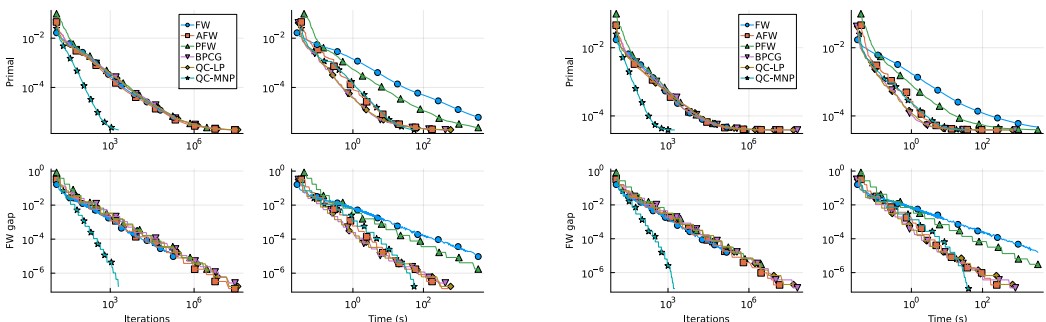

Figure 6: Entanglement detection for a state $\boldsymbol{\rho}_0$ given in (66) with different noise levels $v \in \{0.95, 1.0\}$ applied on a fixed state from (65) with parameter $a = 0.5$

### F.3 Projection onto the intersection of the Birkhoff polytope and a shifted $\ell_2$ ball

**Experiment details** The Birkhoff polytope $B(n)$ is the set of all $n \times n$ doubly stochastic matrices, and its vertices are permutation matrices and therefore particularly sparse. Furthermore, one can solve linear minimization problems over the Birkhoff polytope with a complexity of $\mathcal{O}(n^3)$ using the Hungarian algorithm [Combettes and Pokutta, 2021].

For shifting the center of the $\ell_2$ ball, we first sample vertices $\mathbf{v}_1, \ldots, \mathbf{v}_m$ of the Birkhoff polytope by calling the LMO with uniform random directions $\mathbf{d}_1, \ldots, \mathbf{d}_m$. The ball is then shifted to $\mathbf{s} = \bar{\mathbf{v}} - c \frac{\bar{\mathbf{d}}}{\|\bar{\mathbf{d}}\|_2}$ where $\bar{\mathbf{v}} = \frac{1}{m} \sum_{i=1}^m \mathbf{v}_i$ and $\bar{\mathbf{d}} = \frac{1}{m} \sum_{i=1}^m \mathbf{d}_i$. The radius of the ball is set to $r = 1$, such that the ball and the polytope intersect if and only if $c \leqslant 1$. Note the diameter of the Birkhoff polytope is $\sqrt{2n}$ and therefore much larger than the given radius of the ball. This setting can be understood as a projection onto the Birkhoff polytope with some noise or flexibility in the projection direction.

The number of sampled vertices $m$ controls the dimension of the face where $\bar{\mathbf{v}}$ is located, i.e., the number of non-zero entries in $\bar{\mathbf{v}}$. In particular, the expected number of non-zero entries in $\bar{\mathbf{v}}$ is

$$\mathbb{E}\left[\|\bar{\mathbf{v}}\|_0\right] = n^2 \left(1 - \left(1 - \frac{1}{n}\right)^m\right) \approx n^2 \left(1 - e^{-m/n}\right) = n^2 \cdot q,$$

for $m = -n \ln(1 - q)$. Let $B_2(r, \mathbf{c})$ denote the $\ell_2$ ball of radius $r$ and center $\mathbf{c}$. The problem we consider is,

$$\min_{\mathbf{X} \in B(n) \cap B_2(r,\mathbf{s})} f(\mathbf{X}) = \frac{1}{n^2} \|\mathbf{X} - \mathbf{X}_0\|_F^2,$$

where $\mathbf{X}_0 \in \mathbb{R}^{n \times n}$ is sampled with uniform distributed entries over $[0, 1]$.

Since the SCG method alternates between updating the point on the Birkhoff polytope and the shifted $\ell_2$ ball, we use a cyclic block-coordinate scheme [Lacoste-Julien et al., 2013, Beck et al., 2015] with different update steps on the two sets. While we perform vanilla FW steps for the $\ell_2$ ball, we compare different methods for updating the point on the Birkhoff polytope. Just like in the other experiments, we compare vanilla FW steps, which are used in the original version of SCG, with BPCG and the mentioned hybrid methods. Note, we use the new penalty schedule $\lambda_t = \ln(t + 2)$ proposed in Theorem 9, but not the proposed monotonic step size. BPCG relies on step sizes that consider the current FW gap or pairwise gap to perform proper update steps on the active set. The monotone step size would lead to suboptimal updates and thus give the QC methods an advantage, as they do not depend on any line search. Consequently, we use the new secant line search proposed by Hendrych et al. [2025].

**Additional results** In this paragraph, we present additional results for related problem settings. In particular, we decompose the above problem and solve relaxed settings, a projection just onto the

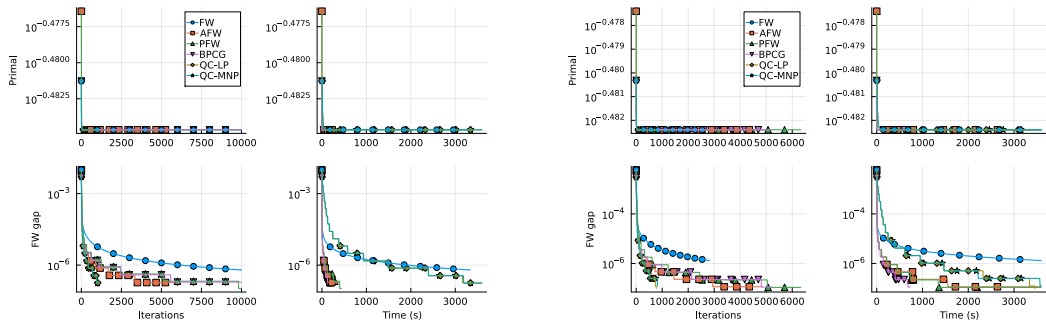

Figure 7: Projection onto the Birkhoff polytope for $n \in \{300, 500\}$

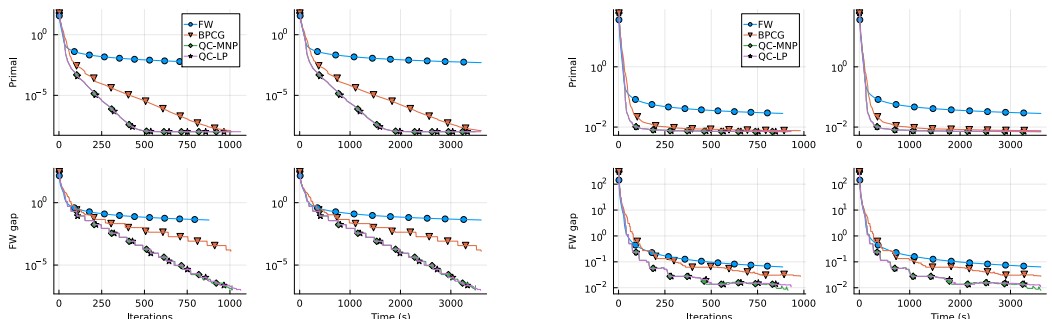

Figure 8: ALM applied to the intersection of a shifted $\ell_2$ ball and the Birkhoff polytope for the intersecting scenario $c = 0.9$ and the disjoint one with $c = 1.1$

Birkhoff polytope, and the intersection problem between the Birkhoff polytope and a shifted $\ell_2$ ball for different intersection scenarios. Furthermore, we present how SCG with vanilla FW steps on both sets perform for the original problem, comparing the new proposed step size schedule with the one from [Woodstock and Pokutta, 2025].

First, we consider the problem of projecting onto the Birkhoff polytope, i.e., without the additional $\ell_2$ ball constraint. We ran the experiment for $n \in \{300, 500\}$ with a time limit of 3600 seconds. For the hybrid methods, we used a fixed interval of $N = 20$ for the quadratic correction. The results are shown in Figure 7. While both QC methods outperform the baselines with respect to the FW gap in terms of the number of iterations, the runtime is worse for both instances. This demonstrates the need for fine-tuning the rate of the quadratic correction, especially for easier problems like projections, where the Hessian is the identity matrix.

In the next set of experiments, we consider the problem of finding a point in the intersection of the Birkhoff polytope and a shifted $\ell_2$ ball. We use the same setup for placing the $\ell_2$ ball as described above. However, since we do not have an additional objective, we use the ALM method by Braun et al. [2023] to solve the problem. We consider the cases of $c \in \{0.9, 1.1\}$, i.e., when the two sets have a full-dimensional intersection and when the sets are disjoint. For all instances, we again used $q = 0.1$, $r = 1$, $n = 500$, and $N = 1$. Additionally, we disabled the quadratic corrections until the active set has at least 30 atoms. This helps to avoid the computational overhead of quadratic corrections in the first iterations when ALM adds and drops atoms very quickly due to its alternating nature. Besides BPCG and the QC methods, we also compare the vanilla FW steps from the original version of ALM. The results of the two experiments are shown in Figure 8.

In both experiments, the two QC methods show a very similar behavior. For $c = 0.9$, i.e., when the two sets have a full-dimensional intersection, both QC methods and BPCG enjoy linear convergence, while the vanilla FW steps show sublinear convergence. However, QC-MNP and QC-LP converge faster and achieve a smaller FW gap, in terms of the number of iterations and time. In the case of $c = 1.1$, i.e., when the two sets are disjoint, all four methods show sublinear convergence. While QC-LP and QC-MNP accelerate the convergence, especially of the FW gap, the benefit of the QC methods is not as pronounced as in the previous experiment.

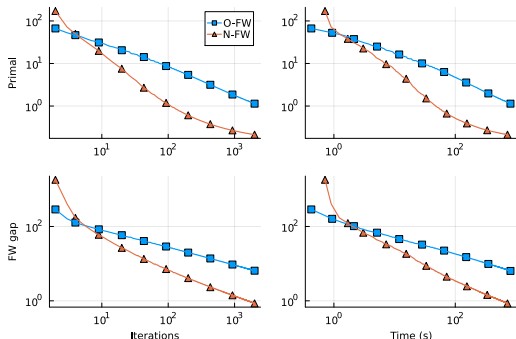

Figure 9: Comparison of the new step size and penalty-schedule (N-FW) proposed in Theorem 9 with the original ones (O-FW) in Woodstock and Pokutta [2025]

Finally, we compare the step size and penalty schedule proposed in Theorem 9 with the ones in the original paper [Woodstock and Pokutta, 2025]. We compare the new setting for vanilla FW steps, which are used in the original version of SCG. For the experiment, we again used $c = 0.9$, $q = 0.1$, and $n = 300$. The results are depicted in Figure 9. We use a logarithmic scale on the horizontal axis to visualize the difference in the convergence rates more clearly. The results confirm our theoretical results that SCG enjoys a faster convergence with the new step size and penalty schedule.

### F.4 Projecting with quadratic correction in Second-Order Conditional Gradient Sliding

In the experiment of Subsection 4.4, we tested the acceleration provided by both quadratic corrections QC-LP and QC-MNP by comparing the hybrid methods with BPCG for solving the Inexact PVM step at Line 9 in Algorithm 10. For the outer step, we use the lazified BPCG.

Due to numerical instabilities for the PVM stop condition at Line 9 in Algorithm 10 (we were getting threshold values $\varepsilon_t \geqslant 0$ too close to 0), we replaced the stop condition at Line 9 by a fixed number $k \in \{50, 200\}$ of inner steps. Tightening the lower bound estimations $lb(\mathbf{x}_t)$ of the primal gaps or designing stop conditions for the Inexact PVM step, which preserve the convergence rate of SOCGS without any lower bound estimation, are left for future work.

During the initial phase of SOCGS, only the outer steps are selected. To avoid the computational overhead of quadratic corrections, we do not perform any QC steps in the first 25 iterations of SOCGS. This leads to identical trajectories between BPCG, QC-LP, and QC-MNP at the beginning in Figure 4. After iteration 25 of SOCGS, both QC-LP and QC-MNP perform in each PVM step a QC step at the first inner iteration. A second quadratic correction is performed after 30 atoms were added to the active set $\tilde{S}_{t+1}^{h+1}$ of the PVM step. This yields a quick adjustment to the new quadratic model and an additional correction if a relevant amount of new vertices were added to the active set. The experiments were run with a limit of 100 SOCGS iterations and a time limit of 2000 seconds.

We consider the same structured logistic regression problem over the $\ell_1$ ball as in Carderera and Pokutta [2020]. We solve

$$\min_{\mathbf{x} \in B_1(1)} \frac{1}{m} \sum_{i=1}^{m} \ln\left(1 + \exp(-y_i \langle \mathbf{x}, \mathbf{z}_i \rangle)\right) + \frac{1}{2m} \|\mathbf{x}\|^2.$$

The labels $y_i \in \{-1, 1\}$ and feature vectors $\mathbf{z}_i \in \mathbb{R}^n$ are taken from the `gisette` training dataset [Guyon et al., 2007], so that $n = 5000$ and $m = 6000$. As mentioned in [Sun and Tran-Dinh, 2019, Subsection 6.1], the function $f$ is an example of $(M, 3)$-generalized self-concordant function as in (43), where $M = \sqrt{m} \max \{\|\mathbf{z}_i\| \mid 1 \leqslant i \leqslant m\}$.

The evaluation of the gradient of $f$,

$$\nabla f(\mathbf{x}) = -\frac{1}{m} \sum_{i=1}^{m} \frac{y_i \mathbf{z}_i}{1 + \exp(y_i \langle \mathbf{x}, \mathbf{z}_i \rangle)} + \frac{1}{m}\mathbf{x} \in \mathbb{R}^n,$$

is computationally demanding, as one has to compute $m$ inner products of size $n$ for each gradient evaluation. For the quadratic approximations $\hat{f}_t$, we use an exact Hessian approximation $\mathbf{H}_t =$

$\nabla^2 f(\mathbf{x}_t)$ with

$$\nabla^2 f(\mathbf{x}) = \frac{1}{m} \sum_{i=1}^{m} \frac{\mathbf{z}_i \mathbf{z}_i^T}{\left(1 + \exp(-y_i \langle \mathbf{x}, \mathbf{z}_i \rangle)\right)\left(1 + \exp(y_i \langle \mathbf{x}, \mathbf{z}_i \rangle)\right)} + \frac{1}{m} \mathbf{I}^n \in \mathbb{R}^{n \times n},$$

where $\mathbf{I}^n \in \mathbb{R}^{n \times n}$ is the identity matrix. The gradients $\nabla \hat{f}_t$ of the quadratic approximations $\hat{f}_t$ are given by $\nabla \hat{f}_t(\mathbf{x}) = \nabla f(\mathbf{x}_t) + \mathbf{H}_t(\mathbf{x} - \mathbf{x}_t)$.

It is worth noticing that in our current implementation, we are storing the full Hessian matrices for each PVM step. This could be avoided by computing Hessian-vector products and Hessian-induced norms for any given $\mathbf{x}$ using the decomposition of the Hessian as rank-one terms given in its expression.

**Experimenting with line search after the Inexact PVM step.** We experimented with adding a secant line search [Hendrych et al., 2025] after the Inexact PVM step and before the `if` clause at Line 14 in Algorithm 10. We did not see a significant change in the trajectories as plotted in Figure 4, except for a slight computational overhead of the line search.

### F.5 Tensor completion

We consider the non-negative tensor completion problem from Bugg et al. [2022], in which one reconstructs a non-negative tensor from some entries. The problem is NP-hard, and the corresponding LMO can be implemented as a mixed-integer linear problem, with a polyhedral feasible set. We compare the Blended Conditional Gradient algorithm [Braun et al., 2019], which was used in the original paper to BPCG, and the two quadratic corrections variants.

Results are presented in Figure 10 for different types of instances. All algorithms use lazification. The BCG, BPCG and PFW methods perform well for instances with a low radius and a low number of vertices forming the optimal solution. The QC algorithms outperform these methods for larger settings of these parameters, and are particularly advantageous for problems in which LMO calls are much costlier than solving any number of linear programs.

### F.6 Performance profile and success rates

In this final section, we investigate how quadratic corrections can accelerate Frank-Wolfe variants beyond BPCG. In particular, we augment three additional methods with both quadratic correction steps (QC-LP and QC-MNP): an active-set variant of vanilla Frank-Wolfe (FW), Away Frank-Wolfe (AFW), and Pairwise Frank-Wolfe (PFW). For clarity of presentation, we omit the prefix "QC" in the method names; for example, Away Frank-Wolfe with QC-MNP is denoted AFW-MNP. In total, we compare twelve methods, four baseline algorithms, and eight variants enhanced with quadratic corrections. We consider 37 quadratic problem instances from the $K$-sparse regression problem, the entanglement detection, the Birkhoff projection, and the tensor completion problem with varying problem parameters and seeds. The experiments with SCG and SOCGS are not considered here.

First, we present a performance profile illustrating the number of problem instances solved within a given time. All methods had a time limit of one hour and 10000 iterations, except for the entanglement detection problem, which allowed $10^8$ iterations. An instance is solved if either the optimality criterion, i.e., FW gap is smaller than $10^{-7}$ (or $10^{-5}$ for the tensor completion problem), or if the iteration or time limit is reached. The results are given in Figure 11.

The graphic demonstrates that QC-MNP improves the performance of all baseline methods. For both small and large instances, methods with QC-MNP belong to the fastest solvers. The QC-LP variants are not as effective as QC-MNP, but still outperform the baseline methods in most cases. This was expected as the benefit of QC-LP is only gained when the affine minimizer lies in the convex hull of the active set, which is not necessary for QC-MNP. In the case of PFW-LP, the benefit of QC-LP is not as pronounced as for other methods. The additional runtime for solving the linear problem is not compensated for by the reduced number of iterations.

Finally, we present the success rates for the aforementioned methods, i.e., the ratio of QC steps that are fully-corrective. There are no guarantees as to when the affine minimizer lies within the convex hull of the active sets. However, the performance of the quadratic correction steps depends heavily on

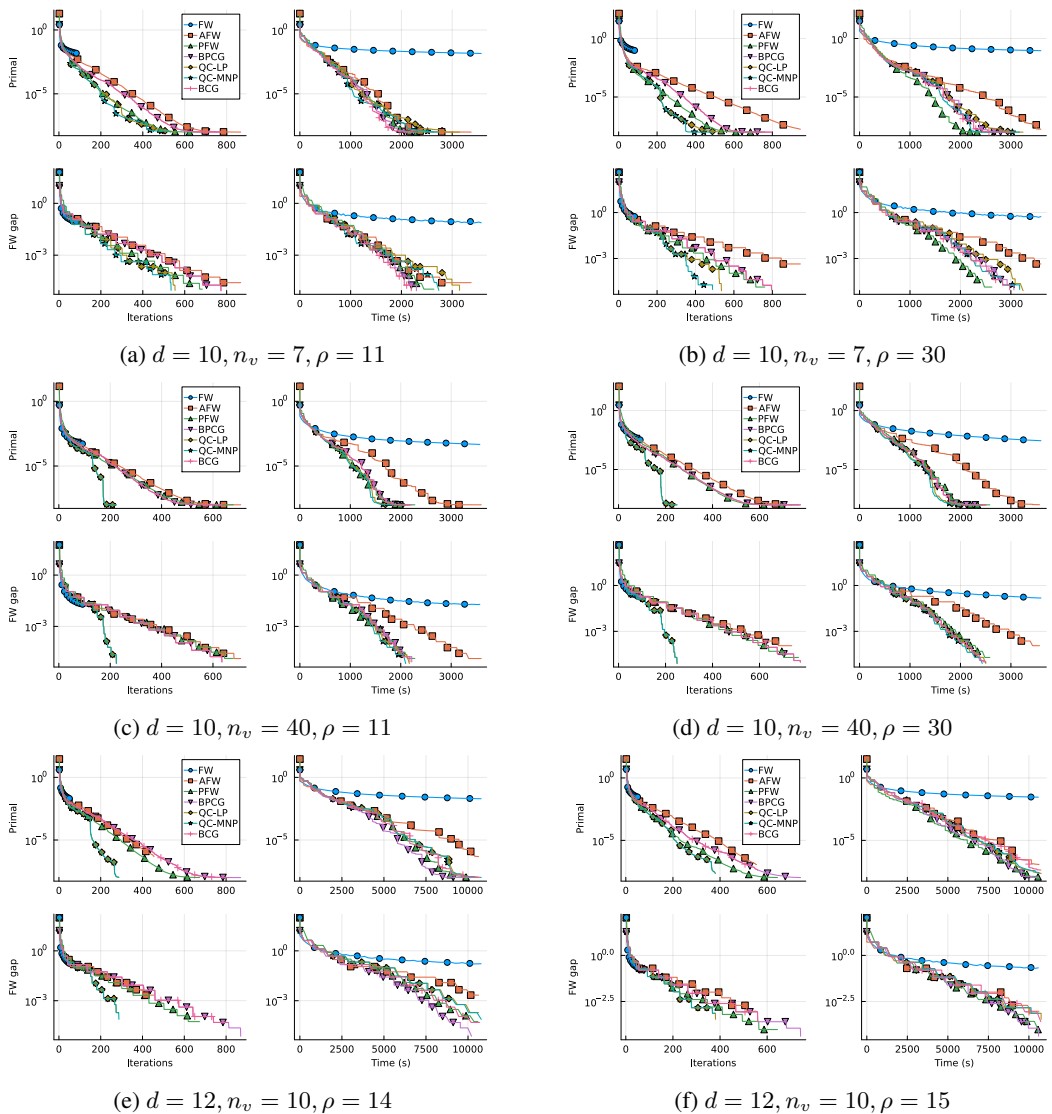

Figure 10: Results on the tensor completion problem. The dimension of the tensor is $d \times d \times d$. The parameter $n_v$ indicates how many vertices were sampled to construct the underlying ground truth, and $\rho$ is the radius of the tensor norm ball.

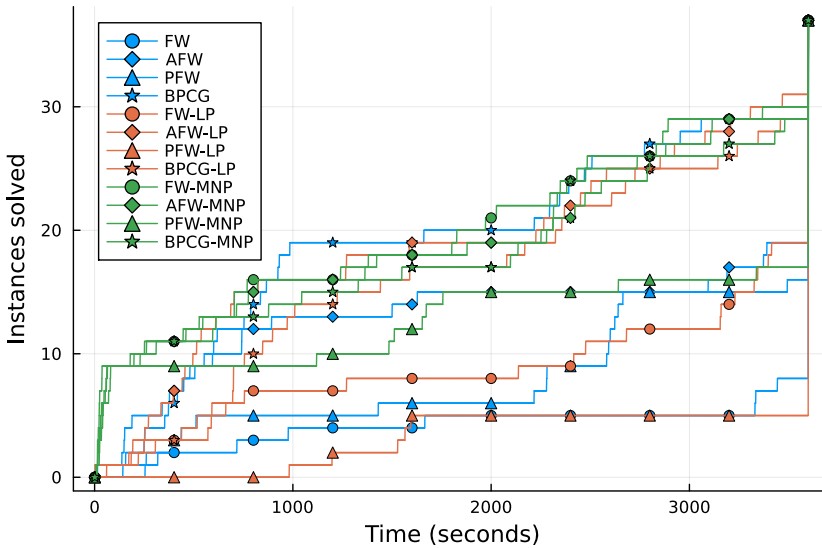

Figure 11: Performance profile of different FW methods with quadratic corrections over a problem set of 37 instances.

this. In cases where the affine minimizer lies outside the convex hull, QC-LP uses the local pairwise step, yielding significantly less progress, and QC-MNP must truncate its update.

The results are given in Table 1 and Table 2. We present the shifted geometric mean of the number of fully-corrective QC steps and the total number of QC steps for different problem instances per experiment.

For both the $K$-sparse regression and the Birkhoff projection problem, the success rates are very high for both QC methods. This result aligns with the fast convergence observed in the previous section, where QC-LP and QC-MNP outperformed the baselines, especially in terms of iteration counts. In the entanglement detection problem, the success rates are moderately high for QC-MNP and very low for QC-LP, explaining why QC-LP performs so poorly, sometimes even slower than some baselines in this problem class. In the tensor completion problem, the success rates are moderately high. However, the overall numbers are comparably small due to the low number of total iterations.

| Method | FW-LP | AFW-LP | PFW-LP | BPCG-LP |
|---|---|---|---|---|
| $K$-sparse regression | 16 / 40 | 29 / 30 | 3 / 4 | 29 / 30 |
| Entanglement detection | 2 / 14803 | 2 / 544 | 1 / 274 | 2 / 473 |
| Birkhoff projection | 0 / 45 | 19 / 28 | 1 / 2 | 18 / 19 |
| Tensor completion | 4 / 6 | 3 / 5 | 1 / 3 | 3 / 4 |

Table 1: Shifted geometric mean of the number of successful QC-LP steps and the total number of QC-LP steps across different problem instances.

| Method | FW-MNP | AFW-MNP | PFW-MNP | BPCG-MNP |
|---|---|---|---|---|
| $K$-sparse regression | 29 / 30 | 29 / 30 | 3 / 5 | 29 / 30 |
| Entanglement detection | 68 / 746 | 492 / 897 | 651 / 935 | 596 / 889 |
| Birkhoff projection | 20 / 24 | 21 / 28 | 1 / 2 | 18 / 19 |
| Tensor completion | 4 / 6 | 3 / 4 | 1 / 3 | 3 / 4 |

Table 2: Shifted geometric mean of the number of successful QC-MNP steps and the total number of QC-MNP steps across different problem instances.

