# OpenReview forum: "Efficient Quadratic Corrections for Frank-Wolfe Algorithms"
_NeurIPS.cc/2025/Conference — NeurIPS 2025 poster_

### Official Review · Reviewer_4JW7 · 2025-07-02

**Clarity:** 3
**Significance:** 4
**Originality:** 3
**Rating:** 4
**Confidence:** 3

**Summary:**

The authors introduce Corrective Frank-Wolfe, a framework unifying many frank-wolfe algorithms. They describe two correction schemes for quadratic problems which are implementations of the new framework. They also mention two specific algorithms which should benefit from the proposed quadratic corrections. Finally the authors present some empirical results demonstrating the convergence rates of their corrected algorithms.

**Questions:**

See questions under “weaknesses”.
Some proofing mistakes:
* In algorithm 1, the global FW vertex uses V(P) which is undefined (I assume you mean V(X)?)
* In 2.1 you don’t prove your theorems, you only state them, so you should write “in the following we state two theorems”. It’s fine to have proofs in the appendix if they aren’t particularly enlightening.

**Ethical Concerns:**

["NO or VERY MINOR ethics concerns only"]

**Final Justification:**

Due to the authors' adequate response to my clarity concerns and their thorough new set of experiments, I'm increasing my scores for quality and clarity. While I still believe the paper requires significant revisions for clarity, particularly in the presentation and explanation of figures, the response demonstrates a clear effort to improve, so I changed my scores accordingly.

**Limitations:**

yes

**Paper Formatting Concerns:**

No major formatting issues in this paper; they adhere to the formatting instructions.

**Quality:**

4

**Strengths And Weaknesses:**

Strengths
1. The framework of Corrected Frank-Wolfe seems to be a useful generalization of existing algorithms, potentially enriching future work.
2. The specific implementation of the framework for quadratic optimization is novel and seems useful as well.
3. The experiments show significant improvement over the baseline.

Weaknesses

1. Remark 1 after proposition 1 should have been elaborated on. Why would choosing the pairwise gap lead to fewer LMO calls than choosing the away gap?
2. The paper is very dense in details making it hard to follow. Almost every paragraph contains either new ideas or new formalisms, which leaves little room for explanations. For example one of the major claims of the paper is about SCG and SOCGS, but SOCGS only has one small paragraph and SCG has a theorem with no proof. From reading the paper alone I couldn’t figure out your contribution to these algorithms.
3. The experiments only compare with one baseline algorithm. More should have been measured, there’s no guarantee that BPCG is the fastest algorithm for the two example problems given, making its choice as baseline questionable.
4. The experimental section contains graphs which aren’t explained well. There are 8 sets of images for each figure but its not clear what’s the differences between them. Are they different experiments? Different seeds?
5. It would be beneficial to add a direct speed comparison between the algorithms. How much faster is your approach at reaching a specified gap value?

---

> ### Author Rebuttal · Authors · 2025-07-31
>
> We thank the referee for their encouragement and feedback and address several points below.
>
> ### Weaknesses
> > Remark 1 after proposition 1 should have been elaborated on. Why would choosing the pairwise gap lead to fewer LMO calls than choosing the away gap?
>
> Thank you for pointing this out. For standard Corrective Frank-Wolfe (CFW), the local pairwise gap yields the same number of LMO calls but fewer FW steps since it is larger than the away gap. The lazified version of CFW, which is introduced later in the paper, only calls the LMO if an FW step is performed. Thus, the number of LMO calls for LCFW is smaller. We will update the corresponding sentence to clarify this.
>
> > The paper is very dense in details making it hard to follow. Almost every paragraph contains either new ideas or new formalisms, which leaves little room for explanations. For example one of the major claims of the paper is about SCG and SOCGS, but SOCGS only has one small paragraph and SCG has a theorem with no proof. From reading the paper alone I couldn’t figure out your contribution to these algorithms.
>
> We will streamline the presentation of the article. The theorem for SCG is proved in the appendix due to space constraints; we will mention it specifically. Similarly, for SOCGS, we will state the theorem on self-concordance in the main part of the text.
>
> > The experiments only compare with one baseline algorithm. More should have been measured, there’s no guarantee that BPCG is the fastest algorithm for the two example problems given, making its choice as baseline questionable.
>
> We agree that relying on a single baseline could limit the conclusions drawn from the experiments.
> To address this, we will add results for additional baseline algorithms, including AFW, PFW, and an active set-based variant of vanilla FW.
> To ensure a fair comparison, we also augmented each of these methods with QC steps using the same hybrid strategy previously applied to BPCG. These additions provide a broader and more balanced evaluation of the practical benefit of QC across different FW methods. We present below the results for the first experiment showing the absolute and relative values of the runtime in seconds and the number of iteration. The relative values in parentheses compare to the corresponding baseline. The full evaluation will be included in the revised version of the paper.
>
> |Runtime in seconds|FW|AFW|PFW|BPCG|FW-LP|AFW-LP|PFW-LP|BPCG-LP|FW-MNP|AFW-MNP|PFW-MNP|BPCG-MNP|
> |-|-|-|-|-|-|-|-|-|-|-|-|-|
> |$K$-Sparse regression $n=500$, $K=5$|2604|1673|1265|1451|**490** (0.19)|534 (0.32)|1197 (0.95)|731 (0.50)|945 (0.36)|950 (0.57)|1374 (1.09)|1130 (0.78)|
> |$K$-Sparse regression $n=500$, K=20$|148|169|222|206|**112** (0.75)|139 (0.83)|249 (1.12)|178 (0.86)|112 (0.75)|146 (0.87)|242 (1.09)|186 (0.90)|
> |$K$-Sparseregression $n=1000, K=5$|3600|3600|3600|3600|3566 (0.99)|**1433** (0.40)|3486 (0.97)|2001 (0.56)|2503 (0.70)|2683 (0.75)|3576 (0.99)|2882 (0.80)|
> |$K$-Sparseregression $n=1000, K=20$|1028|1028|979|1073|486 (0.47)|**548** (0.53)|1454 (1.49)|642 (0.60)|627 (0.61)|654 (0.64)|1467 (1.50)|736 (0.69)|
>
> |Number of iterations|FW|AFW|PFW|BPCG|FW-LP|AFW-LP|PFW-LP|BPCG-LP|FW-MNP|AFW-MNP|PFW-MNP|BPCG-MNP|
> |-|-|-|-|-|-|-|-|-|-|-|-|-|
> |$K$-Sparse regression $n=500, K=5$|10000|5225|3291|3710|**388** (0.04)|431 (0.08)|899 (0.27)|611 (0.16)|**388** (0.04)|431 (0.08)|904 (0.27)|611 (0.16)|
> |$K$-Sparse regression $n=500, K=20$|481|492|784|668|**165** (0.34)|225 (0.46)|414 (0.53)|254 (0.38)|**165** (0.34)|225 (0.46)|408 (0.52)|254 (0.38)|
> |$K$-Sparse regression $n=1000, K=5$|7431|7480|7401|6718|1123 (0.15)|747 (0.10)|1473 (0.20)|1066 (0.16)|**631** (0.08)|700 (0.09)|1159 (0.16)|988 (0.15)|
> |$K$-Sparse regression $n=1000, K=20$|2010|2032|1939|1896|**315** (0.16)|380 (0.19)|736 (0.38)|411 (0.22)|**315** (0.16)|380 (0.19)|721 (0.37)|411 (0.22)|
>
> For each instance, quadratic corrections outperform the baseline both in the number of iterations and runtime. Moreover, QC steps reduce the number of iterations for all baselines by at least 48%. The runtime of the QC-equipped baselines is shorter in every case except for PFW-MNP.
>
> > The experimental section contains graphs which aren’t explained well. There are 8 sets of images for each figure but its not clear what’s the differences between them. Are they different experiments? Different seeds?
>
> Thank you for pointing this out. We will add an explanation of the plots at the beginning of the section, helping the reader to interpret the results.
> For each problem, the figures show the results of two experiments with different parameter settings specified in their caption. For each experiment, we have a 2 by 2 presentation of the trajectories of the primal value and the FW gap over the number of iterations and time.
> The FW gap is an upper bound on the primal gap and, therefore, a commonly used optimality measure.
>
> > It would be beneficial to add a direct speed comparison between the algorithms. How much faster is your approach at reaching a specified gap value?
>
> We will include a performance profile together with additional statistics in the appendix of the paper to offer an aggregated view of the performance of the algorithm.
>
> ### Questions
> > - In algorithm 1, the global FW vertex uses V(P) which is undefined (I assume you mean V(X)?)
> > - In 2.1 you don’t prove your theorems, you only state them, so you should write “in the following we state two theorems”. It’s fine to have proofs in the appendix if they aren’t particularly enlightening.
>
> We thank the referee for their careful reading and will update the paper following the suggestions.

---

> > ### Comment · Reviewer_4JW7 · 2025-08-05
> > **Response to Rebuttal**
> >
> > The two main issues I had with your paper were:
> > 1. Clarity - I think your response is adequate to this point. I still think the paper will come out very dense, and will require some major modifications - which I’ll only see after the reviewing process is finished. I’ve increased my score to 3 and not 4, because of the scope of the changes I think are required. Also I reiterate my recommendation that you take extra care in explaining the key figures in your paper - improving the presentation of your results could drastically increase the clarity of your work. Maybe show fewer figures showing the same thing, and replace the others with the aggregate view you decided to put in the appendices.
> > 2. Experimental quality - your new set of experiments is very thorough and address my concerns well. I’ve increased the quality score to 4.
> >
> > I wish you luck in getting this work accepted.

---

> > > ### Author Response · Authors · 2025-08-07
> > >
> > > We want to thank the reviewer for their constructive feedback and for increasing the scores based on our revisions. We are glad that the new experimental results adequately address the concerns regarding experimental quality.
> > >
> > > Regarding the plots, we fully agree that clarity is essential and thank the reviewer for their suggestions. We will pay special attention to improving their presentation in the revision. In particular, we plan to reduce redundancy and include a performance profile to provide a more aggregated view of the results.

---

### Official Review · Reviewer_eCQa · 2025-07-03

**Clarity:** 2
**Significance:** 2
**Originality:** 3
**Rating:** 4
**Confidence:** 3

**Summary:**

The authors propose a Frank-Wolfe algorithm with corrective steps. Their modification relies on a relaxed optimality condition, as detailed in Algorithm 2. They then specialize the proposed algorithm and establish convergence guarantees for quadratic functions minimized over an affine hull.

The paper also presents results for constraint sets formed by the intersection of two convex sets. Finally, it extends the analysis from optimizing strongly convex functions over polytopes to self-concordant functions.

The main message of the paper is that the proposed modification accelerates the algorithm, and this claim is supported by numerical experiments presented in the final section.

**Questions:**

1. Could the authors clarify whether their claim is that they analytically outperform existing algorithms for a specific family of functions or constraint sets? Or is it primarily an intuition they aim to convey by guiding the reader through the steps of the algorithm? In particular, could the authors be more precise about the following statements:
“It converges within a single step to the optimal value as soon as …” and “This is both computationally more efficient and avoids dropping too many potentially good vertices from the active set.”

2.  In Section 3.1, which algorithm was ultimately used? Is it the one from [1] with a modified step size? If it is exactly the algorithm from [1], what is the relationship between this section and the rest of the paper? If it is not the same as [1], why is this not clearly stated in the theorem’s statement?

3. It would be helpful if the authors could specify what the step size was in [1] and what the main technical challenges are in improving the logarithmic factor beyond simply adjusting the step size.

4. In Section 3.2, the authors state:
“Unlike regular FW variants, CGS requires only
$\mathcal{O}(\sqrt{\epsilon}$ gradient calls to achieve  $\epsilon$-optimality.” It would be useful if the authors could also mention the gradient call complexity required by standard Frank-Wolfe algorithms for comparison.

5. Regarding the result mentioned in Section 3.2, “we extend this rate to generalized self-concordant functions,”
have the authors actually extended the algorithm to self-concordant functions? Could they please clarify this part and elaborate on the technical challenges involved?

6. Regarding the results for quadratic functions, could you generalize your analysis to quadratic functions with a positive semi-definite matrix instead of a positive definite one? Additionally, instead of restricting to strongly convex functions, could you provide comments on functions that satisfy the Polyak-Łojasiewicz (PL) condition?



[1] Zev Woodstock and Sebastian Pokutta. Splitting the conditional gradient algorithm. SIAM Journal on Optimization.

**Ethical Concerns:**

["NO or VERY MINOR ethics concerns only"]

**Final Justification:**

The authors have addressed all of my concerns; therefore, I am maintaining my score.

**Limitations:**

I address the limitations that I have in mind in the previous sections..

**Paper Formatting Concerns:**

I have no formatting concerns.

**Quality:**

2

**Strengths And Weaknesses:**

Weaknesses:

The main weakness of the paper is that some of its claims feel vague and hand-wavy, lacking concrete justification. I attempt to clarify these points by raising questions in the following sections.

Strengths:

Aside from these concerns, the algorithms and technical proofs are presented very clearly and are a pleasure to read. The literature review covers most of the important related works.

---

> ### Author Rebuttal · Authors · 2025-07-31
>
> We thank the referee for their encouragement and feedback and address several points below.
>
> ### Weaknesses:
>
> > The main weakness of the paper is that some of its claims feel vague and hand-wavy, lacking concrete justification. I attempt to clarify these points by raising questions in the following sections.
>
> Thank you for pointing out that some of our claims lacked clarity. We will revise the manuscript accordingly to make the arguments more precise and to provide clearer, more concrete explanations where needed.
>
> ### Questions:
>
> > Could the authors clarify whether their claim is that they analytically outperform existing algorithms for a specific family of functions or constraint sets? Or is it primarily an intuition they aim to convey by guiding the reader through the steps of the algorithm?
>
> Our main result is that the novel quadratic corrections outperform existing FW methods for quadratic convex objectives, provably in certain settings we expand below.
> Computationally, the experiments provide empirical evidence for this performance.
>
> We can prove that under suitable assumptions, CFW with QC steps converges in a finite number of iterations to the optimum, a strong result that otherwise requires fully-corrective FW (FCFW). This finite-time convergence to the optimum is independent of any suboptimality tolerance; we reach the optimum up to floating-point error in finite time.
> We will include this additional result in the corrected version of the paper and state it below:
>
> The affine minimizer computed in equation (6) lies in the convex hull of the active set if the optimal face is completely contained in this convex hull.
> In consequence, both quadratic correction steps perform a fully corrective step, which in this case yields the overall optimal point in $\mathcal{X}$. Thus, quadratic corrections reduce the time of convergence to the optimum to the time until the optimal face is contained in the convex hull of extreme points in the active set.
> Assuming $\mathcal{X}$ is a polyhedron, the function is convex and smooth, and strict complementarity holds, we can leverage the fact that AFW and BPCG identify the optimal fact in a finite number of iterations, a result proved in [2] and extended to BPCG in [3]. Identification here means that all vertices contained in the active set of $S_t$ are contained in the optimal face after this finite number of iterations.
> This identification result extends to CFW with QC steps since they operate local pairwise and FW steps, which constitute BPCG, and QC steps, which do not add new extreme points to the active set.
> Grouping the two results, we can conclude that CFW with QC steps converges in finitely many iterations given the additional assumptions and therefore provably outperforms existing methods while requiring only tractable steps.
>
> This result will be added to Section 2 of the revised paper.
>
> > In particular, could the authors be more precise about the following statements: “It converges within a single step to the optimal value as soon as …” and “This is both computationally more efficient and avoids dropping too many potentially good vertices from the active set.”
>
> The theoretical result relating to the first statement is explained in detail in the previous section.
> We will formulate the given statement more explicitly in the revised version of the paper.
>
> Regarding the second statement, we will also clarify the benefits of QC-MNP over the MNP-Correction step from [4].
> Unlike QC-MNP, the MNP correction step performs multiple rounds of the affine minimization and truncation until the affine minimizer lies inside the convex hull. Thus, QC-MNP comes with a smaller per-iteration cost. Furthermore, both methods can drop vertices from the optimal face in the worst case. Therefore, multiple rounds of truncation in the MNP correction step can become costly in terms of LMO calls.
> Finally, and maybe most importantly, the MNP correction step does not specify how the affine minimization is performed.
> In the case of quadratic objectives, our method can be directly used for this.
>
>
> > In Section 3.1, which algorithm was ultimately used? Is it the one from [1] with a modified step size? If it is exactly the algorithm from [1], what is the relationship between this section and the rest of the paper? If it is not the same as [1], why is this not clearly stated in the theorem’s statement?
>
> We will improve the explanation of how Section 3.1 connects with the rest of the paper.
> The algorithm we propose here is the modified step size, which improves the convergence guarantees of SCG.
> Theorem 5 performs Split Conditional Gradient (SCG) with standard FW steps, thus it does not rely on CFW.
> If the original function is quadratic, SCG performs a FW step on a quadratic subproblem changing at each iteration,
> we therefore propose to use QC to accelerate convergence.
> The modified step size and resulting accelerated convergence were added to improve the algorithm, besides the acceleration by QC steps, complementing the computational acceleration from QC with an algorithmic improvement.
>
>
> >It would be helpful if the authors could specify what the step size was in [1] and what the main technical challenges are in improving the logarithmic factor beyond simply adjusting the step size.
>
> The main technical challenge is the analysis of an alternative auxiliary objective, which ultimately allows us to use a smaller step size, which would not have worked in the original proof. Furthermore, we simplify steps of the original proof by providing an explicit schedule of the penalty parameter $\lambda$ instead of the recursive one. We will state these challenges in the corresponding paragraph.
>
>
> >In Section 3.2, the authors state: “Unlike regular FW variants, CGS requires only $\mathcal{O}(\sqrt{\epsilon})$ gradient calls to achieve $\epsilon$-optimality.” It would be useful if the authors could also mention the gradient call complexity required by standard Frank-Wolfe algorithms for comparison.
>
> Standard FW methods compute the gradient in each iteration, therefore require $\mathcal{O}(\epsilon)$ many gradient calls in the general convex, smooth setting. We will add this fact to the given paragraph.
>
>
> > Regarding the result mentioned in Section 3.2, “we extend this rate to generalized self-concordant functions,” have the authors actually extended the algorithm to self-concordant functions? Could they please clarify this part and elaborate on the technical challenges involved?
>
> We kept the improved convergence result in the appendix due to space limitations.
> In the revised version, we will add a simplified version in the given section.
> We established that SOCGS converges and established the linear convergence rate for generalized self-concordant functions that are not globally Lipschitz-smooth. The challenge lay in the replacement of properties resulting from Lipschitz smoothness with inequalities derived from the generalized self-concordance.
>
>
> > Regarding the results for quadratic functions, could you generalize your analysis to quadratic functions with a positive semi-definite matrix instead of a positive definite one? Additionally, instead of restricting to strongly convex functions, could you provide comments on functions that satisfy the Polyak-Łojasiewicz (PL) condition?
>
> Our analysis does not require positive definiteness of the quadratic objective; both quadratic correction steps only assume convexity, so they remain applicable when the objective is defined by a positive semi-definite (PSD) matrix. In particular, the results extend to general convex quadratic functions, even when the Hessian is singular.
>
> Regarding the PL condition: although we do not assume it directly, we prove linear convergence under the assumption that the objective is 1/2-sharp. For smooth objectives, 1/2-sharpness implies the Polyak–Łojasiewicz (PL) condition, so our results cover a broad class of functions beyond the strongly convex case, including those satisfying PL.
>
> The only place in the paper where we do require strong convexity is the analysis of SOCGS; indeed, a strongly convex function is required and inherited from the original SOCGS paper, since the Hessian needs to induce a metric. It is an open question whether this assumption could be removed in future work; communication with the SOCGS authors confirmed that removing this assumption prevents the analysis from being carried over.
>
> ### References
>
> [1] Splitting the conditional gradient algorithm, Zev Woodstock and Sebastian Pokutta. SIAM Journal on Optimization.
>
> [2] Active set complexity of the away-step Frank-Wolfe algorithm, Immanuel Bomze, Francesco Rinaldi, Damiano Zeffiro, SIAM Journal on Optimization.
>
> [3] The pivoting framework: Frank-Wolfe algorithms with active set size control, Elias Wirth, Mathieu Besançon, Sebastian Pokutta, AISTATS 2025.
>
> [4] On the Global Linear Convergence of Frank-Wolfe Optimization Variants, Simon Lacoste-Julien, Martin Jaggi, 2015

---

> ### Comment · Reviewer_eCQa · 2025-08-05
>
> I wish to thank the authors for their responses. I agree with most of the points raised by the other reviewers. I strongly encourage the authors to carefully address the suggestions made by the reviewers, as incorporating these points will significantly improve the clarity of the paper. Given the authors’ promised revisions, my score remains unchanged.

---

> > ### Author Response · Authors · 2025-08-05
> >
> > We want to thank the reviewer again for their valuable feedback. As outlined in our initial response, we will include the additional theoretical results concerning optimal face identification and finite convergence properties. Furthermore, we will clarify the convergence result of SCG and its connection to quadratic corrections. We will also include a simplified version of the generalization of SOCGS to self-concordant functions in the corresponding section.

---

### Official Review · Reviewer_K84e · 2025-07-03

**Clarity:** 3
**Significance:** 3
**Originality:** 3
**Rating:** 4
**Confidence:** 2

**Summary:**

This work develops a variant of the Frank-Wolfe algorithm, which they refer to as the Corrective Frank-Wolfe (CFW) algorithm. The authors claim it to be a generalization of a previous variant called the Fully Corrective Frank-Wolfe  (FCFW) Algorithm. The authors claim that FCFW is often computationally impractical and their proposed CFW method is a means to overcome this challenge. Their main results derive convergence rates for this algorithm and its lazified variant. As part of their CFW framework, the work also designs two quadratic correction steps: (i) QC-LP which involves a direct linear program and (ii) QC-MNP which is a specialized variant of the minimum-norm point algorithm. With these new quadratic corrections, the authors show that the Split Conditional Gradient (SCG) algorithm enjoys a faster convergence rate of $O(1/\sqrt{t}),$ improving a $\log t$ factor from the previously known result. Secondly, for Second-Order Conditional Gradient Sliding (SOCGS), the authors show global convergence for generalized self-concordant functions. Finally, the work empirically demonstrates the computational benefits of their quadratic corrections on four specific problem instances.

**Questions:**

1. In Algorithm 2, can you explictly highlight when the drop step would be executed and when the descent step would be executed?

2. It is unclear how the three steps in Proposition 1 are to be utilized. Should they be tried one by one in order until either the drop or descent step criterion is met?

3. I am new to this field. However, in my understanding, the convergence rates in Theorem 2 and 3 do not appear to be new. So, what is the benefit of your proposed algorithm? Shouldn't you be quantifying the reduction in the computational effort to demonstrate the superiority of your algorithm?

**Ethical Concerns:**

["NO or VERY MINOR ethics concerns only"]

**Final Justification:**

The work presents a framework of corrective steps that unifies already existing Frank-Wolfe algorithms, which I find to be interesting and useful. Despite being an outsider, I found the paper well-written.

However, there are some aspects that the current draft doesn't address, such as computational benefits of using the new corrective step.

Accordingly, I recommend a borderline accept for this paper.

**Limitations:**

Yes

**Quality:**

4

**Strengths And Weaknesses:**

**Strengths**:

1. Extremely well-written paper.
2. The proposed algorithm introduces several corrective steps that can potentially improve computational efficiency.
3. Empirical results demonstrate significant improvement in iteration complexity over BPCG across a range of tasks.

**Weaknesses**:

1. The theoretical results do not quantify the computational benefits of using the new corrective steps. In particular, it is unclear why the corrective steps lead to computational benefits over the traditional FW update rules.

**Minor issues**:

1. The first two sentences in your abstract are too long. Please break them into smaller, easy-to-digest

2. Explicitly define the vertex set of a compact convex set in the preliminaries section.

3. What is $V(P)$ in Step 5 of your algorithm? I presume you meant $V(\mathcal{X})$.

---

> ### Author Rebuttal · Authors · 2025-07-31
>
> We thank the referee for their encouragement and feedback and address several points below.
>
> ### Weaknesses
>
> >The theoretical results do not quantify the computational benefits of using the new corrective steps. In particular, it is unclear why the corrective steps lead to computational benefits over the traditional FW update rules.
>
> We will strengthen the theoretical results, proving that CFW with QC can converge in some settings in finite time to an optimal solution, independently of any suboptimality tolerance, a strong result that otherwise requires fully-corrective FW (FCFW), which is in general not tractable.
> We will include this additional result in the corrected version of the paper and state it below:
>
> The affine minimizer computed in equation (6) lies in the convex hull of the active set if the optimal face is completely contained in this convex hull.
> In consequence, both quadratic correction steps perform a fully corrective step, which in this case yields the overall optimal point in $\mathcal{X}$. Thus, quadratic corrections reduce the time of convergence to the optimum to the time until the optimal face is contained in the convex hull of extreme points in the active set.
> Assuming $\mathcal{X}$ is a polyhedron, the function is convex and smooth, and strict complementarity holds, we can leverage the fact that AFW and BPCG identify the optimal fact in a finite number of iterations, a result proved in [1] and extended to BPCG in [2]. Identification here means that all vertices contained in the active set of $\mathcal{S}_t$ are contained in the optimal face after this finite number of iterations.
> This identification result extends to CFW with QC steps since they operate local pairwise and FW steps, which constitute BPCG, and QC steps, which do not add new extreme points to the active set.
> Grouping the two results, we can conclude that CFW with QC steps converges in finitely many iterations given the additional assumptions and therefore provably outperforms existing methods while requiring only tractable steps.
>
>
> ### Minor issues:
>
> We thank the referee for their careful reading and suggestions to improve our manuscript.
>
> >The first two sentences in your abstract are too long. Please break them into smaller, easy-to-digest
>
> In the revised version, we will split the sentences as suggested.
>
> >Explicitly define the vertex set of a compact convex set in the preliminaries section.
>
> We agree that the notion of the vertices of general convex sets is not clear. Therefore, we have generalized the definition of the set $V(\mathcal{X})$ for non-polyhedral sets to be the set of extreme points of $\mathcal{X}$, i.e., the points that cannot be written as a strict convex combination of points in $\mathcal{X}$.
>
>
> > What is V(P) in Step 5 of your algorithm? I presume you meant V(X)
>
> We thank the reviewer for their careful reading and will fix this typo in the revised version.
>
> ### Questions
>
> >In Algorithm 2, can you explicitly highlight when the drop step would be executed and when the descent step would be executed?
>
> Algorithm 2 is only a template for possible correction steps, giving conditions for valid corrections, ensuring that CFW converges. Therefore, it depends on the actual corrective step when a drop step or descent step would be executed. Furthermore, these two steps also do not exclude each other, i.e. a corrective step could meet both conditions at the same time. We will highlight that this is a template of requirements for a corrective step in the revised manuscript.
>
> >It is unclear how the three steps in Proposition 1 are to be utilized. Should they be tried one by one in order until either the drop or descent step criterion is met?
>
> Proposition 1 shows that the mentioned steps satisfy the conditions of corrective steps; hence, the criteria do not need to be checked during runtime. Therefore, we conclude that these algorithms suit the framework of corrective FW. Furthermore, we mention in remark 2 how these steps can be used as fallback options for new steps, which do not satisfy the conditions of corrective steps in each iteration.
>
>
> >I am new to this field. However, in my understanding, the convergence rates in Theorem 2 and 3 do not appear to be new. So, what is the benefit of your proposed algorithm? Shouldn't you be quantifying the reduction in the computational effort to demonstrate the superiority of your algorithm?
>
> The idea of CFW is to present a framework of corrective steps that unifies already existing algorithms through the concepts of local updates on the active step.
> By providing proofs for convergence rates in two problem settings, we encourage other researchers developing new algorithms for which they only have to check that their corrective steps meet the mild conditions in Algorithm 2.
> Quadratic corrections, however, enjoy faster convergence guarantees under the additional assumption that the objective is convex quadratic. CFW with QC steps converges to the optimum in finite time if $f$ is convex and smooth, strict complementarity holds, and $\mathcal{X}$ is a polytope.
>
> [1] Active set complexity of the away-step Frank-Wolfe algorithm, Immanuel Bomze, Francesco Rinaldi, Damiano Zeffiro, SIAM Journal on Optimization.
>
> [2] The pivoting framework: Frank-Wolfe algorithms with active set size control, Elias Wirth, Mathieu Besançon, Sebastian Pokutta, AISTATS 2025.

---

> > ### Comment · Reviewer_K84e · 2025-08-05
> >
> > > We will strengthen the theoretical results
> >
> > Regarding the additional result, I would need to see the details before making comments.
> >
> > ---
> > Regardless, I have looked at the comments of other reviewers and the authors response. I find them satisfactory. Hence, I retain my score.

---

> > > ### Author Response · Authors · 2025-08-06
> > >
> > > We want to thank the reviewer for their careful reading and maintaining the positive evaluation. As mentioned, we will include a new the theoretical result in the revised version of the manuscript. In particular, we will add the following theorem, which establishes a finite convergence guarantee under a strict complementarity condition:
> > >
> > > Let $\mathcal{X}$ be a polytope and $f \colon \mathcal{X} \to \mathbb{R}$ be a convex, $L$-smooth function.
> > > Let $\mathcal{X}^* $ be the set of minimizers of $f$ over $\mathcal{X}$ and $A^* $ be the minimal face containing $\mathcal{X}^* $.
> > > Assume strict complementarity holds, i.e., there exists $\delta > 0$ such that for $x^* \in \mathcal{X}^* $ and $v \in V(\mathcal{X})$ we have $\langle \nabla f(x^* ), v - x^* \rangle = 0$ if $v \in A^* $ and $\langle \nabla f(x^* ), v - x^* \rangle > \delta$ otherwise.
> > > Then, CFW with either QC-LP, QC-MNP or full correction steps converges to an optimal solution in finitely many iterations.
> > >
> > > We believe this result offers meaningful theoretical support for the practical advantages of using corrective steps.

---

### Official Review · Reviewer_i92J · 2025-07-04

**Clarity:** 4
**Significance:** 3
**Originality:** 3
**Rating:** 5
**Confidence:** 4

**Summary:**

This paper introduces approximate versions of the fully-corrective Frank-Wolfe
(FCFW) method applied to quadratic objectives.
First, the authors develop a framework for general corrective
steps and prove fast convergence rates for their algorithm, which they call
corrective Frank-Wolfe (CFW).
Then they propose two quadratic corrections which specialize their algorithm:
Quadratic Correction LP, which solves a linear program and then checks if
it yields a feasible point, and Quadratic Correction MNP, which solves
a linear system and computes the closes feasible point along the chord between
the solution and the current iterate.
Experiments show that these corrective steps outperform significantly outperform
blended pairwise conditional gradients (BPCG) on synthetic and real-world problems.
In addition, the authors develop a faster convergence rate for split
condition gradient (shaving off a log term) and also show how to incorporate
lazy evaluation of the linear minimization oracle into their framework.

**Questions:**

- Algorithm 1: I assume that every extreme point of $\mathcal{X}$ is treated as
  a vertex if $\mathcal{X}$ is not polyhedral? Or do you require a polyhedral
  constraint set?

- Algorithm 1, Line 5: What is $P$? Should this be $\mathcal{X}$?

- Algorithm 1, Line 10: Do you have perform this minimization exactly?

- Algorithm 2: Where do $s, v$ get used? Aren't these equivalent to $u, w$ in
    the descent step?

- Algorithm 3: How often can we expect Eq. (7) to return an interior point
    (see also detailed discussion)?

**Ethical Concerns:**

["NO or VERY MINOR ethics concerns only"]

**Final Justification:**

The authors addressed my concerns. They include new baseline comparisons and an empirical study of the success rate for QC steps. As a result, I raise my scores to 5.

**Limitations:**

Limitations are reasonably addressed with the exception of using only a single
experimental baseline.
The authors should provide a strong argument justifying this choice in the
paper or they should extend their comparison to include additional methods.

**Paper Formatting Concerns:**

See minor comments.

**Quality:**

3

**Strengths And Weaknesses:**

**Strengths**

- The theoretical framework for CFW is general and may encourage other
  researchers to develop corrective steps for non-quadratic objectives.

- The quadratic corrective steps work very well compared to BPCG, both
    in terms of iterations and wall-clock time.

- Corrective steps can also be applied to split conditional gradient and
    and conditional gradient sliding.

- The authors improve the best known convergence rate for split conditional
    gradient.

- The paper is well-written.

**Weaknesses**

- The corrective steps the authors propose only apply to quadratic objectives.
    This may not be a strong restriction since Frank-Wolfe requires a linear
    minimization oracle, which already restricts the function class in practice.

- No evaluation of the success rates (i.e. how often they reduce to FCFW steps)
    of the quadratic corrections is provided.

- The experiments only compare to one baseline algorithm, BPCG. It's
    not clear how much CFW under-performs the ideal method (FCFW) and how
    it compares to similar similar methods, like approximate FCFW or pairwise
    Frank-Wolfe.


### Detailed Comments

I am not an expert in Frank-Wolfe methods, so it is difficult for me to judge
the novelty and potential impact of this paper.
However, I found it interesting to read and I think there are several clever
ideas here which will be useful to other researchers.
I particularly like that CFW is presented in a general framework, which will
encourage other researchers to develop new corrective steps for more general
objectives.
With this said, I think there are several important issues (discussed in detail
below) whose resolution will improve the paper.
Thus, I will recommend weak accept for the moment, with the option to raise
my score if these issues are resolved satisfactorily.

**Limitation to Quadratics**:
While the theoretical results for CFW apply to any corrective steps which
satisfy the structure of Algorithm 2, the two proposed corrections are only
useful for quadratic objective functions.
The authors are straightforward about this limitation, which is refreshing.
Moreover, Frank-Wolfe methods already require access to a linear minimization
oracle (which restricts the applicable function class in practice), so this
may not be too significant a drawback.

**Limited Experimental Comparison**:
I list that CFW significantly outperforms the BPCG baseline as a strength,
which it is.
Yet, the main weaknesses of this paper is still the experimental comparison.
Comparing to only one baseline in a crowded area like Frank-Wolfe methods
is limited at best and misleading at worst.
CFW is an approximation to FCFW, yet there are no experiments presented in the
main paper comparing these two methods.
The authors also draw attention to approximate FCFW and pairwise Frank-Wolfe as
the two closest methods to theirs, but then don't compare them experimentally.
I think this is a glaring problem in a paper which advertises "the excellent
computation benefits" of their approach.

**Success Rates of Quadratic Corrections**:
The authors mention in Section 4.2 that "the LP in the QC-LP is almost never
feasible, leading to pairwise steps and thus a similar trajectory as BPCG".
This raises a key question which isn't addressed theoretically or empirically:
how often do the quadratic correction methods succeed and return feasible
solutions which correspond to FCFW steps?
Section 4.2 shows that sometimes they mostly fail, meaning solving the
LP/linear system is a waste of computation overall.
While theoretical analysis may impractical, the paper would be strengthened by
including ablation studies on the success of proposed quadratic corrections.

### Minor Comments

- Line 65: It would be helpful to define the MNP algorithm. Is this Wolfe's
    minimum-norm point that was mentioned in the abstract? Edit: I see this
    is introduced in Line 87 but I think it would better to state it here.

- Line 72: The large number of acronyms in this paragraph make the discussion
  difficult to follow. Some paragraphs later on also devolve into "acronym
  soup".

- Line 118: It would be useful to define $a_t, s_t$, and $x_t$ outside of
    Algorithm 1. Otherwise the reader has to leave the first paragraph
    of Section 2, reader Algorithm 1, and then combine back in order to
    understand the discussion.

- Algorithm 2: I interpret $S' \subset S$ as non-strict inclusion equivalent to
  $S' \subseteq S$. The drop step seems to imply $S' \subset S$ only in that
  case.

- Line 168: I think it's simpler and clearer to say "writing $x^*$ as an affine
    combination of vertices.

- Algorithms 3 and 4: I think the presentation would be better if these were
    top-aligned.

- Line 176: "The QC-LP approach assumes enforces the" --- unnecessary word?

- Algorithm 3: $\tilde{S}$ appears to define the solution $x^*$ in barycentric
    coordinates. If this is true, how does the local pairwise step
    reduce the cardinality compared to $S$ as required by Algorithm 2?
    Or is the local pair-wise step guaranteed to make sufficient progress?

- Algorithms 3, 4: It would be helpful if the return values matched Algorithm 2.

- Line 192: is it ever possible that the number of atoms exceeds the dimension
    of the atoms? In this case, it will be more efficient to solve directly
    in terms of $x$, so you can depend on minimum of the dimension and
    the size of the active set. Is this meaningful at all?

- Theorem 5: This result fits awkwardly into the current story. Maybe
    you can remind the reader before this paragraph that each
    sub-step of SCG can be combined with the ideas from CFW to obtain
    faster convergence. Thus, it is desirable to improve the rates for
    SCG and bring them into correspondence with the standard CFW framework.

- All Figures: the lines and font-sizes are too small to be legible without
    unreasonable zooming.

- Figure 1: The scale of the y-axis in the first row prevents these plots
    from showing any useful information.
    I suggest subtracting the optimal value (if you aren't already) and
    changing the y-limit to cut-off the first few iterations.

- Figure 2: Same comments as for Figure 1.

- Line 271: "Mentioning that quadratic corrections..."  --- was this sentence
    supposed to be an internal comment?

- Line 278: "We remind that each iteration..." --- Where does the quadratic
    sub-problem come from? Is this a feature of AFW? I know the authors
    have limited space, but providing more details would make this experiment
    easier to understand.

- I think you could cut down on the discussion in Section 1 to get more space.
    In particular, the contributions subsection is long and could
    be reduced by summarizing the just key contributions in a bulleted list.
    Three pages for the introduction is quite a lot for a NeurIPS paper;
    two pages is perhaps standard.

- It's not necessary to capitalize every word in an acronym when you introduce
    it. For example, in "Fully-Corrective Frank-Wolfe", only Frank-Wolfe
    should capitalized, since these are proper nouns.

---

> ### Author Rebuttal · Authors · 2025-07-31
>
> We thank the referee for their encouragement and feedback and address several points below.
> ### Weaknesses
> **Comment on Limitation to Quadratics:**
>
> The class of quadratic functions already encompasses many real-world applications, making our approach broadly relevant. Moreover, Quadratic Correction (QC) steps can be used to accelerate algorithms that depend on solving quadratic subproblems via FW-type approaches, including SCG and SOCGS.
> In particular, we have observed significant potential for QC to improve the convergence of SOCGS when optimizing general convex functions.
> Beyond SCG and SOCGS, QC steps are also applicable for submodular function minimization [1] and Frank-Wolfe variants of the Difference Convex algorithm [2,3].
> Finally, we note that requiring access to a linear minimization oracle (LMO) primarily restricts the feasible region, rather than the objective function class.
>
> **Comment on Limited Experimental Comparison:**
>
> We will add results for additional baseline algorithms, including AFW, PFW, and an active set-based variant of vanilla FW.
> To ensure a fair comparison, we also augmented each of these methods with QC steps using the same hybrid strategy previously applied to BPCG. We present below the results for the first experiment showing the absolute and relative values of the runtime in seconds and the number of iteration. The relative values in parentheses compare to the corresponding baseline. The full evaluation will be included in the revised version of the paper.
>
> |Runtime in seconds|FW|AFW|PFW|BPCG|FW-LP|AFW-LP|PFW-LP|BPCG-LP|FW-MNP|AFW-MNP|PFW-MNP|BPCG-MNP|
> |-|-|-|-|-|-|-|-|-|-|-|-|-|
> |$K$-Sparse regression $n=500$, $K=5$|2604|1673|1265|1451|**490** (0.19)|534 (0.32)|1197 (0.95)|731 (0.50)|945 (0.36)|950 (0.57)|1374 (1.09)|1130 (0.78)|
> |$K$-Sparse regression $n=500$, K=20$|148|169|222|206|**112** (0.75)|139 (0.83)|249 (1.12)|178 (0.86)|112 (0.75)|146 (0.87)|242 (1.09)|186 (0.90)|
> |$K$-Sparseregression $n=1000, K=5$|3600|3600|3600|3600|3566 (0.99)|**1433** (0.40)|3486 (0.97)|2001 (0.56)|2503 (0.70)|2683 (0.75)|3576 (0.99)|2882 (0.80)|
> |$K$-Sparseregression $n=1000, K=20$|1028|1028|979|1073|486 (0.47)|**548** (0.53)|1454 (1.49)|642 (0.60)|627 (0.61)|654 (0.64)|1467 (1.50)|736 (0.69)|
>
> |Number of iterations|FW|AFW|PFW|BPCG|FW-LP|AFW-LP|PFW-LP|BPCG-LP|FW-MNP|AFW-MNP|PFW-MNP|BPCG-MNP|
> |-|-|-|-|-|-|-|-|-|-|-|-|-|
> |$K$-Sparse regression $n=500, K=5$|10000|5225|3291|3710|**388** (0.04)|431 (0.08)|899 (0.27)|611 (0.16)|**388** (0.04)|431 (0.08)|904 (0.27)|611 (0.16)|
> |$K$-Sparse regression $n=500, K=20$|481|492|784|668|**165** (0.34)|225 (0.46)|414 (0.53)|254 (0.38)|**165** (0.34)|225 (0.46)|408 (0.52)|254 (0.38)|
> |$K$-Sparse regression $n=1000, K=5$|7431|7480|7401|6718|1123 (0.15)|747 (0.10)|1473 (0.20)|1066 (0.16)|**631** (0.08)|700 (0.09)|1159 (0.16)|988 (0.15)|
> |$K$-Sparse regression $n=1000, K=20$|2010|2032|1939|1896|**315** (0.16)|380 (0.19)|736 (0.38)|411 (0.22)|**315** (0.16)|380 (0.19)|721 (0.37)|411 (0.22)|
>
> Furthermore, FCFW is an idealized algorithm lacking a practical implementation of the correction step for general convex functions. This was also one of the motivations for the QC steps, providing a specific algorithm for one problem class that can be performed efficiently and without relying on a tolerance or working limit for the correction step.
>
> Finally, approximate FCFW provides a general framework similar to CFW. The focus of our computational analysis lies on quadratic corrections. The idea of CFW is to provide a flexible framework that unifies already existing methods. Therefore, we do not include a direct experimental comparison between CFW and approximate FCFW.
>
> **Comment on Success Rates of Quadratic Corrections:**
>
> To provide insight into the computational benefit of QC steps, we measured success rates—i.e., how often the affine minimizer lies within the convex hull of the current active set. Therefore, we not only measure how often the LP for QC-LP is feasible, but also how often the QC-MNP step is truncated.
> We will provide a table with the per-experiment success rates in the appendix of the paper and present an excerpt from this here:
>
> |Experiment|QC-LP|QC-MNP|
> |-|-|-|
> |$K$-sparse regression $K=5$|37 / 37|37 / 37|
> |$K$-sparse regression $K=20$|15 / 15|15 / 15|
> |Entanglement detection $a=0.5$|2 / 557|6 / 526|
> |Entanglement detection $a=0.25$|2 / 599|7 / 577|
> |Birkhoff projection $n=300$|13 / 17|15 / 23|
> |Birkhoff projection $n=500$|19 / 19|19 / 19|
>
> While in the $K$-sparse regression and the Birkhoff projection task, both QC steps are almost always successful, the affine minimizer usually lies outside the convex hull in the entanglement detection experiment.
>
> Furthermore, we would like to emphasize that the scheduling of the QC steps is an active topic of further research. Instead of using a fixed schedule, as in our current experiments, one could employ adaptive strategies that consider the current size of the active set or the success rate of previous QC to avoid computational overhead, as in the mentioned experiment.
>
> Finally, we would like to point to an additional theoretical result that relates to this topic.
> One can show that under suitable assumptions, the affine minimizer computed in QC steps lies in the convex hull of the active set if the optimal face is completely contained in this hull. In consequence, both quadratic correction steps perform a fully corrective step, which in this case yields the overall optimal point in $\mathcal{X}$. Thus, quadratic corrections reduce the time of convergence to the optimum to the time until the optimal face is contained in the convex hull of extreme points in the active set.
>
> We will include this result in the revised version and expand on a strong theoretical consequence of this:
> Assuming $\mathcal{X}$ is a polyhedron, the function is convex and smooth, and strict complementarity holds, we can leverage the fact that AFW and BPCG identify the optimal fact in a finite number of iterations, a result proved in [4] and extended to BPCG in [5]. These results extend to CFW with QC steps since they operate local pairwise and FW steps, which constitute BPCG, and QC steps, which do not add new extreme points to the active set.
> In consequence, CFW with QC steps converges in finitely many iterations given the additional assumptions, a strong result that otherwise requires fully-corrective FW.
>
>
> ### Minor comments
>
> We thank the reviewer for the detailed list of editorial suggestions, which helped us to improve the quality and clarity of the paper. There is one content-related comment, which we would like to address:
>
> >Line 192: is it ever possible that the number of atoms exceeds the dimension of the atoms? In this case, it will be more efficient to solve directly in terms of x, so you can depend on minimum of the dimension and the size of the active set. Is this meaningful at all?
>
> A simple example where the number of atoms exceeds the dimension arises with the unit cube in $\mathbb{R}^2$, where FW selects at least three vertices when the minimizer lies in the interior. In practice, this gap between the dimension and the active set size can be even larger.
> Furthermore, as stated in line 199, one can use the pivoting framework from [5] to decrease the number of atoms to the dimension of the spanned subspace, which is usually much smaller than the full dimension in practice. Empirical results in [5] also showed that BPCG never maintains more vertices than necessary to represent the current iterate.
>
> ### Questions
>
> >Algorithm 1: I assume that every extreme point of $\mathcal{X}$ is treated as a vertex if $\mathcal{X}$ is not polyhedral? Or do you require a polyhedral constraint set?
>
> No, we do not require $\mathcal{X}$ to be a polyhedral constraint set.
> We generalized the definition of the set $V(\mathcal{X})$ for non-polyhedral sets to be $\mathcal{X}$, the set of extreme points, i.e., the points that cannot be written as a strict convex combination of points in $\mathcal{X}$.
>
> >Algorithm 1, Line 5: What is P? Should this be X?
>
> Yes, this is a typo and will be corrected in the revised version.
>
> >Algorithm 1, Line 10: Do you have perform this minimization exactly?
>
> An exact line search is not required; there exist several step size strategies guaranteeing sufficient descent ([6,7]), which also apply to CFW. We will comment on this in the paper.
>
> >Algorithm 2: Where do s, v get used? Aren't these equivalent to u, w in the descent step?
>
> We agree, $u$ and $w$ are exactly the local FW vertex $s$ and the away vertex $a$. We will remove $u$ and $w$, simplifying the constraint.
>
> >Algorithm 3: How often can we expect Eq. (7) to return an interior point (see also detailed discussion)?
>
> The frequency with which the LP in (7) is feasible is problem-specific and can vary significantly, as demonstrated by the experiments on sparse regression and entanglement detection. However, we introduced the result that QC steps are successful when the optimal face is contained in the convex hull of the active set and therefore converge to the optimal point in a single additional step.
>
>
> ### References
>
> [1] Provable Submodular Minimization using Wolfe’s Algorithm, D. Chakrabarty, P. Jain, P. Kothari, 2014
>
> [2] Revisiting Frank-Wolfe for Structured Nonconvex Optimization, H. Maskan, Y. Hou, S. Sra, A. Yurtsever, 2025
>
> [3] Scalable DC Optimization via Adaptive Frank-Wolfe Algorithms, S. Pokutta, 2025
>
> [4] Active set complexity of the away-step Frank-Wolfe algorithm, I. Bomze, F. Rinaldi, D. Zeffiro, SIAM Journal on Optimization.
>
> [5] The pivoting framework: Frank-Wolfe algorithms with active set size control, E. Wirth, M. Besançon, S. Pokutta, AISTATS 2025.
>
> [6] Linearly Convergent Frank-Wolfe with Backtracking Line-Search, F. Pedregosa, G. Negiar, A. Askari, M. Jaggi, 2022
>
> [7] The Frank-Wolfe Algorithm: A Short Introduction, S. Pokutta, 2024

---

> > ### Comment · Reviewer_i92J · 2025-08-06
> >
> > Many thanks for including the additional experiment comparisons and information.
> >
> > Regarding the entanglement detection, it's interesting to see that the QC steps are rarely successful. Is there anything salient about this problem that explains why this happens?
> >
> > Regardless, my concerns have been addressed, so I will raise my score.

---

> > > ### Author Response · Authors · 2025-08-07
> > >
> > > We thank the reviewer for their valuable feedback and for updating their score. We also appreciate their interest in the entanglement detection problem. In this setting, the feasible region—the set of separable bipartite states—is not a polytope, unlike in our other experiments. Additionally, the linear minimization oracle used here is an inexact one. We believe that both factors may contribute to the lower success rate of the quadratic correction steps in this experiment.

---

### Decision · Program_Chairs · 2025-09-17

**Decision:**

Accept (poster)

**Comment:**

This paper first proposes Corrective Frank-Wolfe (CFW)---a new framework for FW variants that perform corrective steps on the active set, and proves linear convergence for smooth, sharp functions over polytopes. Then, as a detailed implementation of corrective steps in the CFW algorithm, two new Quadratic Correction (QC) steps tailored for quadratic objective functions have also been proposed. Moreover, the authors showed that the corrective steps can also be applied to existing algorithms, including split conditional gradient and conditional gradient sliding.

Reviewers appreciate the theoretical framework of CFW and the practical benefits of the quadratic corrections. Nonetheless, reviewers have also raised some concerns about the limitation of the corrective steps, the experiments compare only against BPCG (omitting other close Frank-Wolfe variants), and the clarity issue about some technical details (e.g., computational benefits of using the new corrective step).

After the rebuttal and discussion phase, most concerns have been addressed, and all reviewers give a positive score. Based on my own reading and the reviewers' evaluations, I recommend accepting this paper. The authors should also revise this paper carefully according to the reviewers' comments, especially the concerns about the experimental results and the clarity issues.